# On the Sample Complexity of One Hidden Layer Networks with Equivariance, Locality and Weight Sharing

**Arash Behboodi**                                                   *behboodi@qti.qualcomm.com*
*Qualcomm AI Research*[*]

**Gabriele Cesa**                                                      *gcesa@qti.qualcomm.com*
*Qualcomm AI Research*[*]

**Reviewed on OpenReview:** *https://openreview.net/forum?id=Q7aXOnEGgU*

## Abstract

Weight sharing, equivariance, and local filters, as in convolutional neural networks, are believed to contribute to the sample efficiency of neural networks. However, it is not clear how each one of these design choices contributes to the generalization error. Through the lens of statistical learning theory, we aim to provide insight into this question by characterizing the relative impact of each choice on the sample complexity. We obtain lower and upper sample complexity bounds for a class of single hidden layer networks. For a large class of activation functions, the bounds depend merely on the norm of filters and are dimension-independent. We also provide bounds for max-pooling and an extension to multi-layer networks, both with mild dimension dependence. We provide a few takeaways from the theoretical results. It can be shown that depending on the weight-sharing mechanism, the non-equivariant weight-sharing can yield a similar generalization bound as the equivariant one. We show that locality has generalization benefits, however the uncertainty principle implies a trade-off between locality and expressivity. We conduct extensive experiments and highlight some consistent trends for these models.

## 1 Introduction

In recent years, equivariant neural networks have gained particular interest within the machine learning community thanks to their inherent ability to preserve certain transformations in the input data, thereby providing a form of inductive bias that aligns with many real-world problems. Indeed, equivariant networks have shown remarkable performance in various applications, ranging from computer vision to molecular chemistry, where data often exhibit specific forms of symmetry. Equivariant networks are closely related to their more traditional counterparts, Convolutional Neural Networks (CNNs). CNNs are a particular class of neural networks that achieve equivariance to translation. The convolution operation in CNNs ensures that the response to a particular feature is the same, regardless of its spatial location in the input data. However, equivariance extends beyond just translation, accommodating a broader spectrum of transformations such as rotations, reflections, and scaling. Group Convolutional Neural Networks (GCNNs) Cohen and Welling (2016a) are the typical example of equivariant neural networks. The inductive bias introduced by equivariance, in the form of symmetry preservation, offers an intuitive connection to their generalization capabilities. By encoding prior knowledge about the structure of the data, equivariant networks can efficiently exploit the inherent symmetries, reducing sample complexity and improving generalization performance. This ability to generalize from a limited set of examples is crucial to their success. Their mathematical foundation is grounded in group representation theory, which provides a powerful framework to understand and design neural networks that are equivariant or invariant to the action of a group of transformations.

---

[*]Qualcomm AI Research is an in intiative of Qualcomm Technologies, Inc.

Besides the inductive bias of data symmetry, the generalization benefits of CNNs and GCNNs are additionally attributed to the weight-sharing implemented by the convolution operation and the locality implemented by the smaller filter size. In this paper, we study the impact of equivariance, weight-sharing, and locality on generalization within the framework of statistical learning theory. Our focus will be on neural networks with one hidden layer. Similar to works like Vardi et al. (2022); Magen and Shamir (2023), we believe that this study provides a first step toward a better understanding of deeper networks. Getting dimension-free generalization error bounds constitutes an important line of research in the literature. Following this line of work, we provide various dimension-free and norm-based bounds for one hidden layer networks. See Appendix J.1 for discussions on the desiderata of learning theory.

**Contributions.** We consider a class of one hidden layer networks where the first layer is a multi-channel equivariant layer followed by point-wise non-linearity, a pooling layer, and the final linear layer. We assume that the $\ell_2$-norm of the parameters of each layer is bounded. For architectures based on group convolution with point-wise non-linearity, we provide generalization bounds for various pooling operations that are entirely dimension-free. We obtain a similar norm-based bound for general equivariant networks. We also provide a lower bound on Rademacher complexity that shows the tightness of our bound. We extend the results to max-pooling, combining various covering number arguments for Rademacher complexity analysis. The bound is only dimension-dependent on the number of hidden layer channels and logarithmic-dependent on the group size; otherwise, it is independent of other dimensions. We also extend the result to multi-layer networks and discuss its limitations. We show that no gain is observed if the analysis is conducted for the networks parameterized in the frequency domain. When a layer replaces the equivariant layer with, not necessarily equivariant, weight-sharing, we also provide a dimension-free bound. By studying the bound, it can be seen that some particular weight-sharing schemes, although not all, can provide similar generalization guarantees. Next, we give another bound for networks with local filters and show that the locality can bring additional gain on top of equivariance. We will then show a trade-off between locality in the spatial and frequency domains, which is important for band-limited inputs. The uncertainty principle characterizes the trade-off. Finally, we provide the numerical verification of our bounds. The generalization bounds are all obtained using Rademacher complexity analysis. We relegate all proofs to the appendix.

**Notations.** We introduce some of the notations used throughout the paper. We define $[n] := \{1, \ldots, n\}$ for $n \in \mathcal{N}$. The term $\|\cdot\|$ refers to the $\ell_2$-norm, which for the space of matrices is the Frobenius norm. The spectral norm of a matrix $\boldsymbol{A}$ is denoted by $\|\boldsymbol{A}\|_{2\to2}$. A positively homogeneous activation function $\sigma(\cdots)$ is a function that satisfies $\sigma(\lambda x) = \lambda\sigma(x)$ for all $\lambda \geq 0$. The loss functions are assumed to be 1-Lipschitz. The matrix of the training data is denoted by $\boldsymbol{X} = [\boldsymbol{x}_1 \ldots \boldsymbol{x}_m]$. The terms $\mathcal{L}(h)$ and $\hat{\mathcal{L}}(h)$ denotes, respectively, the test and the training error.

## 2 Preliminaries

**Convolution: a simple example.** We start with a simple example of familiar convolutional neural networks, indulging some of the subtleties needed for our later discussions. For an RGB image input, each convolution filter will slide over the input image and act on its receptive field by simple multiplication. If we translate the pixels of the input image, ignoring the *corner* pixels, the output of the previous convolution operation would just get translated as well. The act of translation is an example of a *group action*, and the fact that this act shifts the convolution output is an example of equivariance property[1]. In what follows, we work with a generalized notion of action captured by group theory, which studies objects like translation and rotation that can combine and have an inverse element.

We introduce key concepts from group and representation theories necessary to present our main results in Appendix A. In the rest of this work, we will generally assume a *compact* group $G$. Note that this includes any *finite* group. For the most part, we will consider *Abelian* groups, i.e., commutative groups (such as planar rotations or periodic translations).

---

[1] We would like to emphasize again that the example is not perfect. First, the convolution in CNN is a cross-correlation. Second, the conventional CNNs are only approximately translation equivariant because of the finite input size and the corner pixels.

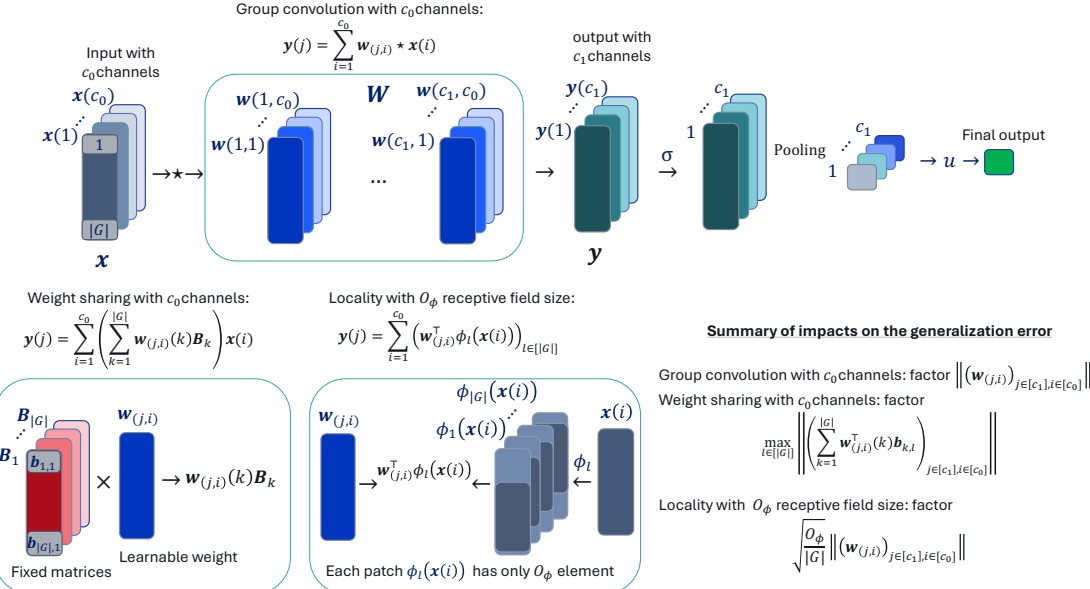

Figure 1: Visualization of the network architectures with equivariance, locality, and weight sharing. On the right, we also summarize how each choice impacts the generalization error in our theory.

**Equivariance.** Given two spaces $\mathcal{X}, \mathcal{Y}$ carrying an action of a group $G$, a function $\phi : \mathcal{X} \to \mathcal{Y}$ is said to be **equivariant** with respect to $G$ if $\phi(g.x) = g.\phi(x)$ for any $x \in \mathcal{X}$, i.e. if $\phi$ commutes with the group's action. For example, the spaces $\mathcal{X}, \mathcal{Y}$ could be vector spaces, and the group action $. : G \times \mathcal{X} \to \mathcal{X}$ could be a linear function.

**Group Convolution.** If $G$ is a *finite* group, the most popular design choice to construct equivariant networks relies on the **group convolution** operator Cohen and Welling (2016a), thereby generalizing typical convolutional neural networks (CNNs). Specifically, given an input signal $x : G \to \mathbb{R}$ over $G$ and a filter $w : G \to \mathbb{R}$, the group convolution produces another output signal $y : G \to \mathbb{R}$ defined as:

$$y(g) = (w \circledast_G x)(g) := \sum_{h \in G} w(g^{-1}h)x(h) \ . \tag{1}$$

Note that the signals $x, w, y$ can be represented as vectors $\boldsymbol{x}, \boldsymbol{w}, \boldsymbol{y} \in \mathbb{R}^{|G|}$; hence, the convolution operation can be expressed as $\boldsymbol{y} = \boldsymbol{W}\boldsymbol{x}$, where $\boldsymbol{W} \in \mathbb{R}^{|G| \times |G|}$ is a group-circulant matrix encoding $G$-convolution with the filter $\boldsymbol{w}$.

Like in classical CNNs, one typically considers multi-channel input signals $x : G \to \mathbb{R}^{c_0}$, filters $w : G \to \mathbb{R}^{c_1 \times c_0}$ and output signals $y : G \to \mathbb{R}^{c_1}$. We represent a multi-channel signal $x : G \to \mathbb{R}^{c_0}$ as a stack of features over the group $G$, namely, a tensor $(\boldsymbol{x}(1), \ldots, \boldsymbol{x}(c_0))$ of shape $|G| \times c_0$ (i.e. with $c_0$ channels and each $\boldsymbol{x}(i) \in \mathbb{R}^{|G|}$). Then, a convolution layer consists of $c_1$ convolutional filters $\{\boldsymbol{w}_j, j \in [c_1]\}$, each of size $|G| \times c_0$, parametrized by per-channel convolutions $\boldsymbol{w}_{(j,i)} \in \mathbb{R}^{|G|}, i \in [c_0]$; the convolution operation, which yields the output channel $j$, is given by

$$\boldsymbol{y}(j) = \sum_{i=1}^{c_0} \boldsymbol{w}_{(j,i)} \circledast_G \boldsymbol{x}(i) \in \mathbb{R}^{|G|}, j \in [c_1]$$

In summary, a group convolution layer can be visualized as

$$
\underbrace{\begin{bmatrix} \boldsymbol{W}_{(1,1)} & \boldsymbol{W}_{(2,1)} & \ldots & \boldsymbol{W}_{(c_0,1)} \\ \boldsymbol{W}_{(1,2)} & \boldsymbol{W}_{(2,2)} & \ldots & \boldsymbol{W}_{(c_0,2)} \\ \vdots & \vdots & \ddots & \vdots \\ \boldsymbol{W}_{(1,c_1)} & \boldsymbol{W}_{(2,c_1)} & \ldots & \boldsymbol{W}_{(c_0,c_1)} \end{bmatrix}}_{\boldsymbol{W}} \cdot \underbrace{\begin{bmatrix} \boldsymbol{x}(1) \in \mathbb{R}^{|G|} \\ \boldsymbol{x}(2) \in \mathbb{R}^{|G|} \\ \vdots \\ \boldsymbol{x}(c_0) \in \mathbb{R}^{|G|} \end{bmatrix}}_{\boldsymbol{x}} = \underbrace{\begin{bmatrix} \boldsymbol{y}(1) \in \mathbb{R}^{|G|} \\ \boldsymbol{y}(2) \in \mathbb{R}^{|G|} \\ \vdots \\ \boldsymbol{y}(c_1) \in \mathbb{R}^{|G|} \end{bmatrix}}_{\boldsymbol{y}} \tag{2}
$$

Here, any $\boldsymbol{W}_{(i,j)} \in \mathbb{R}^{|G| \times |G|}$ block is a group circulant matrix corresponding to $G$, which encodes a $G$-convolution parameterised by a filter $\boldsymbol{w}_{i,j} \in \mathbb{R}^{|G|}$. Note that the output of the convolution is a $|G| \times c_1$ signal, i.e. it contains a different value $y(g, j)$ for each group element $g \in G$ and output channel $j \in [c_1]$.

Note also that the group $G$ naturally acts on a signal $x : G \to \mathbb{R}^c$ as $g : x \mapsto g.x$, with $[g.x](h) = x(g^{-1}h)$. Then, one can show that the group convolution operator above is equivariant with respect to this action on its input and output, i.e., $w \circledast_G g.x = g.(w \circledast_G x)$. Additionally, it is well known, e.g., Cohen et al. (2018); Kondor and Trivedi (2018), that group convolutions with learnable filters in this form parameterize the most general linear equivariant maps between feature spaces of signals over a compact group $G$. Finally, GCNN typically uses an activation function applied point-wise on the convolution output. In our 1-layer architecture, a first convolution layer is followed by a pointwise activation layer and, then, a per-channel - average or max - pooling layer, which produces $c_1$ invariant features. These final features are mixed with the final linear layer $\boldsymbol{u}$. See Figure 1 for more details. The final network is then given as:

$$h_{\boldsymbol{u},\boldsymbol{w}}(\boldsymbol{x}) := \boldsymbol{u}^\top P \circ \sigma \begin{pmatrix} \sum_{i=1}^{c_0} \boldsymbol{w}_{(1,i)} \circledast_G \boldsymbol{x}(i) \\ \vdots \\ \sum_{i=1}^{c_0} \boldsymbol{w}_{(c_1,i)} \circledast_G \boldsymbol{x}(i) \end{pmatrix} \tag{3}$$

where $P(\cdot)$ is the pooling operation, and $\boldsymbol{w} := (\boldsymbol{w}_{(j,i)})_{j \in [c_1], i \in [c_0]}$ is the concatenation of all kernels.

## 3 Related Works

**Equivariant and Geometric Deep Learning** Previous works attempted to improve machine learning models by leveraging prior knowledge about the symmetries and the geometry of a problem Bronstein et al. (2021). A variety of design strategies have been explored in the literature to achieve group equivariance, for example via equivariant MLPs Shawe-Taylor (1993; 1989); Finzi et al. (2021), group convolution Cohen and Welling (2016a); Kondor and Trivedi (2018); Bekkers et al. (2018), Lie group convolution Bekkers (2020); Finzi et al. (2020), steerable convolution Cohen and Welling (2016b); Cohen et al. (2018); Worrall et al. (2017); Weiler et al. (2018b;a); Thomas et al. (2018); Weiler and Cesa (2019); Fuchs et al. (2020); Brandstetter et al. (2021); Cesa et al. (2022) and, very recently, by using geometric algebra Ruhe et al. (2023); Brehmer et al. (2023), to only mention a few. Other previous approaches include Mallat (2012); Dieleman et al. (2016); Defferrard et al. (2019). Some of these ideas have also been used to generalize convolution beyond groups to more generic manifolds via the framework of Gauge equivariance Cohen et al. (2019); Weiler et al. (2021). While equivariance is generally considered a powerful inductive bias that improves data efficiency, this large selection of equivariant designs raises the following question: What impact do different architectural choices have on performance, and to what extent do they aid generalization?

**Generalization Properties of Equivariant networks** Some previous works tried to answer some aspects of this question. Shawe-Taylor (1991) is one of the first works studying the relation between invariance and generalization. For example, Sokolić et al. (2017a) extend the robustness-based generalization bounds found in Sokolić et al. (2017b) by assuming the set of transformations of interest change the inputs drastically, thereby proving that the generalization error of a $G$-invariant classifier scales $1/\sqrt{|G|}$, where $|G|$ is the cardinality of the finite equivariance group $G$. Bietti and Mairal (2019) study the stability of some particular compact group equivariant models and also describe an associated Rademacher complexity and generalization bound. Lyle et al. (2020) investigate the effect of invariance under the lenses of the PAC-Bayesian framework but do not provide an explicit bound. Later, Elesedy and Zaidi (2021) use VC-dimension analysis and derive more concrete bounds. Elesedy (2022) studies compact-group equivariance in PAC learning framework and equates learning with equivariant hypotheses and learning on a reduced data space of orbit representatives to obtain a sample complexity bound. Sannai et al. (2021) propose a similar idea, proving that equivariant models work in a reduced space, the Quotient Feature Space (QFS), whose volume directly affects the generalization error. While the result relaxes the robustness assumption in Sokolić et al. (2017a), the final bound is suboptimal with respect to the sample size. PAC learnability under transformation invariance is also studied in Shao et al. (2022). Zhu et al. (2021) characterize the generalization benefit of invariant models using an argument based on the covering number induced by a set of transformations. More recently, Behboodi et al. (2022) leverages

a representation-theoretic construction of equivariant networks and provides a norm-based PAC-Bayesian generalization bound inspired by Neyshabur et al. (2018). Finally, Petrache and Trivedi (2023) considers a more general setting, allowing for approximate, partial, and misspecified equivariance and studying how the relation between data and model equivariance error impacts generalization.

**Generalization in generic neural networks**  Many works in the literature have previously investigated the generalization properties of deep learning methods. Zhang et al. (2017) first noted that a model trained on random labels can achieve small training errors while producing arbitrarily large generalization errors. This result raised a new challenge in the field since popular uniform complexity measures such as VC dimensions are inconsistent with this finding. This inspired many recent works which tried to explain generalization in terms of other quantities, such as margin or norms of the learnable weights; a few non-exhaustive examples are Wei and Ma (2019); Sokolić et al. (2017b); Neyshabur et al. (2018); Arora et al. (2018); Bartlett et al. (2017); Golowich et al. (2018); Dziugaite and Roy (2018a); Long and Sedghi (2019); Vardi et al. (2022); Ledent et al. (2021); Valle-Pérez and Louis (2020). The PAC Bayesian framework is a particularly popular method. It has been applied to neural networks in many previous works, e.g. see Neyshabur et al. (2018); Biggs and Guedj (2022); Dziugaite and Roy (2017; 2018b;a); Dziugaite et al. (2020); Lotfi et al. (2022). Jiang et al. (2020) perform a thorough experimental comparison of many complexity measures and identifies some failure cases; see also Nagarajan and Kolter (2019); Koehler et al. (2021); Negrea et al. (2021) for further discussion on uniform complexity measures and their limitations. The norm-based bounds can still be tight and informative for shallow networks considered in this work.

**Generalization bounds for Convolutional neural networks**  As one of the most popular deep learning architectures, convolutional neural networks (CNNs) have received particular attention in the literature. Generalization studies on CNNs are especially relevant for our work since we focus significantly on equivariance to finite and abelian groups, which can often be realized via periodic convolution. Pitas et al. (2019); Long and Sedghi (2019); Vardi et al. (2022); Ledent et al. (2021) previously studied these architectures. The authors in Vardi et al. (2022) represented a convolutional layer as a linear layer applied on local patches and used Rademacher complexity to get the bound. In Long and Sedghi (2019), the bound is derived using Vapnik-Chervonenkis analysis Vapnik and Chervonenkis (2015); Giné and Guillou (2001) and depends on the number of parameters but independent of the number of pixels in the input. In Graf et al. (2022), the authors use two covering-numbers-based bounds for Rademacher complexity analysis of convolutional models. Please see Appendix J for further discussions.

## 4 Sample Complexity Bounds For Equivariant Networks

We consider the group convolutional networks as defined in eq. 3. We assume that the input to the network is bounded as $\|\boldsymbol{x}\| \leq b_x$[2]. Consider the following hypothesis space:

$$\mathcal{H} := \{h_{\boldsymbol{u},\boldsymbol{w}} : \|\boldsymbol{u}\| \leq M_1, \|\boldsymbol{w}\| \leq M_2\}, \tag{4}$$

where $\boldsymbol{u} \in \mathbb{R}^{c_1}, \boldsymbol{w} \in \mathbb{R}^{c_0 c_1 |G|}$. The hypothesis space is the group convolution network class that has bounded Euclidean norm on the kernels. For this network, we derive dimension-free bounds.

Since the pooling operation is a permutation invariant operation, we can use Theorem 7 in Zaheer et al. (2017) to show that one can always find two functions $\phi : \mathbb{R}^{|G|+1} \to \mathbb{R}$ and $\rho : \mathbb{R} \to \mathbb{R}^{|G|+1}$ such that $P \circ \boldsymbol{z} = \phi\left(\frac{1}{|G|} \sum_{i=1}^{|G|} \rho(z_i)\right)$. We can provide the following generalization bound for a subset of such representations, namely for positively homogeneous $\phi : \mathbb{R} \to \mathbb{R}$ and $\rho : \mathbb{R} \to \mathbb{R}$.

**Theorem 4.1.** *Consider the hypothesis space $\mathcal{H}$ defined in eq. 4. If $P(\cdot)$ is the pooling operation represented as $P \circ \boldsymbol{z} = \phi(\frac{1}{|G|} \mathbf{1}^\top \rho(\boldsymbol{z}))$, where the two functions $\rho(\cdot), \phi(\cdot)$ and the activation function $\sigma(\cdot)$ are all 1-Lipschitz positively homogeneous activation function, then with probability at least $1 - \delta$ and for all $h \in \mathcal{H}$, we have:*

$$\mathcal{L}(h) \leq \hat{\mathcal{L}}(h) + 2\frac{b_x M_1 M_2}{\sqrt{m}} + 4\sqrt{\frac{2\log(4/\delta)}{m}}.$$

---

[2]For all the results in the paper, we can use a data-dependent bound on the input. Namely, it suffices to assume that $\max_{i \in [m]} \|\boldsymbol{x}_i\| \leq b_x$.

The proof leverages Rademacher complexity analysis and is presented in Appendix C.5. Note that the assumption of positively homogeneity is only needed to utilize the peeling technique of Golowich et al. (2018). We expect that similar techniques from Vardi et al. (2022) can be used to extend the result to Lipschitz activation functions.

**Average pooling.** Consider now the special case of average pooling operation $P(\boldsymbol{x}) = \frac{1}{|G|} \mathbf{1}^{\top} \boldsymbol{x}$. This is a linear layer; therefore, it can be combined with the last layer $\boldsymbol{u}$ to yield a standard two-layer neural network. We can utilize the results from Vardi et al. (2022); Golowich et al. (2018). The combination of the layer $\boldsymbol{u}$ and average pooling is a linear layer with the norm $M_1/\sqrt{|G|}$, and the matrix $\boldsymbol{W}$, a circulant matrix, has the Frobenius norm $\sqrt{|G|}M_2$. The product of these norms would be bounded by $M_1 M_2$. Being a special case of the model in Vardi et al. (2022), we can use their results off-the-shelf to get an upper bound on the sample complexity that depends only on $M_1 M_2$ for Lipschitz-activation functions (Theorem 2 of Vardi et al. (2022)). We provide an independent proof for positively homogeneous activation functions in Appendix C.2 with a clean form. Note that the authors in Vardi et al. (2022) provide a result for average pooling that contains the term $O_\phi$, the maximal number of patches that any single input coordinate appears. In our case, this term equals $|G|$, which is canceled out by the average pooling term. Their bound depends on the spectral norm of the underlying circulant matrix, which can be bigger than the norm of the filter provided here. But, their proof can be reworked with the norm of convolutional filter instead, in which case, their result will be our special case.

**Impact of group size.** The above theorem is dimension-free, and there is no dependence on the number of input and output channels $c_0$ and $c_1$, as well as the group size $|G|$. Note that for average pooling and sufficiently smooth activation functions, we can use even the stronger result (Theorem 4 of Vardi et al. (2022)) and get a bound that depends on $M_1 M_{2 \to 2}/\sqrt{|G|}$, where $M_{2 \to 2}$ is the spectral norm of $\boldsymbol{W}$. This manifests the impact of group size on the generalization similar to Sokolić et al. (2017a); Behboodi et al. (2022).

**Max pooling.** We cannot rely on the previous results for the max pooling operation since the peeling argument would not work. Theorem 7 and 8 of Vardi et al. (2022) provide a bound for max pooling, however their network does not contain the linear aggregation $\boldsymbol{u}$ after max-pooling, and their bound contains dimension dependencies such as $\log(|G|c_1)$ or $\log(m)$. We provide various bounds for max pooling in Appendix C.3 and C.4. The proof technique is different from the above results. Again, the bounds have different dimension dependencies. We believe these dependencies are proof artifacts. Removing them would be an interesting direction for future work.

To summarize, we have established that for a single hidden layer group convolution network, we can have a dimension-free bound that depends merely on the norm of filters.

*Remark* 4.2 (Frequency Domain Analysis). The authors in Behboodi et al. (2022) improved the PAC-Bayesian generalization bound by conducting their analysis using the representation in the frequency domain. In our Rademacher analysis, such a shift would not bring any additional gain, and we recover the same bound. We provide the details in the supplementary materials.

## 4.1 Bounds for Multi-Layer Equivariant Network

In this section, we study a simple extension to multi-layer group equivariant networks. Consider the following network with $L$ hidden layer:

$$h_{\boldsymbol{u}, \{\boldsymbol{w}^{(l)}, l \in [L]\}}(\boldsymbol{x}) := \boldsymbol{u}^{\top} P \circ \sigma(\boldsymbol{W}^{(L)} \sigma(\boldsymbol{W}^{(L-1)} \ldots \sigma(\boldsymbol{W}^{(1)}) \boldsymbol{x} \ldots) \tag{5}$$

where $P(\cdot)$ is the average pooling operation, $\sigma(\cdot)$ is the ReLU function. Each linear layer $\boldsymbol{W}^l$ is the same as the mapping in eq. 2 but with $c_{l-1}$ and $c_l$ as the input and output channels. The new hypothesis space is given by:

$$\mathcal{H}^{(L)} := \left\{ h_{\boldsymbol{u}, \{\boldsymbol{w}^{(l)}, l \in [L]\}} : \|\boldsymbol{u}\| \leq M_1, \|\boldsymbol{w}_i\| \leq M_{i+1}, i \in [L] \right\}. \tag{6}$$

We have the following theorem.

**Theorem 4.3.** *Consider the hypothesis space $\mathcal{H}^{(L)}$ defined in eq. 6. With probability at least $1 - \delta$ and for all $h \in \mathcal{H}$, we have:*

$$\mathcal{L}(h) \leq \hat{\mathcal{L}}(h) + 2\frac{b_x|G|^{\frac{L-1}{2}}M_1 M_2 \ldots M_{L+1}}{\sqrt{m}} + 4\sqrt{\frac{2\log(4/\delta)}{m}}.$$

The proof can be found in Appendix C.6. We first comment on the dimension dependency of the bound. Although our bound has no dependence on the number of input and hidden layer channels, it still has dependence on $|G|^{(L-1)/2}$. It is not entirely dimension-free. Using some other techniques, for example, see Section 3 in Golowich et al. (2018), it can be improved to $(L-1)\log|G|$ for $L > 1$. However, dimension dependence is only one concern of the above result. It is worth pointing out that the generalization bounds for deeper networks suffer from many shortcomings and generally fail to correlate well with the empirical generalization error. For example, various norm bounds were considered in Jiang et al. (2020), and the correlation with the generalization error was explored. Generally, the norm-based bounds performed poorly, while sharpness-aware bounds showed promises. Besides, in Nagarajan and Kolter (2019), the norm-based bounds were shown to increase drastically with the training set size, leading to looser bounds with increasing training set size. Therefore, we think these norm-based bounds have limitations in deep network generalization analysis.

## 4.2 Lower bound on the Rademacher Complexity Analysis

A natural question that might come up is whether the obtained bound is tight or not. In this section, we provide an answer to this by showing that the Rademacher complexity is lower bounded similarly for a class of networks.

**Theorem 4.4.** *Consider the hypothesis space $\mathcal{H}$ defined in eq. 4. If $P(\cdot)$ is the average pooling operation, and $\sigma(\cdot)$ is the ReLU activation function, then there is a data distribution such that the Rademacher complexity is lower bounded by $c\frac{b_x M_1 M_2}{\sqrt{m}}$ where $c$ is an independent constant.*

The proof is provided in the Appendix E. Note that Rademacher complexity (RC) bounds are known to be tight for shallow models, such as SVMs. Indeed, if one fixes the first layer and only trains the last linear layer with weight decay, the model is equivalent to hard-margin SVM, for which RC bounds are tight. We also provide additional evidence in the numerical result section.

## 5 General Equivariant Networks

The focus of our analysis in the above section has been on group convolutional networks and related architectures. In particular, we had considered the equivariance for finite Abelian groups for which the filters were implemented using multi-channel convolution operation. We now look at the general equivariant networks w.r.t. a compact group. As we will see, the analysis for these networks would be based on their MLP structure.

We follow the procedure described in Cohen and Welling (2016b); Cesa et al. (2022); Weiler et al. (2018b). We assume a general input space where the action of the compact group $G$ on the space is given by a linear map $\rho_0(g)$ for $g \in G$. For this space, the filters of the first layer, similarly to GCNs, are parametrized in the frequency domain. The generalization of Fourier analysis to compact groups is done via the notion of *irreducible representations* (irreps). An irrep, typically denoted by $\psi$, can be thought as a frequency component of a signal over the group.

Using tools from the representation theory, the map $\rho_0(g)$ can be represented in the frequency domain as the direct sum of irreps $\boldsymbol{Q}_0^\top \left( \bigoplus_\psi \bigoplus_{i=1}^{m_{0,\psi}} \psi \right) \boldsymbol{Q}_0$ where $m_{0,\psi}$ is the multiplicity of the irrep $\psi$. The matrix $\boldsymbol{Q}_0$ is a unitary matrix representing the generalized Fourier transform given by $\hat{\boldsymbol{x}} = \boldsymbol{Q}_0 \boldsymbol{x}$. See Appendix A.1 for detailed definitions. A similar frequency domain representation can be obtained for the hidden layer in terms of irreps, each with multiplicity $m_{1,\psi}$. The block diagonal structure of filters in the multi-channel GCNs given in eq. 33 is now obtained by a general block-diagonal structure. The equivariant neural network is

represented as

$$h_{\hat{\boldsymbol{u}},\hat{\boldsymbol{W}}} := \hat{\boldsymbol{u}}^\top \boldsymbol{Q}_2 \sigma \left( \boldsymbol{Q}_1 \bigoplus_\psi \bigoplus_{i=1}^{m_{1,\psi}} \sum_{j=1}^{m_{0,\psi}} \hat{\boldsymbol{W}}(\psi,i,j)\hat{\boldsymbol{x}}(\psi,j) \right).$$

Since we are working with the point-wise non-linearity $\sigma$ in the *spatial domain*, two unitary transformations $\boldsymbol{Q}_1$ and $\boldsymbol{Q}_2$ are applied as Fourier transforms from the frequency domain to the spatial domain. The last layer $\hat{\boldsymbol{u}}$ should be chosen to yield a group invariant function, which means that the vector $\hat{\boldsymbol{u}}$ only aggregates the frequencies of the trivial representation $\psi_0$, and it is zero otherwise. To use an analogy with group convolutional networks, $\boldsymbol{Q}_1$ and $\boldsymbol{Q}_2$ are the Fourier matrices, and $\hat{\boldsymbol{u}}$ is a combination of the pooling, which projects into the trivial representation of the group, and the last aggregation step. The general equivariant networks are defined in Fourier space as follows:

$$\mathcal{H}_{\hat{\boldsymbol{u}},\hat{\boldsymbol{W}}} := \left\{ h_{\hat{\boldsymbol{u}},\hat{\boldsymbol{W}}} : \|\hat{\boldsymbol{u}}\| \le M_1, \left\|\hat{\boldsymbol{W}}\right\| \le M_2 \right\}.$$

Note that the hypothesis space assumes a bounded norm of the filters' parameters in the frequency domain. For this hypothesis space, we can get a similar dimension-free bound.

**Theorem 5.1.** *Consider the hypothesis space $\mathcal{H}_{\hat{\boldsymbol{u}},\hat{\boldsymbol{W}}}$ of equivariant networks with bounded weight norms. If the activation function $\sigma(\cdot)$ is 1-Lipschitz positively homogeneous, then with probability at least $1 - \delta$ for all $h \in \mathcal{H}_{\hat{\boldsymbol{u}},\hat{\boldsymbol{W}}}$, we have:*

$$\mathcal{L}(h) \le \hat{\mathcal{L}}(h) + 2\frac{b_x M_1 M_2}{\sqrt{m}} + 4\sqrt{\frac{2\log(4/\delta)}{m}}.$$

The proof is presented in Appendix D. This result provides another dimension-free bounds for equivariant networks that depend merely on the norm of the kernels. Note that the network has a standard MLP structure, so the result can be obtained using techniques used in Golowich et al. (2018). Also, note that constructing equivariant networks in the frequency domain is useful in steerable networks to deal with continuous rotation groups Poulenard and Guibas (2021); Cesa et al. (2022), while the parametrization in the spatial domain is pursued in works like Bekkers (2020); Finzi et al. (2020).

## 6 Generalization Bounds for Weight Sharing

In this section, we try to answer the question whether the gain of our bound lies in the specific equivariant architecture or weight sharing. The answer is subtle. Indeed, an arbitrary type of weight sharing will not bring out the generalization gain. However, weight-sharing schemes can lead to a similar gain without necessarily being equivariant.

For a fair comparison with the group convolution network explained above, we consider an architecture with the same number of effective parameters shared similarly. The weight-sharing network is specified as follows:

$$h_{\boldsymbol{u},\boldsymbol{w}}^{w.s.}(\boldsymbol{x}) := \boldsymbol{u}^\top P \circ \sigma \begin{pmatrix} \sum_{c=1}^{c_0} \left( \sum_{k=1}^{|G|} \boldsymbol{w}_{(1,c)}(k)\boldsymbol{B}_k \right) \boldsymbol{x}(c) \\ \vdots \\ \sum_{c=1}^{c_0} \left( \sum_{k=1}^{|G|} \boldsymbol{w}_{(c_1,c)}(k)\boldsymbol{B}_k \right) \boldsymbol{x}(c) \end{pmatrix}, \tag{7}$$

where $\boldsymbol{B}_k$'s are fixed $|G| \times |G|$ matrices inducing the weight sharing scheme. These matrices are not trained using data and merely specify how the weights are shared in the network. For example, if $\boldsymbol{B}_k$'s are chosen as the basis for the space of circulant matrices, the setup boils down to the group convolution network.

The corresponding hypothesis space is defined as:

$$\mathcal{H}^{w.s.} = \left\{ h_{\boldsymbol{u},\boldsymbol{w}}^{w.s.}(\boldsymbol{x}) : \|\boldsymbol{u}\| \le M_1, \|\boldsymbol{w}\|_{\boldsymbol{B}} \le M_2^{w.s.} \right\}$$

where

$$\|\boldsymbol{w}\|_{\boldsymbol{B}} := \max_{l \in [|G|]} \left\| \left( \sum_{k=1}^{|G|} \boldsymbol{w}_{(j,c)}(k)\boldsymbol{b}_{k,l}^\top \right)_{c \in [c_0], j \in [c_1]} \right\|_F.$$

The following proposition provides a purely norm-based generalization bound.

**Proposition 6.1.** For the class of functions $h$ in the hypothesis space $\mathcal{H}^{w.s.}$ with the average pooling operation $P$, $\sigma(\cdot)$ as a 1-Lipschitz positively homogeneous activation function, the generalization error is bounded as:

$$\mathcal{L}(h) \leq \hat{\mathcal{L}}(h) + 2\frac{b_x M_1 M_2^{w.s.}}{\sqrt{m}} + 4\sqrt{\frac{2\log(4/\delta)}{m}}.$$

Note that if $\boldsymbol{b}_{k,l}$'s are the rows of the circulant matrices, then we end up with the same result as before with $M_2^{w.s.} = M_2$. Interestingly, if the vectors $\{\boldsymbol{b}_{k,l} : k \in [|G|]\}$'s are orthogonal to each other for each $l \in [|G|]$, and all of them have unit norm, i.e., $\|\boldsymbol{b}_{k,l}\| = 1$, then we also get $M_2^{w.s.} = M_2$. The conclusion is that the weight sharing can impact the generalization similarly to the group convolution, even if the construction does not arise from the group convolution. This does not generally hold, particularly if the row vectors of matrices $\boldsymbol{B}_k$ are not orthogonal. Note that throughout the paper, we have not assumed anything regarding the underlying distribution, which can be without symmetry. On the other hand, the gain of equivariance shows itself for distributions with built-in symmetries if we assume that the underlying task has invariance or equivariance property.

# 7 Generalization Bounds for Local Filters

In previous sections, we observed that the impact of equivariance on the generalization is very similar to that of weight sharing under an appropriately chosen basis. Another argument relates the generalization benefits of convolutional neural networks to the use of local filters, where the same filter is applied to a number of patches. The patches have lower dimensions, and the filters similarly have lower dimensions than the total input dimension. Note that this goes beyond the benefit of weight sharing mentioned in the previous section. One example is Theorem 6 in Vardi et al. (2022); we follow their setup. In their case, each convolution operation $\boldsymbol{w}_{(j,k)} \circledast_G \boldsymbol{x}(k)$ can be represented as $\left(\boldsymbol{w}_{(j,k)}^\top \phi_1(\boldsymbol{x}(k)) \ldots \boldsymbol{w}_{(j,k)}^\top \phi_{|G|}(\boldsymbol{x}(k))\right)$, where $\phi_l(\cdot)$ represents the patch $l$. Each patch selects a subset of input entries. The patches $\Phi = \{\phi_l, l \in [|G|]\}$ are assumed to be *local*, in the sense that each coordinate in $\boldsymbol{x}(k)$ appears at most in $O_\Phi$ number of patches with $O_\Phi < |G|$. More formally, for any vector $\boldsymbol{x}$ and a set $S \subset [|G|]$, define the vector $\boldsymbol{x}_S := (x_i)_{i \in S}$, i.e., with entries selected from the index in $S$. Define the patch $\phi_l : \mathbb{R}^{|G|} \to \mathbb{R}^{n'}$ as $\phi_l(\boldsymbol{x}) := \boldsymbol{x}_{S_l}$ for a subset $S_l$. The group convolution network with locality is defined as:

$$h_{\boldsymbol{u},\boldsymbol{w}}^\Phi(\boldsymbol{x}) = \boldsymbol{u}^\top P \circ \sigma \begin{pmatrix} \left(\sum_{k=1}^{c_0} \left(\boldsymbol{w}_{(1,k)}^\top \phi_l(\boldsymbol{x}(k))\right)\right)_{l \in [|G|]} \\ \vdots \\ \left(\sum_{k=1}^{c_0} \left(\boldsymbol{w}_{(c_1,k)}^\top \phi_l(\boldsymbol{x}(k))\right)\right)_{l \in [|G|]} \end{pmatrix}.$$

By reorganizing the weights in each row corresponding to each patch, the whole network is presented as $\boldsymbol{u}^\top P \circ \sigma(\boldsymbol{W}\boldsymbol{x})$, where we assume that the matrix $\boldsymbol{W}$ conforms to the set of patches $\Phi$. The hypothesis space with locality is defined as

$$\mathcal{H}^\Phi = \{h_{\boldsymbol{u},\boldsymbol{w}}^\Phi(\boldsymbol{x}) : \|\boldsymbol{u}\| \leq M_1, \|\boldsymbol{w}\| \leq M_2\}.$$

For this class of functions, it can be shown that there is a generalization benefit in using local filters besides the gain of equivariance or proper weight sharing. This conclusion is captured in the following result.

**Proposition 7.1.** Consider the hypothesis space of functions $\mathcal{H}^\Phi$ constructed by patches $\Phi$, the average pooling operation $P$, 1-Lipschitz positively homogeneous activation function $\sigma$. Let $O_\Phi$ be the maximal number of patches that any input entry appears in. Then, with probability at least $1 - \delta$ and for all $h \in \mathcal{H}^\Phi$, we have:

$$\mathcal{L}(h) \leq \hat{\mathcal{L}}(h) + 2\sqrt{\frac{O_\Phi}{|G|}} \frac{b_x M_1 M_2}{\sqrt{m}} + 4\sqrt{\frac{2\log(4/\delta)}{m}}.$$

Compared with the result above, the locality assumption brings an additional improvement of $\frac{O_\Phi}{|G|}$. Although the proof focuses on average pooling and group convolution, the same results can be obtained for weight sharing. We should only change the filter parametrization from the matrix $\boldsymbol{B}_k$ basis to the vector basis $\boldsymbol{b}_k$ given as $\left(\sum_{k=1}^{|G|} \boldsymbol{w}_{(j,c)}(k)\boldsymbol{b}_k\right)$.

The architecture considered in Vardi et al. (2022) is a single-channel version of our result (see more discussions in Appendix J). In that case, the difference in their bounds is in using the spectral norm of $\boldsymbol{W}$. However, their proof only needs a bound on the Euclidean norm of weights, which would end in the same result as ours.

The analysis so far has focused on the local filters in the spatial domain. However, the filters are also applied per frequency in the frequency domain and are, therefore, local. This can be seen from the block diagonal structure of filters $(\bigoplus_\psi \hat{w}(\psi))$. In this sense, the filters in the frequency domain are already local. When the filters are band-limited in the frequency domain, we have an additional notion of locality. In practice, this is useful when the input is also band-limited; therefore, not all the frequencies are useful for learning. This introduces a fundamental trade-off for locality benefits arising from the uncertainty principle Donoho and Stark (1989); Folland and Sitaram (1997). According to the uncertainty principle, the band-limited filters with $B$ non-zero components will have at least $|G|/B$ non-zero entries in the spatial domain (see Section I for more details). The trade-off is captured in the following proposition.

**Proposition 7.2.** Consider the hypothesis space of group convolution networks with band-limited filters. If the filters have $B$ non-zero entries in the frequency domain, then the generalization error term in Proposition 7.1 for the smallest spatial filters is at least of order $\mathcal{O}(b_x M_1 M_2 / \sqrt{mB})$.

As it can be seen, the assumption of band-limitedness can potentially bring the gain of order $1/B$ if the filters are spatially local. To summarize, there is a gain in using local filters. However, since band-limited filters are more efficient for band-limited signals, the uncertainty principle imposes a minimum spatial size for band-limited filters.

## 8 Numerical Results

Let us first introduce the two family of groups we consider in our experiments: the *cyclic group* $C_n$ containing $n$ discrete rotations and the *dihedral group* $D_n$ of $n$ discrete rotations and $n$ reflections (see Def. A.2 and Def. A.3 for precise definitions of these groups).

In this section, we validate our theoretical results on a variation of the rotated MNIST and CIFAR10 datasets where we consider a simpler binary classification task (determining whether a digit is smaller or larger than 4 in MNIST and whether an image belongs to the first 5 classes of CIFAR10 or not). Rather than working with raw images, we pre-process the dataset by linearly projecting each image via a fixed randomly initialized $G = D_{32}$-steerable convolution layer Weiler and Cesa (2019) to a single 6400-dimensional feature vector[3] - interpreted as a $c_0 = 100$-channels signal over $G = D_{32}$; see Appendix K for more details and Fig. 5 for a visualization of this linear projection. When constructing equivariance to a subgroup $G < D_{32}$, this feature vector is interpreted as a $c_0 = 100 \cdot \frac{|D_{32}|}{|G|}$-channels signal over $G$. We design our equivariant networks as in Eq. 3 and use $c_1 \cdot |G| \approx 2000$ total intermediate channels for all groups $G$ in the MNIST experiments and $c_1 \cdot |G| \approx 4000$ in the CIFAR10 ones. The dataset is then augmented with random $D_{32}$ transformations to make it symmetric. Hence, we expect that $G$-equivariant models with $G$ closer to $D_{32}$ to perform best.

In these experiments, we are interested in how well our theoretical bound correlates with the observed trends. In particular, we focus on the $\frac{M_1 M_2}{\sqrt{m}}$ term which is the main component in our bounds. In Fig 2, we evaluate the networks on different values of training set size $m$ and on different equivariance groups $G$. We first observe that the $\frac{M_1 M_2}{\sqrt{m}}$ term correlates well with the empirical generalization error on both the MNIST dataset in Fig. 2a and the CIFAR10 dataset in Fig. 2c. In particular, larger groups $G$ achieve both lower generalization error and lower value of our norm bound. We also note that this term decreases as $1/\sqrt{m}$ in Fig. 2b, which

---

[3] This is a random linear transformation of the input images into feature vectors: the reader can think of this as analogous to reshaping an image before feeding it into an MLP, but with some care to preserve rotation and reflection equivariance. Note also that, since the output (6400) is much larger than the number of input pixels, this transformation is practically invertible and preserves all relevant input information.

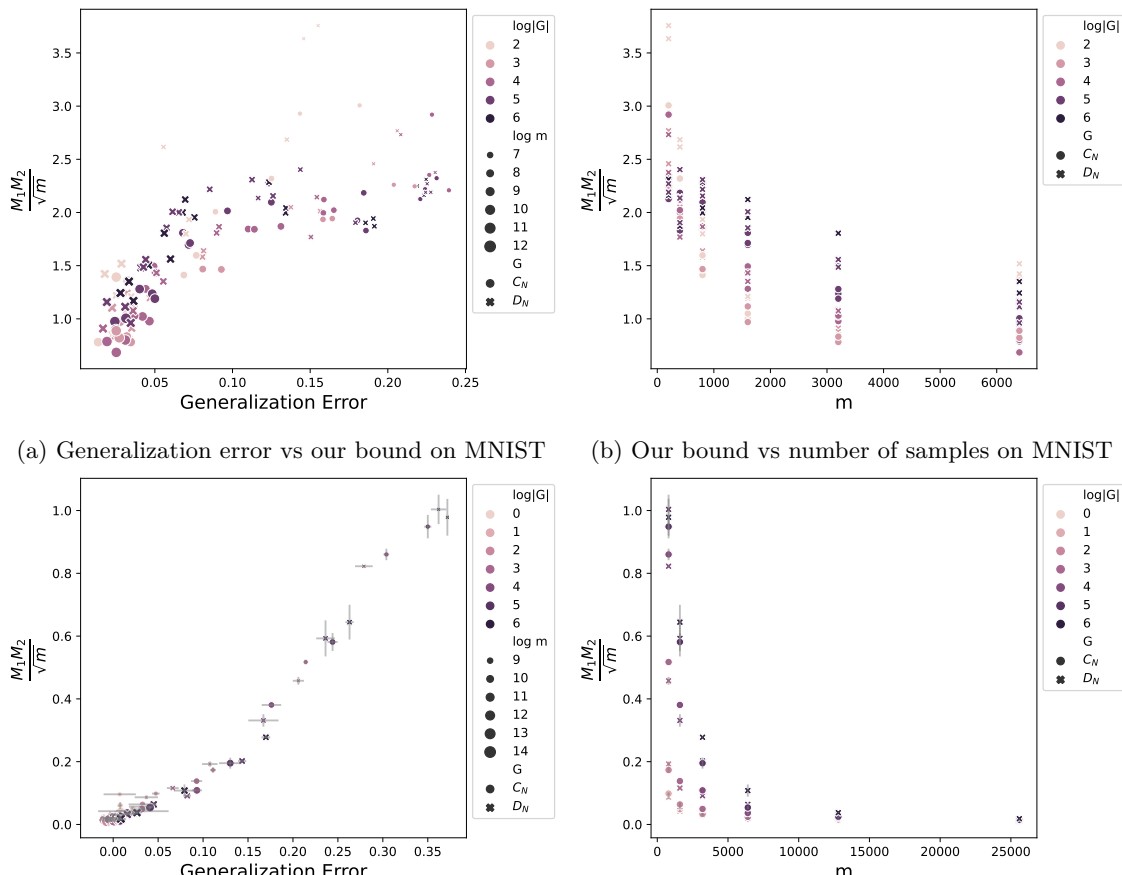

(a) Generalization error vs our bound on MNIST
(b) Our bound vs number of samples on MNIST

(c) Generalization error vs our bound on CIFAR10
(d) Our bound vs number of samples on CIFAR10

Figure 2: Numerical results for the generalization error on the rotated MNIST and CIFAR10 datasets. The plots on the left, (a)-(c), confirm that our theoretical bound captures the effect of different configurations - equivariance groups $G$, training set sizes $m$ and datasets - on the generalization error, i.e. there is a positive correlation between the generalization error and our theoretical bound $\frac{M_1 M_2}{\sqrt{m}}$ across all these cases. On the right, (b)-(d), we verify the bound decreases following a trend similar to $\frac{1}{\sqrt{m}}$ and approaching zero for large training set sizes $m$.

indicates that our bound becomes non-vacuous as $m$ increases, and we do not suffer from deficiencies of other norm based bounds Nagarajan and Kolter (2019).

Overall, *these results suggest that our bound can explain generalization across different choices of equivariance group $G$, as well as different training settings.*

Next, we investigate how other equivariant design choices affect generalization and verify if our bound can model this effect. Precisely, when parameterizing the filters over the group $G$ in the frequency (i.e. *irreps*) domain (as in Sec 5), it is common to only use a smaller subset of all frequencies. A bandlimited sub-set of frequencies can be interpreted as a form of locality in the frequency domain, in analogy to the local filters in Sec 7. In Fig.3, we experiments on CIFAR10 with different variations of our equivariant architecture obtained by varying the maximum frequency used to parameterize filters. In these experiments, we keep the training dataset size fixed to $m = 3200$ and we only consider the largest groups (since a wider range of frequencies in $[0, \ldots, N/2]$ can be chosen).

In Fig.3, we find that *our bound captures the effect of locality in the frequency domain on the generalization.*

Figure 3: Our bound vs. the generalization error on CIFAR10 (training set size $m = 3200$) when varying the maximum frequency used to parameterize the filters, which is the locality effect in the frequency domain. Each dot and its error bars represent the mean and standard deviation over at least 3 runs with the same configuration. As expected, *architectures leveraging a lower frequency design achieve lower generalization error* and our bound can exactly capture this effect. Note that increased frequencies are still beneficial for the final test performance: we study the trade-off between generalization and test performance as a function of the model frequency in Fig. 4.

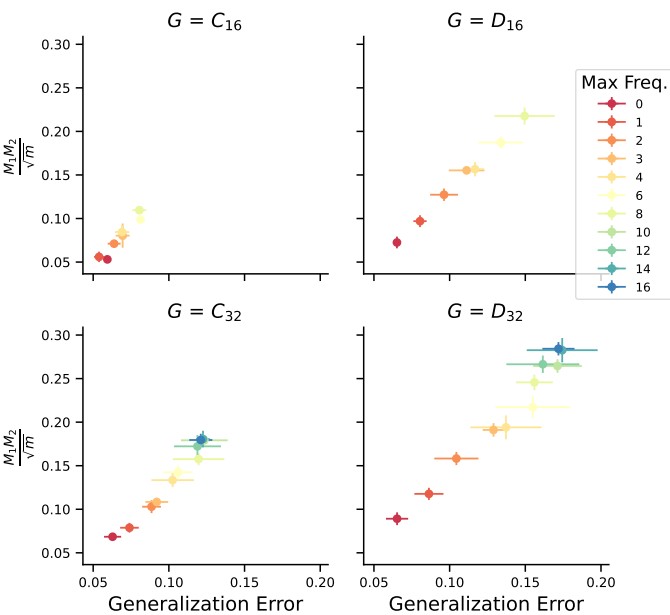

In Fig. 3, we observed that higher frequencies typically lead to worse generalization. For this reason, we include a final study to explore the effect of locality in the frequency domain - i.e., band-limitation - on the final test performance of the models and the associated trade-offs between generalization and expressivity. Fig. 4 compares the generalization error with the final test accuracy of some of our models when varying the maximum frequency of the filters and the training set size $m$. This visualization highlights a particular behavior: low-frequency models tend to achieve lower generalization error because of worse data fitting (lower test accuracy). Conversely, higher frequency models often show higher generalization errors (especially in the low data regime) but achieve much higher test performance. Moreover, high-frequency models benefit the most from increased dataset size $m$. These properties lead to the (multiple) V-shaped patterns in Fig. 4: on the smaller datasets (light dots in Fig. 4), while increasing frequencies leads to improved test accuracy, it also increases the generalization error. On the other hand, in the larger dataset (dark dots in Fig. 4), higher frequencies directly improve the test performance and even show minor improvements in generalization error. When varying dataset size, these different behaviors form multiple V-shaped patterns in Fig. 4 (highlighted with the dotted lines).

Figure 4: Test accuracy vs. Generalization error on CIFAR10 when varying the *maximum frequency* used to parameterize the filters and the training set size $m$. Each dot is the average performance over at least 3 runs with the same configuration. In Fig. 3, we found that higher frequencies correlated with higher generalization error: here, we study the trade-off between generalization and test performance. For each dataset size $m$, increasing the frequency improves the test performance until a certain saturation point; beyond that, increased frequencies mostly lead to increased generalization error. We highlight this effect by drawing four dotted lines following the trends for varying frequencies with $G = C_{32}$ on four different dataset sizes $m$: the different slopes of the curves correspond to the different saturation effects.

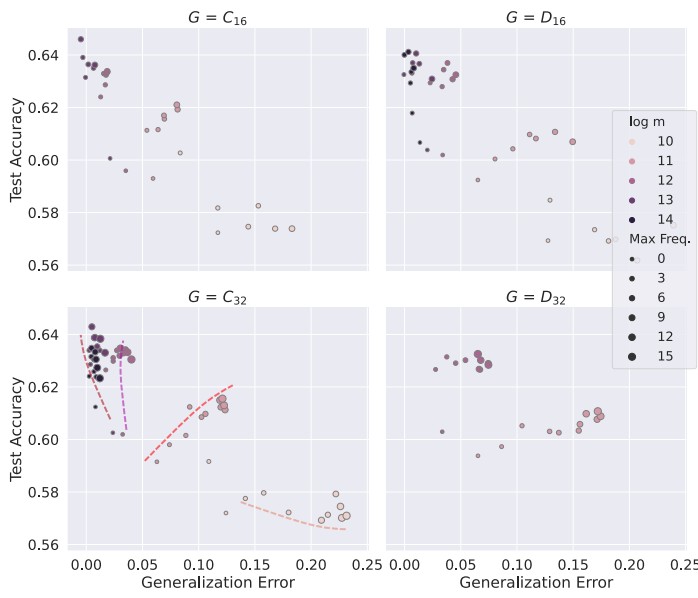

## 8.1 Main Insights from Numerical Results

We would like to summarize some of the main insights from the numerical examples, some of which pose interesting theoretical questions.

The generalization error seems to be lowest at the medium group sizes as seen in MNIST or CIFAR experiments, although the trend is unclear. There seems to be a group size that leads to the lowest generalization error, and if this conjecture is correct is an open question.

The other observation is about the relation between test accuracy and the generalization error. In general, it can be seen that the highest test accuracies have lower generalization errors in all the plots in Figure 4. It is interesting to directly explore the impact of the group size on the test error, which requires a different machinery.

# 9 Conclusion

In this work, we provided various generalization bounds on equivariant networks with a single hidden layer as well as a bound for multi-layer neural networks. We have used Rademacher complexity analysis to derive these bounds, where the proofs were based on either direct analysis of the Rademacher complexity or covering number arguments. The bounds are mostly dimension-free, as they do not depend on the input and output dimensions. They only depend on the norm of learnable weights. The situation changes for multi-layer scenarios and max-pooling where some dimension dependency appears in the bound. In light of the results on the limitation of uniform complexity bounds for deeper neural networks Nagarajan and Kolter (2019); Jiang et al. (2020), we believe that the bound on multi-layer scenario would suffer from similar limitations. However, the emergence of some dimensions in the bound for max-pooling seems related to how covering number argument was applied. Whether the dependency can be removed for max-pooling is an interesting research direction.

We have considered equivariant models in spatial and frequency domains, models with weight sharing, and local filters. The first insight from our analysis is that suitable weight-sharing techniques should be able to provide similar guarantees. This is not surprising, as we did not assume any symmetry in the data distribution. Other works in the literature have highlighted the benefit of equivariance if such symmetries exist - see, for example, Sannai et al. (2021); Sokolić et al. (2017a). Finally, local filters can potentially provide an additional gain, although the story for band-limited filters is more subtle. We also provide a lower bound on Rademacher complexity. We have conducted extensive numerical experiments and investigated the correlation between our generalization error and the true error, as well as the relation between the number of samples, group size, and frequency.

We have focused on positively homogeneous activation functions. We expect that the result can be extended to general Lipschitz activations using techniques in Vardi et al. (2022). However, the current proof techniques would not work for norm-based nonlinearities used in equivariant literature Worrall et al. (2017); Weiler et al. (2018a); Weiler and Cesa (2019). We encourage readers to consult Appendix J for comparison with other works and future directions.

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

# A  An Overview of Representation Theory and Equivariant Networks

We provide a brief overview of some useful concepts from (compact group) Representation Theory in this supplementary section.

**Definition A.1** (Group). A *group* a set $G$ of elements together with a binary operation $\cdot : G \times G \to G$ satisfying the following three group axioms:

- Associativity:    $\forall a, b, c \in G \quad a \cdot (b \cdot c) = (a \cdot b) \cdot c$

- Identity:    $\exists e \in G : \forall g \in G \quad g \cdot e = e \cdot g = g$

- Inverse:    $\forall g \in G \ \exists g^{-1} \in G : \quad g \cdot g^{-1} = g^{-1} \cdot g = e$

The inverse elements $g^{-1}$ of an element $g$, and the identity element $e$ are *unique*. Moreover, if the binary operation $\cdot$ is also *commutative*, the group $G$ is called *abelian group*. To simplify the notation, we commonly write $ab$ instead of $a \cdot b$.

The **order** of a group $G$ is the *cardinality* of its set and is indicated by $|G|$. A group $G$ is **finite** when $|G| \in \mathbb{N}$, i.e. when it has a finite number of elements. A **compact** group is a group that is also a compact topological space with continuous group operation. Every finite group is also compact (with a discrete topology).

We now present two examples of finite groups that we used throughout our experiments, the *cyclic* and the *dihedral groups*.

**Definition A.2** (Cyclic Group). The *cyclic group* $C_N$ of order $N \in \mathbb{N}$ is the group of $N$ discrete rotations by angles which are integer multiples of $\frac{2\pi}{N}$, i.e. $\{R_{p\frac{2\pi}{N}} \mid p \in [0, 1, \ldots, N-1]\}$.

The binary operation combines two rotations to generate the sum of the two rotations. This is represented by integer sum modulo $N$, i.e.

$$R_{p\frac{2\pi}{N}} \cdot R_{q\frac{2\pi}{N}} = R_{(p+q \mod N)\frac{2\pi}{N}}$$

**Definition A.3** (Dihedral Group). The *dihedral group* $D_N$ of order $2N \in \mathbb{N}$ is the group of $N$ discrete rotations (by angles which are integer multiples of $\frac{2\pi}{N}$) and $N$ reflections (generated by a reflection along an axis followed by any of the $N$ rotations), i.e.

$$\{R_{p\frac{2\pi}{N}} \mid p \in [0, 1, \ldots, N-1]\} \cup \{R_{p\frac{2\pi}{N}}F \mid p \in [0, 1, \ldots, N-1]\}$$

where $F$ is a reflection along an axis. Note that the group $D_N$ has size $2N$ and contains the group $C_N$ as a subgroup.

Another important concept is that of *group action*:

**Definition A.4** (Group Action). The *action* of a group $G$ on a set $\mathcal{X}$ is a map $. : G \times \mathcal{X} \to \mathcal{X}, \ (g, x) \mapsto g.x$ satisfying the following axioms:

- identity: $\forall x \in \mathcal{X} \quad e.x = x$

- compatibility: $\forall a, b \in G \ \forall x \in \mathcal{X} \quad a.(b.x) = (ab).x$

For example, a group can act on functions over the group's elements: given a signal $x : G \to \mathbb{R}$, the action of $g \in G$ on $x$ is defined as $[g.x](h) := x(g^{-1}h)$, i.e., $g$ "translates" the function $x$.

The **orbit** of $x \in \mathcal{X}$ through $G$ is the set $G.x := \{g.x | g \in G\}$. The orbits of the elements in $\mathcal{X}$ form a *partition* of $\mathcal{X}$. By considering the equivalence relation $\forall x, y \in \mathcal{X} \quad x \sim_G y \iff x \in G.y$ (or, equivalently, $y \in G.x$), one can define the **quotient space** $\mathcal{X}/G := \{G.x | x \in \mathcal{X}\}$, i.e. the set of all different orbits.

**Definition A.5** (Linear Representation). Given a group $G$ and a vector space $V$, a linear representation of $G$ is a homomorphism $\rho : G \to \mathrm{GL}(V)$ associating to each element of $g \in G$ an invertible matrix acting on $V$, such that:

$$\forall g, h \in G, \quad \rho(gh) = \rho(g)\rho(h).$$

i.e., the matrix multiplication $\rho(g)\rho(h)$ needs to be compatible with the group composition $gh$.

The most simple representation is the *trivial representation* $\psi : G \to \mathbb{R}, g \mapsto 1$, mapping all elements to the multiplicative identity $1 \in V = \mathbb{R}$. The common 2-dimensional rotation matrices are an example of representation on $V = \mathbb{R}^2$ of the group $\mathrm{SO}(2)$:

$$\rho(r_\theta) = \begin{bmatrix} \cos\theta & -\sin\theta \\ \sin\theta & \cos\theta \end{bmatrix}$$

with $\theta \in [0, 2\pi)$. In the complex field $\mathbb{C}$, circular harmonics are other representations of the rotation group:

$$\psi_k(r_\theta) = e^{-ik\theta} \in V = \mathbb{C}^1$$

where $k \in \mathbb{Z}$ is the harmonic's frequency. Similarly, these representations can be constructed also for the finite *cyclic group* $\mathrm{C}_N \cong \mathbb{Z}/N\mathbb{Z}$:

$$\psi_k(p) = e^{-ikp\frac{2\pi}{N}} \in V = \mathbb{C}^1$$

with $p \in \{0, 1, \dots, N-1\}$.

**Regular Representation** A particularly important representation is the *regular representation* $\rho_{\mathrm{reg}}$ of a *finite* group $G$. This representation acts on the space $V = \mathbb{R}^{|G|}$ of vectors representing functions over the group $G$. The regular representation $\rho_{\mathrm{reg}}(g)$ of an element $g \in G$ is a $|G| \times |G|$ *permutation matrix*. Each vector $\boldsymbol{x} \in V = \mathbb{R}^{|G|}$ can be interpreted as a function over the group, $x : G \to \mathbb{R}$, with $x(g_i)$ being $i$-th entries of $\boldsymbol{x}$. Then, the matrix-vector multiplication $\rho_{\mathrm{reg}}(g)\boldsymbol{x}$ represents the action of $g$ on the function $x$ which generates the "translated" function $g.x$. These representations are of high importance because they describe the features of group convolution networks.

**Direct Sum** Given two representations $\rho_1 : G \to \mathrm{GL}(\mathbb{R}^{n_1})$ and $\rho_2 : G \to \mathrm{GL}(\mathbb{R}^{n_2})$, their *direct sum* $\rho_1 \oplus \rho_2 : G \to \mathrm{GL}(\mathbb{R}^{n_1+n_2})$ is a representation obtained by stacking the two representations as follows:

$$(\rho_1 \oplus \rho_2)(g) = \begin{bmatrix} \rho_1(g) & 0 \\ 0 & \rho_2(g) \end{bmatrix}.$$

Note that this representation acts on $\mathbb{R}^{n_1+n_2}$ which contains the concatenation of the vectors in $\mathbb{R}^{n_1}$ and $\mathbb{R}^{n_2}$. By combining $c$ copies of the *regular representation* via the direct sum, one obtains a representation of $G$ acting on the features of a group convolution network with $c$ channels.

**Fourier Transform** The classical Fourier analysis of periodic discrete functions can be framed as the representation theory of the $\mathbb{Z}/N\mathbb{Z}$ group. This is summarised by the following result: the regular representation of $G = \mathbb{Z}/N\mathbb{Z}$ is equivalent to the direct sum of all circular harmonics with frequency $k \in \{0, \dots, N-1\}$, i.e. there exists a unitary matrix $\boldsymbol{F} \in \mathbb{C}^{N \times N}$ such that:

$$\rho_{\mathrm{reg}}(p) = \boldsymbol{F}^* \left( \bigoplus_k \psi_k(p) \right) \boldsymbol{F}$$

for $p \in \{0, \dots, N-1\} = \mathbb{Z}/N\mathbb{Z}$ and where $\psi_k$ is the circular harmonic of frequency $k$ as defined earlier in this section. The unitary matrix $\boldsymbol{F}$ is what is typically referred to as the (unitary) *discrete Fourier Transform* operator, while its conjugate transpose $\boldsymbol{F}^*$ is the *Inverse Fourier Transform* one.

By indexing the dimensions of the vector space $V \cong \mathbb{C}^N$ on which $\rho_{\mathrm{reg}}$ acts with the elements $p \in \{0, \dots, N-1\}$ of $G = \mathbb{Z}/N\mathbb{Z}$, the Fourier transform matrix can be explicitly constructed as

$$F_{k,p} = \frac{1}{\sqrt{N}} \psi_k(p) = \frac{1}{\sqrt{N}} e^{-ikp\frac{2\pi}{N}} \tag{8}$$

One can verify that the matrix is indeed unitary.

In representation theoretic terms, the circular harmonics $\{\psi_k\}_k$ are referred to as the **irreducible representations** (or *irreps*) of the group $G = \mathbb{Z}/N\mathbb{Z}$. While we are mostly interested in commutative (abelian) finite groups in this work, this construction can be easily extended to square-integrable signals over non-abelian compact groups by using the more general concept of irreducible representations Behboodi et al. (2022); Cesa et al. (2022). See also Serre (1977) for rigorous details about representation theory.

**Notation**    Given a signal $x : G \to \mathbb{C}$, we typically write $\boldsymbol{x} \in V = \mathbb{C}^{|G|}$ to refer to the vector containing the values of $x$ at each group element. Note that this is the vector space on which $\rho_{\text{reg}}$ acts. We also use $\hat{\boldsymbol{x}} = \boldsymbol{F}\boldsymbol{x}$ to indicate the vector of Fourier coefficients and $\hat{x}$ to indicate the function associating to each frequency $k$ (or, equivalently, irrep $\psi_k$) the corresponding Fourier coefficient $\hat{x}(\psi_k) \in \mathbb{C}$.

**Group Convolution**    Given a vector space $V$ associated with a representation $\rho : G \to \text{GL}(V)$ of a group $G$, the *group convolution* $\boldsymbol{w} \circledast_G \boldsymbol{x} \in \mathbb{C}^{|G|}$ of two elements $\boldsymbol{x}, \boldsymbol{w} \in V$ is defined as

$$\forall g \in G \quad (\boldsymbol{w} \circledast_G \boldsymbol{x})(g) := \boldsymbol{w}^T \rho(g)^T \boldsymbol{x} \ . \tag{9}$$

What we defined is technically a **group cross-correlation** and so it differs from the usual definitions of convolution over groups. We still refer to it as group convolution to follow the common terminology in the Deep Learning literature. One can prove that group convolution is **equivariant** to $G$, i.e.:

$$\boldsymbol{w} \circledast_G \rho(g)\boldsymbol{x} = \rho_{\text{reg}}(g) \, (\boldsymbol{w} \circledast_G \boldsymbol{x})$$

If $V = \mathbb{C}^{|G|}$, then $x, w : G \to \mathbb{C}$ can be interpreted as signals over the group, and group convolution take the more familiar form

$$(w \circledast_G x)(g) = \sum_{h \in G} w(g^{-1}h)x(h) \tag{10}$$

Finally, we recall two important properties of the Fourier transform: given two signals $w, x : G \to \mathbb{C}$, the following properties hold

$$\widehat{g.x}(\psi) = \psi(g)\hat{x}(\psi) \tag{11}$$

$$\widehat{w \circledast_G x}(\psi) = \hat{w}(\psi)\hat{x}(\psi) \tag{12}$$

Eq. 11 guarantees that a transformation by $g$ of $x$ does not mix the coefficients associated to different irreps/frequencies. Eq. 12 is the typical *convolution theorem*.

Finally, some of these results related to the Fourier transform are generalized by the following theorems. This last property is related to the more general result expressed by *Schur's Lemma*:

**Theorem A.6** (Schur's Lemma). *Let $G$ be a compact group (not necessarily abelian) and $\psi_1$ and $\psi_2$ two irreps of $G$. Then, there exists a non-zero linear map $\boldsymbol{W}$ such that*

$$\psi_1(g)\boldsymbol{W} = \boldsymbol{W}\psi_2(g)$$

*for any $g \in G$ (i.e. an* equivariant *linear map) if and only if $\psi_1$ and $\psi_2$ are equivalent representations, i.e. they differ at most by a change of basis $Q$: $\psi_1(g) = Q\psi_2(g)Q^{-1}$ for all $g \in G$.*

A second result generalizes the notion of Fourier transform to representations beyond the regular one:

**Theorem A.7** (Peter-Weyl Theorem). *Let $G$ be a compact group and $\rho$ a unitary representation of $G$. Then, $\rho$ decomposes into a direct sum of irreducible representations of $G$, up to a change of basis, i.e.*

$$\rho(g) = U \left( \bigoplus_{\psi} \bigoplus_{i}^{m_\psi} \psi \right) U^T$$

*Each irrep $\psi$ can appear in the decomposition with a multiplicity $m_\psi \geq 0$.*

In particular, in the case of a regular representation, the change of basis is given precisely by the Fourier transform matrix $\boldsymbol{F}$.

The combination of these two results is typically used in steerable CNNs and other equivariant designs to characterize arbitrary equivariant networks by reducing the study to individual irreducible components. See also the next section.

### A.1   Equivariant Networks

We now discuss some popular equivariant neural network designs.

In this work, we consider *real* valued networks with 2 layers, i.e., we limit our discussion to a hypothesis class of the form

$$\mathcal{H} := \left\{ \boldsymbol{u}^T \sigma(\boldsymbol{W}\boldsymbol{x}) \right\}.$$

In a $G$-equivariant network, the group $G$ carries an *action* on the intermediate features of the model; these actions are specified by *group representations*, which we introduced in the previous section. We assume the action of an element $g \in G$ on the input and the output of the first layer is given respectively by the matrices $\rho_0(g)$ and $\rho_1(g)$. The first layer, including the activation function, is **equivariant** with respect to $G$ if:

$$\forall g \in G, \quad \sigma(\boldsymbol{W}\rho_0(g)\boldsymbol{x}) = \rho_1(g)\sigma(\boldsymbol{W}\boldsymbol{x})$$

Instead, the last layer of the network is invariant with respect to the action of the group $G$ on the input data via $\rho_0$, which means that:

$$\forall g \in G, \quad \boldsymbol{u}^T \sigma(\boldsymbol{W}\boldsymbol{x}) = \boldsymbol{u}^T \rho_1(g)\sigma(\boldsymbol{W}\boldsymbol{x}) = \boldsymbol{u}^T \sigma(\boldsymbol{W}\rho_0(g)\boldsymbol{x})$$

**GCNNs**   A typical way to construct networks equivariant to a finite group $G$ is via **group convolution**, which we introduced in the previous section; this is the *Group Convolution neural network* (GCNN) design Cohen and Welling (2016a). Indeed, if the representation $\rho_1$ is chosen to be the *direct sum* of $c_1$ copies of the *regular representation* $\rho_{\mathrm{reg}}$ of $G$, i.e. $\rho_1 = \bigoplus^{c_1} \rho_{\mathrm{reg}}$ and the input $\rho_0 = \bigoplus^{c_0} \rho_{\mathrm{reg}}$ also consists of $c_0$ copies of the regular representation, then the linear layer $\boldsymbol{W}$ can always be expressed as $c_1 \times c_0$ group-convolution linear operators as in eq. 9:

$$\boldsymbol{W} = \left[ \begin{array}{c|c|c|c} \boldsymbol{W}_{(1,1)} & \boldsymbol{W}_{(2,1)} & \cdots & \boldsymbol{W}_{(c_0,1)} \\ \hline \boldsymbol{W}_{(1,2)} & \boldsymbol{W}_{(2,2)} & \cdots & \boldsymbol{W}_{(c_0,2)} \\ \hline \vdots & \vdots & \ddots & \vdots \\ \hline \boldsymbol{W}_{(1,c_1)} & \boldsymbol{W}_{(2,c_1)} & \cdots & \boldsymbol{W}_{(c_0,c_1)} \end{array} \right]$$

where each $\boldsymbol{W}_{(i,j)} \in \mathbb{R}^{|G| \times |G|}$ is a $G$-circular matrix (i.e. rows are "rotations" of each others via $\rho_{\mathrm{reg}}$) of the form:

$$\boldsymbol{W}_{(i,j)} = \left[ \begin{array}{c} \boldsymbol{w}_{(i,j)}^T \rho_{\mathrm{reg}}(g_1) \\ \vdots \\ \boldsymbol{w}_{(i,j)}^T \rho_{\mathrm{reg}}(g_{|G|}) \end{array} \right]$$

In this case, note that the representation $\rho_1$ acts via permutation and, therefore, any activation function $\sigma$ which is applied to each entry of $\boldsymbol{W}\boldsymbol{x}$ entry-wise is equivariant: this includes the typical activations used in deep learning (e.g. ReLU).

**General linear layer**   The representation theoretic tools we introduced earlier allow for a quite general framework to describe a wide family of equivariant networks beyond group-convolution by considering the generalization of the Fourier transform we discussed in the previous section. By using Theorem A.7, a representation $\rho_l$ (for $l = 0, 1$) decomposes into direct sum of irreps as follows:

$$\rho_l = U_l \left( \bigoplus_{\psi} \bigoplus_{i=1}^{m_{l,\psi}} \psi \right) U_l^\top,$$

| Activation function | Definition |
|---|---|
| Norm ReLU Weiler et al. (2018b); Worrall et al. (2017) | $\eta(\|\boldsymbol{x}\|) = \text{ReLU}(\|\boldsymbol{x}\| - b)$   $(b <= 0)$ |
| Squashing Sabour et al. (2017) | $\eta(\|\boldsymbol{x}\|) = \frac{\|\boldsymbol{x}\|^2}{\|\boldsymbol{x}\|^2 + 1}$ |
| Gated Weiler et al. (2018b) | $\eta(\|\boldsymbol{x}\|) = \frac{1}{1 + e^{-s(\boldsymbol{x})}} \|\boldsymbol{x}\|$ |

Table 1: Equivariant activation functions

where $U_l$ is an orthogonal matrix, and $m_{l,\psi}$ is the multiplicity of the irrep $\psi$ in the representation $\rho_l$. If $\dim_\psi$ is the dimensionality of the irrep $\psi$, the width of the network $n$ is equal to $\sum m_{1,\psi} \dim_\psi$, and the input dimension $d$ is given by $\sum m_{0,\psi} \dim_\psi$.

Next, by combining this result with Theorem A.6, an equivariant network can be parameterized entirely in terms of irreps, which is analogous to a Fourier space parameterization of a convolutional network. Defining $\widehat{\boldsymbol{W}}_l = U_l^{-1} \boldsymbol{W}_l U_{l-1}$, the equivariance condition writes as

$$\forall g \in G, \quad \widehat{\boldsymbol{W}}_l \left( \bigoplus_\psi \bigoplus_{i=1}^{m_{l-1,\psi}} \psi(g) \right) = \left( \bigoplus_\psi \bigoplus_{i=1}^{m_{l,\psi}} \psi(g) \right) \widehat{\boldsymbol{W}}_l.$$

This induces a block structure in $\hat{\boldsymbol{W}}_l$, where the $(\psi, i; \psi', j)$-th block block maps the $i$-th input block transforming according to $\psi$ to the $j$-th output block transforming under $\psi'$, with $i \in [m_{l-1,\psi}]$ and $j \in [m_{l,\psi'}]$. By Theorem A.6, there are no linear maps equivariant to $\psi$ and $\psi'$ whenever these representations are in-equivalent; this implies the matrix $\hat{\boldsymbol{W}}_l$ is sparse since its $(\psi, i; \psi', j)$-th block is non-zero only when $\psi \cong \psi'$. Then, the *non-zero* $(\psi, i, j)$-th block (here, we drop the redundant $\psi'$ index) is denoted by $\hat{w}_{i,j}(\psi)$ and should commute with $\psi(g)$ for any $g \in G$. This is achieved by expressing this matrix as a linear combination of a few fixed basis matrices spanning the space; see Behboodi et al. (2022); Cesa et al. (2022) for more details.

In the special case of GCNN with abelian group $G$, each irrep in the intermediate features occurs exactly $c_1$ times. Similarly, if the input representation $\rho_0$ consists of $c_0$ copies of the regular representation, each irrep appears exactly $c_0$ times in $\rho_0$. Then, $\hat{w}_{i,j}(\psi)$ is the Fourier transform of the filter $\boldsymbol{w}_{(i,j)} \in \mathbb{R}^{|G|}$ at the frequency/irrep $\psi$.

**Activation function $\sigma$**  Each module in an equivariant network should commute with the action of the group on its own input in order to guarantee the overall equivariance of the model. Hence, the activation function $\sigma$ used in the intermediate layer should be equivariant with respect to $\rho_1(g)$ as well, i.e., $\sigma(\rho_1(g)\boldsymbol{x}) = \rho_2(g)\sigma(\boldsymbol{x}), \forall g \in G$. Different activation functions are used in the literature. As we use unitary representations, one can use norm nonlinearities, $\sigma(\boldsymbol{x}) = \eta(\|\boldsymbol{x}\|) \frac{\boldsymbol{x}}{\|\boldsymbol{x}\|}$ for a suitable function $\eta : \mathbb{R}_0^+ \to \mathbb{R}_0^+$. It can be seen that $\sigma(\rho_1(g)\boldsymbol{x}) = \rho_1(g)\sigma(\boldsymbol{x})$. Some examples are norm ReLU Worrall et al. (2017), squashing Sabour et al. (2017) and gated nonlinearities Weiler et al. (2018b).

As introduced in the GCNN paragraph, when the intermediate representation is built from the regular representation, any pointwise activation is admissible: since the group's action permutes the features' entries in the spatial domain, it intertwines with the non-linearity applied entry-wise. There exist also other type of activations, such as tensor-product non-linearities (based on the Clebsh-Gordan transform) but we do not consider them here.

**Last layer $\boldsymbol{u}$**  The last layer maps the features after the activation to a single scalar and is assumed to be invariant. This linear layer can be thought of as a special case of the general framework above, with the choice of output representation $\rho_2$ being a collection of *trivial representations*, i.e., $\rho_2(g) = I$ is the identity. Since the trivial representation is an irrep, by Theorem A.6, this linear map only captures the invariant information in the output of the activation function. In the case of a GCNN with pointwise activation $\sigma$, this layer acts as an averaging pooling operator over the $|G|$ entries of each channel and learns the weights to linearly combine the $c_1$ independent channels. Note that one can also include a custom **pooling operator** $P(\cdot)$ (e.g., max-pooling) between the activation layer $\sigma$ and the final layer $\boldsymbol{u}$ like in eq. 3: this gives additional freedom to construct the final invariant features beyond just average pooling.

# B  Mathematical Preliminaries

This section gathers the main tools we will use throughout the proofs.

First, we introduce the decoupling technique Foucart and Rauhut (2013), which will be used in the proof of the next lemma.

**Theorem B.1.** *Let $\boldsymbol{\epsilon} = (\epsilon_1, \ldots, \epsilon_m)$ be a sequence of independent random variables with $\mathbb{E}\epsilon_i = 0$ for all $i \in [m]$. Let $\boldsymbol{x}_{j,k}$, $j, k \in [m]$ be a double sequence of elements in a finite-dimensional vector space $V$. If $F : V \to \mathbb{R}$ is a convex function, then:*

$$\mathbb{E}F\left(\sum_{\substack{j,k=1 \\ j \neq k}}^{m} \epsilon_j \epsilon_k \boldsymbol{x}_{j,k}\right) \leq \mathbb{E}F\left(4\sum_{j,k=1}^{m} \epsilon_j \epsilon'_k \boldsymbol{x}_{j,k}\right), \tag{13}$$

*where $\boldsymbol{\epsilon}'$ is an independent copy of $\boldsymbol{\epsilon}$.*

The following lemma can be found in Foucart and Rauhut (2013), namely inequality (8.22), within the proof of Proposition 8.13. We re-state the lemma and the proof here, as it is of independent interest.

**Lemma B.2.** *For any positive semidefinite matrix $\boldsymbol{B}$, for $8\beta \|\boldsymbol{B}\|_{2\to2} < 1$, and Rademacher vector $\boldsymbol{\epsilon}$, we have*

$$\mathbb{E}_{\boldsymbol{\epsilon}}\left(\exp\left(\beta\boldsymbol{\epsilon}^\top \boldsymbol{B}\boldsymbol{\epsilon}\right)\right) \leq \exp\left(\frac{\beta\operatorname{Tr}(\boldsymbol{B})}{1 - 8\beta\|\boldsymbol{B}\|_{2\to2}}\right).$$

*Proof.* We start by using Theorem B.1:

$$\mathbb{E}\left(\exp\left(\beta\boldsymbol{\epsilon}^\top \boldsymbol{B}\boldsymbol{\epsilon}\right)\right) = \mathbb{E}\left(\exp\left(\beta\sum_{j=1} B_{jj} + \beta\sum_{j \neq k} \epsilon_j \epsilon_k B_{jk}\right)\right)$$

$$\leq \left(\exp\left(\beta\operatorname{Tr}(\boldsymbol{B})\right)\right)\mathbb{E}\left(\exp\left(4\beta\sum_{j,k}\epsilon_j\epsilon'_k B_{jk}\right)\right)$$

Then using the inequality $\mathbb{E}\exp(\beta\sum_{l=1}^{m} a_1\epsilon_l) \leq \mathbb{E}\exp(\beta^2 \sum_{l=1}^{m} a_1^2/2)$, we have:

$$\mathbb{E}\left(\exp\left(4\beta\sum_{j,k}\epsilon_j\epsilon'_k B_{jk}\right)\right) \leq \mathbb{E}\left(\exp\left(8\beta^2\sum_{k}\left(\sum_{j}\epsilon_j B_{jk}\right)^2\right)\right)$$

Next, we can see that:

$$\sum_{k}\left(\sum_{j}\epsilon_j B_{jk}\right)^2 = \|\boldsymbol{B}\boldsymbol{\epsilon}\|^2 = \left\|\boldsymbol{B}^{1/2}\boldsymbol{B}^{1/2}\boldsymbol{\epsilon}\right\|^2$$

$$\leq \left\|\boldsymbol{B}^{1/2}\right\|_{2\to2}^2 \left\|\boldsymbol{B}^{1/2}\boldsymbol{\epsilon}\right\|^2 = \|\boldsymbol{B}\|_{2\to2}\,\boldsymbol{\epsilon}^\top \boldsymbol{B}\boldsymbol{\epsilon}.$$

Therefore, for $8\beta\|\boldsymbol{B}\|_{2\to2} < 1$, we get:

$$\mathbb{E}\left(\exp\left(\beta\boldsymbol{\epsilon}^\top \boldsymbol{B}\boldsymbol{\epsilon}\right)\right) \leq \exp\left(\beta\operatorname{Tr}(\boldsymbol{B})\right)\mathbb{E}\left(\exp\left(8\beta^2\|\boldsymbol{B}\|_{2\to2}\,\boldsymbol{\epsilon}^\top \boldsymbol{B}\boldsymbol{\epsilon}\right)\right)$$

$$\leq \exp\left(\beta\operatorname{Tr}(\boldsymbol{B})\right)\mathbb{E}\left(\left(\exp\left(\beta\boldsymbol{\epsilon}^\top \boldsymbol{B}\boldsymbol{\epsilon}\right)\right)\right)^{8\beta\|\boldsymbol{B}\|_{2\to2}}.$$

Rearranging the terms yields the final result. $\qquad\square$

## B.1 Rademacher Complexity Bounds

The generalization analysis of this paper is based on the Rademacher analysis. We summarize the main theorems here. The terms $\mathcal{L}(h)$ and $\hat{\mathcal{L}}(h)$ denotes, respectively, the test and the training error, formally defined as follows:

$$\mathcal{L}(h) = \mathbb{E}_{\boldsymbol{x} \sim \mathbb{P}_{\boldsymbol{x}}} \left( \ell \circ h(\boldsymbol{x}) \right), \quad \hat{\mathcal{L}}(h) = \frac{1}{m} \sum_{i=1}^{m} \ell \circ h(\boldsymbol{x}_i). \tag{14}$$

The empirical Rademacher complexity is defined as:

$$\mathcal{R}_{\mathcal{S}}(\mathcal{G}) = \mathbb{E}_{\epsilon} \sup_{g \in \mathcal{G}} \frac{1}{m} \sum_{i=1}^{m} \epsilon_i g(\boldsymbol{x}_i), \tag{15}$$

**Theorem B.3** (Theorem 26.5, Shalev-Shwartz and Ben-David (2014)). *Let $\mathcal{H}$ be a family of functions, and let $\mathcal{S}$ be the training sequence of $m$ samples drawn from the distribution $\mathcal{D}^m$. Let $\ell$ be a real-valued loss function satisfying $|\ell| \leq c$. Then, for $\delta \in (0, 1)$, with probability at least $1 - \delta$ we have, for all $h \in \mathcal{H}$,*

$$\mathcal{L}(h) \leq \hat{\mathcal{L}}(h) + 2\mathcal{R}_{\mathcal{S}}(\ell \circ \mathcal{H}) + 4c\sqrt{\frac{2\log(4/\delta)}{m}}. \tag{16}$$

Using Theorem B.3, we can focus on finding upper bounds for the empirical Rademacher complexity of $\ell \circ \mathcal{H}$. Below, we show how to remove the loss function $\ell(\cdot)$ and focus on $\mathcal{H}$.

A function $\phi : \mathbb{R} \to \mathbb{R}$ is called a contraction if $|\phi(x) - \phi(y)| \leq |x - y|$ for all $x, y \in \mathbb{R}$, or equivalently if the function is 1-Lipschitz. The contraction lemma is a standard result in equation 4.20 of Ledoux and Talagrand (2011). We will use that for the Rademacher complexity analysis.

**Lemma B.4.** *Let $\phi_i : \mathbb{R} \to \mathbb{R}$ be contractions such that $\phi_i(0) = 0$. If $G : \mathbb{R} \to \mathbb{R}$ is a convex and increasing function, then*

$$\mathbb{E}G \left( \sup_{t \in T} \sum_{i=1}^{m} \epsilon_i \phi_i(t_i) \right) \leq \mathbb{E}G \left( \sup_{t \in T} \sum_{i=1}^{m} \epsilon_i t_i \right). \tag{17}$$

Let's rewrite the Rademacher term, including the loss function explicitly:

$$\mathcal{R}_{\mathcal{S}}(\ell \circ \mathcal{H}) = \mathbb{E}_{\boldsymbol{\epsilon}} \left[ \sup_{h \in \mathcal{H}} \frac{1}{m} \sum_{i=1}^{m} \epsilon_i \ell \circ h(\boldsymbol{x}_i) \right].$$

We assume a 1-Lipschitz loss function, for which we can use the contraction lemma, 4.20 in Ledoux and Talagrand (2011), to get:

$$\mathcal{R}_{\mathcal{S}}(\ell \circ \mathcal{H}) \leq \mathcal{R}_{\mathcal{S}}(\mathcal{H}).$$

So finding an upper bound on $\mathcal{R}_{\mathcal{S}}(\mathcal{H})$ is sufficient for the generalization error analysis.

## B.2 Dudley's Inequality and Covering Number Bounds

Although, for most of the proofs, we try to directly bound the Rademacher complexity, there is a way of bounding it using the covering number of the underlying set. The covering number of a set $\mathcal{G}$ is the minimum number of balls required to cover the set $\mathcal{G}$ with the centers within the set, the distances defined according to a metric $d$, and the radius fixed of a given size $\epsilon$. It is denoted by $\mathcal{N}(\mathcal{G}, d_0, \epsilon)$. The covering can be understood is a way approximating each point in $\mathcal{G}$ by a set of finite points of $\mathcal{G}$ with the fidelity $\epsilon$. The following result can be found in many references including Foucart and Rauhut (2013); Bartlett et al. (2017); Mohri et al. (2018).

**Theorem B.5** (Dudley's Inequality)**.** *The Rademacher complexity can be upper bounded using the covering number as follows*

$$\mathbb{E}_\epsilon \sup_{g \in \mathcal{G}} \frac{1}{m} \sum_{i=1}^m \epsilon_i g(\boldsymbol{x}_i) \leq \inf_{\alpha > 0} \left( 4\alpha + \frac{4\sqrt{2}}{\sqrt{m}} \int_\alpha^{\frac{B_{\mathcal{S};\mathcal{G}}}{2}} \sqrt{\log \mathcal{N}(\mathcal{G}, d, u)} \, \mathrm{d}u \right) \tag{18}$$

*where $d_0(s, t)$ is metric defined on $\mathcal{G}$ for any $s, t \in \mathcal{G}$ as follows:*

$$d(s, t) = \left( \frac{1}{m} \sum_{i=1}^m (s(\boldsymbol{x}_i) - t(\boldsymbol{x}_i))^2 \right)^{1/2} \tag{19}$$

*and*

$$B_{\mathcal{S};\mathcal{G}} = \sup_{g \in \mathcal{G}} \sqrt{\frac{\sum_{i=1}^m g(\boldsymbol{x}_i)^2}{m}}.$$

The idea is to find a bound on the covering number and then use Dudley's inequality to bound the Rademacher complexity.

## C   Proof of Main Theorems

### C.1   Rademacher Complexity Bounds for Group Convolutional Networks

Consider the group convolutional network with $c_0$ input channel, $c_1$ intermediate channels, and a last layer with pooling and aggregation. The input space is assumed to be $\mathbb{R}^{|G| \times c_0}$. Remember that the network was given as follows:

$$h_{\boldsymbol{u},\boldsymbol{w}}(\boldsymbol{x}) = \boldsymbol{u}^\top P \circ \sigma \begin{pmatrix} \sum_{k=1}^{c_0} \boldsymbol{w}_{(1,k)} \circledast_G \boldsymbol{x}(k) \\ \vdots \\ \sum_{k=1}^{c_0} \boldsymbol{w}_{(c_1,k)} \circledast_G \boldsymbol{x}(k) \end{pmatrix} = \boldsymbol{u}^\top P \circ \sigma(\boldsymbol{W}\boldsymbol{x}),$$

where $P(\cdot)$ is the pooling operation. We denote the first convolution layer with the circulant matrix $\boldsymbol{W}\boldsymbol{x}$.

To find a bound on the Rademacher complexity, We start with the following inequality, which is shared across the proofs of some of the other theorems in the paper:

$$\mathbb{E}_{\boldsymbol{\epsilon}} \left( \sup_{\boldsymbol{u},\boldsymbol{w}} \frac{1}{m} \sum_{i=1}^m \epsilon_i \boldsymbol{u}^\top P \circ \sigma(\boldsymbol{W}\boldsymbol{x}_i) \right) = \frac{M_1}{m} \mathbb{E}_{\boldsymbol{\epsilon}} \left( \sup_{\boldsymbol{w}} \left\| \sum_{i=1}^m \epsilon_i P \circ \sigma(\boldsymbol{W}\boldsymbol{x}_i) \right\| \right). \tag{20}$$

The above result follows from the Cauchy-Schwartz inequality and peels off the last aggregation layer. We continue the proofs from this step.

### C.2   Proof for Positively Homogeneous Activation Functions with Average Pooling (Theorem C.1)

**Theorem C.1.** *Consider the hypothesis space $\mathcal{H}$ defined in eq. 4. If $P(\cdot)$ is the average pooling operation, $\sigma(\cdot)$ is a 1-Lipschitz positively homogeneous activation function, then with probability at least $1 - \delta$ and for all $h \in \mathcal{H}$, we have:*

$$\mathcal{L}(h) \leq \hat{\mathcal{L}}(h) + 2\frac{b_x M_1 M_2}{\sqrt{m}} + 4\sqrt{\frac{2\log(4/\delta)}{m}}.$$

*Proof.* For an input $\boldsymbol{x} \in \mathbb{R}^{|G| \times c_0}$, the action of the group $G$ on each channel is given by the permutation of the entries. Suppose that the $G-$permutations are given by $\Pi_1, \ldots, \Pi_{|G|}$. We have:

$$
\mathbb{E}_{\boldsymbol{\epsilon}} \left( \sup_{\boldsymbol{w}} \left\| \sum_{i=1}^{m} \epsilon_i P \circ \sigma(\boldsymbol{W} \boldsymbol{x}_i) \right\| \right) = \mathbb{E}_{\boldsymbol{\epsilon}} \left( \sup_{\boldsymbol{w}} \left\| \sum_{i=1}^{m} \epsilon_i \begin{pmatrix} \frac{1}{|G|} \mathbf{1}^\top \sigma \left( \sum_{k=1}^{c_0} \boldsymbol{w}_{(1,k)} \circledast_G \boldsymbol{x}_i(k) \right) \\ \vdots \\ \frac{1}{|G|} \mathbf{1}^\top \sigma \left( \sum_{k=1}^{c_0} \boldsymbol{w}_{(c_1,k)} \circledast_G \boldsymbol{x}_i(k) \right) \end{pmatrix} \right\| \right)
$$

$$
= \mathbb{E}_{\boldsymbol{\epsilon}} \left( \sup_{\boldsymbol{w}} \left\| \sum_{i=1}^{m} \epsilon_i \sum_{l=1}^{|G|} \frac{1}{|G|} \begin{pmatrix} \sigma \left( \sum_{k=1}^{c_0} \boldsymbol{w}_{(1,k)}^\top \Pi_l \boldsymbol{x}_i(k) \right) \\ \vdots \\ \sigma \left( \sum_{k=1}^{c_0} \boldsymbol{w}_{(c_1,k)}^\top \Pi_l \boldsymbol{x}_i(k) \right) \end{pmatrix} \right\| \right)
$$

$$
\leq \mathbb{E}_{\boldsymbol{\epsilon}} \left( \sup_{\boldsymbol{w}} \sum_{l=1}^{|G|} \frac{1}{|G|} \left\| \sum_{i=1}^{m} \epsilon_i \begin{pmatrix} \sigma \left( \sum_{k=1}^{c_0} \boldsymbol{w}_{(1,k)}^\top \Pi_l \boldsymbol{x}_i(k) \right) \\ \vdots \\ \sigma \left( \sum_{k=1}^{c_0} \boldsymbol{w}_{(c_1,k)}^\top \Pi_l \boldsymbol{x}_i(k) \right) \end{pmatrix} \right\| \right)
$$

$$
\leq \sum_{l=1}^{|G|} \frac{1}{|G|} \mathbb{E}_{\boldsymbol{\epsilon}} \left( \sup_{\boldsymbol{w}} \left\| \sum_{i=1}^{m} \epsilon_i \begin{pmatrix} \sigma \left( \sum_{k=1}^{c_0} \boldsymbol{w}_{(1,k)}^\top \Pi_l \boldsymbol{x}_i(k) \right) \\ \vdots \\ \sigma \left( \sum_{k=1}^{c_0} \boldsymbol{w}_{(c_1,k)}^\top \Pi_l \boldsymbol{x}_i(k) \right) \end{pmatrix} \right\| \right)
$$

where we used triangle inequalities for the first inequality. The second inequality follows from swapping the supremum and the sum. Consider an arbitrary summand for a given $l$. We first use the moment inequality to get:

$$
\mathbb{E}_{\boldsymbol{\epsilon}} \left( \sup_{\boldsymbol{w}} \left\| \sum_{i=1}^{m} \epsilon_i \begin{pmatrix} \sigma \left( \sum_{k=1}^{c_0} \boldsymbol{w}_{(1,k)}^\top \Pi_l \boldsymbol{x}_i(k) \right) \\ \vdots \\ \sigma \left( \sum_{k=1}^{c_0} \boldsymbol{w}_{(c_1,k)}^\top \Pi_l \boldsymbol{x}_i(k) \right) \end{pmatrix} \right\| \right)
$$
$$
\leq \sqrt{\mathbb{E}_{\boldsymbol{\epsilon}} \left( \sup_{\boldsymbol{w}} \left\| \sum_{i=1}^{m} \epsilon_i \begin{pmatrix} \sigma \left( \sum_{k=1}^{c_0} \boldsymbol{w}_{(1,k)}^\top \Pi_l \boldsymbol{x}_i(k) \right) \\ \vdots \\ \sigma \left( \sum_{k=1}^{c_0} \boldsymbol{w}_{(c_1,k)}^\top \Pi_l \boldsymbol{x}_i(k) \right) \end{pmatrix} \right\|^2 \right)}. \tag{21}
$$

We can continue the derivation as follows. First, define $\boldsymbol{w}_{(j,:)} := (\boldsymbol{w}_{j,i})_{i \in [c_0]}$, which is the weights used for generating the channel $j$. Now, using the assumption of positively homogeneity, we have:

$$
\mathbb{E}_{\boldsymbol{\epsilon}} \left( \sup_{\boldsymbol{w}} \left\| \sum_{i=1}^{m} \epsilon_i \begin{pmatrix} \sigma \left( \sum_{k=1}^{c_0} \boldsymbol{w}_{(1,k)}^\top \Pi_l \boldsymbol{x}_i(k) \right) \\ \vdots \\ \sigma \left( \sum_{k=1}^{c_0} \boldsymbol{w}_{(c_1,k)}^\top \Pi_l \boldsymbol{x}_i(k) \right) \end{pmatrix} \right\|^2 \right) = \mathbb{E}_{\boldsymbol{\epsilon}} \left( \sup_{\boldsymbol{w}} \sum_{j=1}^{c_1} \left( \sum_{i=1}^{m} \epsilon_i \sigma \left( \sum_{k=1}^{c_0} \boldsymbol{w}_{(j,k)}^\top \Pi_l \boldsymbol{x}_i(k) \right) \right)^2 \right)
$$

$$
= \mathbb{E}_{\boldsymbol{\epsilon}} \left( \sup_{\boldsymbol{w}} \sum_{j=1}^{c_1} \left\| \boldsymbol{w}_{(j,:)} \right\|^2 \left( \sum_{i=1}^{m} \epsilon_i \sigma \left( \frac{1}{\left\| \boldsymbol{w}_{(j,:)} \right\|} \sum_{k=1}^{c_0} \boldsymbol{w}_{(j,k)}^\top \Pi_l \boldsymbol{x}_i(k) \right) \right)^2 \right),
$$

Since $\|\boldsymbol{w}\|^2 \leq M_2^2$, we have

$$
\sum_{j=1}^{c_1} \left\| \boldsymbol{w}_{(j,:)} \right\|^2 \leq M_2^2.
$$

We use a similar trick to the paper Golowich et al. (2018) to focus only on one channel for the rest:

$$\mathbb{E}_{\boldsymbol{\epsilon}}\left(\sup_{\boldsymbol{w}}\sum_{j=1}^{c_1}\left\|\boldsymbol{w}_{(j,:)}\right\|^2\left(\sum_{i=1}^{m}\epsilon_i\sigma\left(\frac{1}{\left\|\boldsymbol{w}_{(j,:)}\right\|}\sum_{k=1}^{c_0}\boldsymbol{w}_{(j,k)}^{\top}\Pi_l\boldsymbol{x}_i(k)\right)\right)^2\right)$$

$$\leq M_2^2\mathbb{E}_{\boldsymbol{\epsilon}}\left(\sup_{\boldsymbol{w}}\sup_{j\in[c_l]}\left(\sum_{i=1}^{m}\epsilon_i\sigma\left(\frac{1}{\left\|\boldsymbol{w}_{(j,:)}\right\|}\sum_{k=1}^{c_0}\boldsymbol{w}_{(j,k)}^{\top}\Pi_l\boldsymbol{x}_i(k)\right)\right)^2\right)$$

$$\leq M_2^2\mathbb{E}_{\boldsymbol{\epsilon}}\left(\sup_{\tilde{\boldsymbol{w}}:\|\tilde{\boldsymbol{w}}\|\leq 1}\left(\sum_{i=1}^{m}\epsilon_i\sigma\left(\sum_{k=1}^{c_0}\tilde{\boldsymbol{w}}_{(k)}^{\top}\Pi_l\boldsymbol{x}_i(k)\right)\right)^2\right).$$

Now, we can use the contraction lemma and get:

$$\mathbb{E}_{\boldsymbol{\epsilon}}\left(\sup_{\tilde{\boldsymbol{w}}:\|\tilde{\boldsymbol{w}}\|\leq 1}\left(\sum_{i=1}^{m}\epsilon_i\sigma\left(\sum_{k=1}^{c_0}\tilde{\boldsymbol{w}}_{(k)}^{\top}\Pi_l\boldsymbol{x}_i(k)\right)\right)^2\right)\leq\mathbb{E}_{\boldsymbol{\epsilon}}\left(\sup_{\tilde{\boldsymbol{w}}:\|\tilde{\boldsymbol{w}}\|\leq 1}\left(\sum_{i=1}^{m}\epsilon_i\sum_{k=1}^{c_0}\tilde{\boldsymbol{w}}_{(k)}^{\top}\Pi_l\boldsymbol{x}_i(k)\right)^2\right)$$

$$\leq\mathbb{E}_{\boldsymbol{\epsilon}}\left(\sup_{\tilde{\boldsymbol{w}}:\|\tilde{\boldsymbol{w}}\|\leq 1}\left(\sum_{k=1}^{c_0}\tilde{\boldsymbol{w}}_{(k)}^{\top}\left(\sum_{i=1}^{m}\epsilon_i\Pi_l\boldsymbol{x}_i(k)\right)\right)^2\right)$$

$$\leq\mathbb{E}_{\boldsymbol{\epsilon}}\left(\left\|\sum_{i=1}^{m}\epsilon_i\Pi_l\boldsymbol{x}_i\right\|^2\right)\leq mb_x^2,\tag{22}$$

where we used the fact that the permutation matrix $\Pi_l$ is unitary. Putting all this together, we get:

$$\sum_{l=1}^{|G|}\frac{1}{|G|}\mathbb{E}_{\boldsymbol{\epsilon}}\left(\sup_{\boldsymbol{w}}\left\|\sum_{i=1}^{m}\epsilon_i\begin{pmatrix}\sigma\left(\sum_{k=1}^{c_0}\boldsymbol{w}_{(1,k)}^{\top}\Pi_l\boldsymbol{x}_i(k)\right)\\\vdots\\\sigma\left(\sum_{k=1}^{c_0}\boldsymbol{w}_{(c_1,k)}^{\top}\Pi_l\boldsymbol{x}_i(k)\right)\end{pmatrix}\right\|\right)\leq\sum_{l=1}^{|G|}\frac{1}{|G|}\frac{b_xM_1M_2}{\sqrt{m}}=\frac{b_xM_1M_2}{\sqrt{m}},$$

which finalizes the desired results. $\qquad\square$

### C.3 Max pooling Operation - Covering Number Based Results

**Single Channel Output.** Before going into another result for max pooling, we focus on a simpler example. We assume ReLU activation throughout this subsection. First, consider the following network:

$$h_{\boldsymbol{w}}(\boldsymbol{x})=P\circ\sigma\left(\sum_{k=1}^{c_0}\boldsymbol{w}_{(1,k)}\circledast_G\boldsymbol{x}(k)\right)$$

where $P(\cdot)$ is the max pooling operation. Just like above, we assume that the norm of the parameters is bounded by $M_2$ while the input norm is bounded by $b_X$. A special case of this setup has been considered in Vardi et al. (2022) in Theorem 7, where $c_0=1$. We recap their proof, providing a more refined final bound. The only difference is that, in our proof, we use Dudley's inequality.

**Theorem C.2** (Single Channel Max pooling)**.** *Consider the hypothesis space $\mathcal{H}$ of single channel group convolutional network with $P(\cdot)$ as max pooling, and $\sigma(\cdot)$ as ReLU. With probability at least $1-\delta$ and for all $h\in\mathcal{H}$, we have:*

$$\mathcal{L}(h)\leq\hat{\mathcal{L}}(h)+\frac{b_xM_2}{\sqrt{m}}\left(144\sqrt{2}\sqrt{\log(2(8\sqrt{m}+2)m|G|+1)}\log(m)\right)+4\sqrt{\frac{2\log(4/\delta)}{m}}.$$

*Proof.* The proof consists of deriving the covering number $\mathcal{N}(\mathcal{H},d,u)$ and then using Dudley's inequality. We have included the essential components of such proofs in the section on mathematical preliminaries.

First, we can rewrite the convolution as:

$$\sum_{k=1}^{c_0} \boldsymbol{w}_{(1,k)} \circledast_G \boldsymbol{x}(k) = \left(\sum_{k=1}^{c_0} \boldsymbol{w}_{(1,k)}^\top \Pi_g \boldsymbol{x}(k)\right)_{g \in G} = \left(\boldsymbol{w}_1^\top \boldsymbol{x}_g\right)_{g \in G},$$

where $\boldsymbol{w}_1$ is the vectorized form of concatenated convolution vectors, and $\boldsymbol{x}_g$ is the vectorized form of concatenated $\Pi_g \boldsymbol{x}(k)$. Using this notation, we have:

$$d(h_{\boldsymbol{w}}, h_{\boldsymbol{v}}) = \left(\frac{1}{m}\sum_{i=1}^m \left(P \circ \sigma \left(\boldsymbol{w}^\top \boldsymbol{x}_{i,g}\right)_{g \in G} - P \circ \sigma \left(\boldsymbol{v}^\top \boldsymbol{x}_{i,g}\right)_{g \in G}\right)^2\right)^{1/2},$$

and using the Lipschitz continuity of max-pooling and ReLU, we get:

$$\left| P \circ \sigma \left(\boldsymbol{w}^\top \boldsymbol{x}_g\right)_{g \in G} - P \circ \sigma \left(\boldsymbol{v}_1^\top \boldsymbol{x}_g\right)_{g \in G} \right| \leq \max_{g \in G} \left|(\boldsymbol{w} - \boldsymbol{v})^\top \boldsymbol{x}_g\right|.$$

This means that it suffices to get a covering number for the linear function $(\boldsymbol{w}^\top \boldsymbol{x}_{i,g})_{i \in [m], g \in G}$ on the extended dataset. We use the following lemma from Zhang (2002), which is stronger than a similar result using Maurey's empirical lemma Pisier (1980; 1989).

**Lemma C.3.** *If $h(x) = \boldsymbol{w}^\top \boldsymbol{x}$ with $\|\boldsymbol{x}\|_p \leq b_x$ and $\|\boldsymbol{w}\|_q \leq w$ for $2 \leq p \leq \infty$ and $1/p + 1/q = 1$, then for all $u > 0$*

$$\log \mathcal{N}(\mathcal{H}, \|\cdot\|_\infty, u) \leq (36(p-1)wb_x/u)^2 \log(2(4wb_x/u + 2)m + 1).$$

*where $m$ is the number of samples.*

This lemma bounds the $\ell_\infty$-norm covering number, which is an upper bound on the covering number $\mathcal{N}(\mathcal{H}, d, u)$ used in Dudley's inequality.

In our case, the effective number of samples is $m|G|$, and therefore the covering number is bounded as:

$$\log \mathcal{N}(\mathcal{H}, d, u) \leq (36 b_x M_2/u)^2 \log(2(4 b_x M_2/u + 2)m|G| + 1).$$

We just need to plug the above covering number into Dudley's inequality and use standard inequalities:

$$4\alpha + \frac{4\sqrt{2}}{\sqrt{m}} \int_\alpha^{\frac{B_{S;\mathcal{H}}}{2}} \sqrt{\log \mathcal{N}(\mathcal{H}, d, u)} \, \mathrm{d}u \leq 4\alpha + \frac{4\sqrt{2}}{\sqrt{m}} \int_\alpha^{\frac{b_x \cdot M_2}{2}} (36 b_x M_2/u) \sqrt{\log(2(4 b_x M_2/u + 2)m|G| + 1)} . \, \mathrm{d}u$$

$$\leq 4\alpha + \frac{4\sqrt{2}}{\sqrt{m}}(36 b_x M_2) \sqrt{\log(2(4 b_x M_2/\alpha + 2)m|G| + 1)} \log(b_x M_2/2\alpha)$$

by choosing $\alpha = b_x . M_2/2\sqrt{m}$, we get the following bound:

$$\frac{2 b_x M_2}{\sqrt{m}} + \frac{4\sqrt{2}}{\sqrt{m}}(36 b_x M_2) \sqrt{\log(2(8\sqrt{m} + 2)m|G| + 1)} \log(\sqrt{m}),$$

which yields the desired result.

$\square$

Note that the bound is independent of the number of channels $c_0$, but has logarithmic dependence on the group size, as well as other constant terms. In comparison with Vardi et al. (2022), we do not have the dependence on the spectral norm of the circulant matrix, although, as we indicated before, their proof works well the norm of parameters as well. In this sense, our result can be seen as the generalization of Vardi et al. (2022) to the multi-channel input.

**Multi-Channel Output.** Next, we focus on the following setup with multi-channel convolution given by the matrix $\boldsymbol{W}$:

$$h_{\boldsymbol{u},\boldsymbol{w}}(\boldsymbol{x}) = \boldsymbol{u}^\top P \circ \sigma (\boldsymbol{W}\boldsymbol{x}).$$

We have the following theorems for this case.

**Theorem C.4** (Multiple Channel Max pooling). *Consider the hypothesis space $\mathcal{H}$ of multiple channel group convolutional network with $P(\cdot)$ as max pooling, and $\sigma(\cdot)$ as ReLU. With probability at least $1 - \delta$ and for all $h \in \mathcal{H}$, we have:*

$$\mathcal{L}(h) \leq \hat{\mathcal{L}}(h) + \frac{b_x M_1 M_2 \sqrt{c_1}}{\sqrt{m}} \left( 288\sqrt{2}\sqrt{\log(2(16\sqrt{m}+2)m|G|+1)} \log(m) \right) + 4\sqrt{\frac{2\log(4/\delta)}{m}}.$$

*Proof.* We would need a covering number for $h_{\boldsymbol{u},\boldsymbol{w}}(\boldsymbol{X}) := (h_{\boldsymbol{u},\boldsymbol{w}}(\boldsymbol{x}_1), \ldots, h_{\boldsymbol{u},\boldsymbol{w}}(\boldsymbol{x}_m))$ in $\ell_2$-norm. We do this in two steps similar to the strategy in Bartlett et al. (2017). As the first step, we find a $\delta_1$-covering $\boldsymbol{u}_k$ for the following set:

$$\mathcal{H}_1 = \{P \circ \sigma (\boldsymbol{W}\boldsymbol{X}) : \|\boldsymbol{w}\| \leq M_2\}$$

where we used the notation $\boldsymbol{W}\boldsymbol{X}$ to denote the concatenation of all points in the training set, namely:

$$\boldsymbol{W}\boldsymbol{X} = (P \circ \sigma (\boldsymbol{W}\boldsymbol{x}_1), \ldots, P \circ \sigma (\boldsymbol{W}\boldsymbol{x}_m)).$$

To simplify the notation, we assume that that matrix operations are broadcasted through $\boldsymbol{X}$ as if we have parallel compute over $\boldsymbol{x}_i$'s.

The second step consists of finding a $\delta_2$-covering for the set of $\boldsymbol{u}^\top \boldsymbol{u}_k$ for each $\boldsymbol{u}_k$'s from the first covering. Using these two coverings, We have:

$$\begin{aligned} d\left(h_{\boldsymbol{u},\boldsymbol{w}}(\boldsymbol{X}), \boldsymbol{v}_l\right) &\leq d\left(h_{\boldsymbol{u},\boldsymbol{w}}(\boldsymbol{X}), \boldsymbol{u}^\top \boldsymbol{u}_k\right) + d\left(\boldsymbol{u}^\top \boldsymbol{u}_k, \boldsymbol{v}_l\right) \\ &\leq M_2 d\left(P \circ \sigma(\boldsymbol{W}\boldsymbol{X}), \boldsymbol{u}_k\right) + d\left(\boldsymbol{u}^\top \boldsymbol{u}_k, \boldsymbol{v}_l\right) \leq M\delta_1 + \delta_2 \end{aligned} \tag{23}$$

where the norm $d$ is defined in eq. 19 for any functions defined over the dataset $(\boldsymbol{x}_1, \ldots, \boldsymbol{x}_m)$. Besides, we assumed the points $\boldsymbol{v}_l$ and $\boldsymbol{u}_k$ are chosen from the cover to satisfy the norm inequalities above. The final covering number is the product of the covering numbers from each step with the covering radius $M\delta_1 + \delta_2$.

We start with the second step, namely to cover the following set for any $\boldsymbol{u}_k$ in the first cover:

$$\mathcal{H}_{2,k} := \{\boldsymbol{u}^\top \boldsymbol{u}_k : \|\boldsymbol{u}\| \leq M_1\}.$$

The elements of the first cover, $\boldsymbol{u}_k$, are themselves instances of the first layer of the network. Therefore, we can bound the second covering number for all $k$ as follows:

$$\mathcal{N}(\mathcal{H}_{2,k}, d, \delta_2) \leq \sup_{\boldsymbol{W}} \mathcal{N}(\{\boldsymbol{u}^\top P \circ \sigma(\boldsymbol{W}\boldsymbol{X}) : \|\boldsymbol{u}\| \leq M_2\}, d, \delta_2).$$

This is an instance of covering linear functions. We will use the following result from Zhang (2002), which leverages Maurey's empirical lemma.

**Lemma C.5.** *If $h(x) = \boldsymbol{w}^\top \boldsymbol{x}$ with $\|\boldsymbol{x}\|_p \leq b_x$ and $\|\boldsymbol{w}\|_q \leq w$, then*

$$\log \mathcal{N}(\mathcal{H}, d, u) \leq (2wb_x/u)^2 \log(2m + 1).$$

*where $m$ is the number of samples.*

Since the norm of $\boldsymbol{u}$ is bounded by $M_1$, we only need to bound the norm of $\boldsymbol{u}_k$. First, note that $\|P \circ \sigma(\boldsymbol{W}\boldsymbol{X})\|^2 = \sum_{i=1}^m \|P \circ \sigma(\boldsymbol{W}\boldsymbol{x}_i)\|^2$. For each element of the sum, we can get the following upper

bound:

$$
\begin{aligned}
\|P \circ \sigma \left(\boldsymbol{W}\boldsymbol{x}_i\right)\|^2 &= \sum_{i=1}^{c_1} \left| P \circ \sigma \left( \sum_{k=1}^{c_0} \boldsymbol{w}_{(i,k)} \circledast_G \boldsymbol{x}(k) \right) \right|^2 \\
&\leq \sum_{i=1}^{c_1} \left| \max \left( \sum_{k=1}^{c_0} \boldsymbol{w}_{(i,k)} \circledast_G \boldsymbol{x}(k) \right) \right|^2 \\
&\leq \sum_{i=1}^{c_1} \left( \sum_{k=1}^{c_0} \left| \max \left( \boldsymbol{w}_{(i,k)} \circledast_G \boldsymbol{x}(k) \right) \right| \right)^2 \\
&\leq \sum_{i=1}^{c_1} \left( \sum_{k=1}^{c_0} \left\| \boldsymbol{w}_{(i,k)} \right\| \left\| \boldsymbol{x}(k) \right\| \right)^2 \\
&\leq \sum_{i=1}^{c_1} \left( \sum_{k=1}^{c_0} \left\| \boldsymbol{w}_{(i,k)} \right\|^2 \right) \left( \sum_{k=1}^{c_0} \left\| \boldsymbol{x}(k) \right\|^2 \right) \\
&\leq \|\boldsymbol{x}\|^2 \sum_{i=1}^{c_1} \sum_{k=1}^{c_0} \left\| \boldsymbol{w}_{(i,k)} \right\|^2 \leq b_x^2 M_1^2,
\end{aligned}
$$

where the first inequality follows from the property of ReLU, and the thrid inequality follows from Young's convolutional inequality. Using this inequality, we can use Lemma C.5 to get the following:

$$
\log \mathcal{N}(\mathcal{H}_{2,k}, d_0, \delta_2) \leq (2 b_x M_1 M_2 \sqrt{m} / \delta_2)^2 \log(2m + 1). \tag{24}
$$

Now we can move to the covering number for $\mathcal{H}_1 = \{P \circ \sigma \left(\boldsymbol{W}\boldsymbol{X}\right) : \|\boldsymbol{w}\| \leq M_1\}$. Given that the ReLU function is 1-Lipschitz, it suffices to find a covering for $P(\boldsymbol{W}\boldsymbol{X})$, and note that the covering norm is a mixed norm, namely:

$$
\begin{aligned}
d(P(\boldsymbol{W}\boldsymbol{X}), \boldsymbol{U}) &= \left( \frac{1}{m} \sum_{i=1}^{m} \left\| P(\boldsymbol{W}\boldsymbol{x}_i) - \boldsymbol{U}_i \right\|^2 \right)^{1/2} \\
&= \left( \frac{1}{m} \sum_{i=1}^{m} \sum_{j=1}^{c_1} \left( \max \left( \sum_{k=1}^{c_0} \boldsymbol{w}_{(j,k)} \circledast_G \boldsymbol{x}_i(k) \right) - \boldsymbol{U}_{i,j} \right)^2 \right)^{1/2}.
\end{aligned}
$$

To find the covering number, we use the following set of inequalities:

$$
\begin{aligned}
d(P(\boldsymbol{W}\boldsymbol{X}), \boldsymbol{U}) &\leq \left( \max_{i \in [m]} \sum_{j=1}^{c_1} \left( \max \left( \sum_{k=1}^{c_0} \boldsymbol{w}_{(j,k)} \circledast_G \boldsymbol{x}_i(k) \right) - \boldsymbol{U}_{i,j} \right)^2 \right)^{1/2} \\
&\leq \left( \max_{i \in [m]} \sum_{j=1}^{c_1} \left\| \boldsymbol{w}_{(j,:)} \right\|^2 \left( \max \left( \sum_{k=1}^{c_0} \frac{\boldsymbol{w}_{(j,k)}}{\left\| \boldsymbol{w}_{(j,:)} \right\|} \circledast_G \boldsymbol{x}_i(k) \right) - \frac{\boldsymbol{U}_{i,j}}{\left\| \boldsymbol{w}_{(j,:)} \right\|} \right)^2 \right)^{1/2}.
\end{aligned}
$$

Suppose that we find a cover $\tilde{\boldsymbol{U}}_{i,j}$ that satisfies the following inequality

$$
\max_{i \in [m], j \in [c_1]} \left| \max \left( \sum_{k=1}^{c_0} \frac{\boldsymbol{w}_{(j,k)}}{\left\| \boldsymbol{w}_{(j,:)} \right\|} \circledast_G \boldsymbol{x}_i(k) \right) - \tilde{\boldsymbol{U}}_{i,j} \right| \leq \frac{\delta_1}{M_1}. \tag{25}
$$

Then, it is easy to see that since $\sum_{j=1}^{c_1} \left\| \boldsymbol{w}_{(j,:)} \right\|^2 \leq M_1$, then

$$
d(P(\boldsymbol{W}\boldsymbol{X}), \boldsymbol{U}) = \left( \frac{1}{m} \sum_{i=1}^{m} \sum_{j=1}^{c_1} \left( \max \left( \sum_{k=1}^{c_0} \boldsymbol{w}_{(j,k)} \circledast_G \boldsymbol{x}_i(k) \right) - \left\| \boldsymbol{w}_{(j,:)} \right\| \tilde{\boldsymbol{U}}_{i,j} \right)^2 \right)^{1/2} \leq \delta_1.
$$

To find such covering, first find a cover for this set:

$$\left\{ \left( \max \left( \sum_{k=1}^{c_0} \tilde{\boldsymbol{w}}_{(k)} \circledast_G \boldsymbol{x}_i(k) \right) \right)_{i \in [m]} : \|\tilde{\boldsymbol{w}}\| \leq 1 \right\}.$$

Note that this is a special case of the covering number of the single-channel networks, which we had computed above. This set is the super-set of functions represented by each channel when the weights are normalized to have unit norm. We find $c_1$ independent covers, one for each channel, and this will give us the desired covering in eq. 25. We use the covering number for the single channel case using $M_2 = 1$ and $u = \delta_1/M_2$ (which basically does not change the bound on the covering number) to get the ultimate covering number for this layer given by:

$$\log \mathcal{N}(\mathcal{H}_1, d, \delta_1) \leq c_1 \mathcal{N}(\mathcal{H}, d, \delta_1/M_2) \leq c_1 (36 b_x M_2/\delta_1)^2 \log(2(4 b_x M_2/\delta_1 + 2) m|G| + 1). \tag{26}$$

The final covering number is the product of the new covering number, and what we obtained for the last layer. Therefore, using the inequality eq. 23 and the covering numbers eq. 24 and eq. 26 , the final covering number can be bounded as:

$$\begin{aligned}
\log \mathcal{N}(\mathcal{H}, d, \delta) &\leq \log \max_k \mathcal{N}(\mathcal{H}_{2,k}, d, \delta/2) + \log \mathcal{N}(\mathcal{H}_1, d, \delta/2M_1) \\
&\leq (4 b_x M_1 M_2 \sqrt{m}/\delta)^2 \log(2m+1) + c_1 (72 b_x M_1 M_2/\delta)^2 \log(2(8 b_x M_1 M_2/\delta + 2) m|G| + 1) \\
&\leq 2 c_1 (72 b_x M_1 M_2/\delta)^2 \log(2(8 b_x M_1 M_2/\delta + 2) m|G| + 1).
\end{aligned}$$

We can plug into Dudley's inequality, which gives us the following bound:

$$\begin{aligned}
4\alpha + \frac{4\sqrt{2}}{\sqrt{m}} \int_\alpha^{\frac{B_{\mathcal{S};\mathcal{H}}}{2}} \sqrt{\log \mathcal{N}(\mathcal{H}, d, u)} \, \mathrm{d}u & \\
&\leq 4\alpha + \frac{4\sqrt{2}}{\sqrt{m}} \sqrt{c_1}(72 b_x M_1 M_2) \sqrt{\log(2(8 b_x M_1 M_2/\alpha + 2) m|G| + 1)} \log(b_x M_1 M_2/2\alpha).
\end{aligned}$$

We can choose $\alpha = b_x . M_1 M_2/2\sqrt{m}$, and we get the following bound:

$$\frac{2 b_x M_1 M_2}{\sqrt{m}} + \frac{2\sqrt{2}}{\sqrt{m}} \sqrt{c_1}(72 b_x M_1 M_2) \sqrt{\log(2(16\sqrt{m} + 2) m|G| + 1)} \log(m)$$

The result follows from a standard inequality. $\qquad \square$

Apart from constant terms and other logarithmic terms, this new theorem has an extra dependence on $\sqrt{c_1}$, which is the dimension of the output channels. For the rest, we recover the term $b_x M_1 M_2/\sqrt{m}$ which is what we expected. Unfortunately, for max-pooling, the dependence on $\sqrt{c_1}$ seems to be the artifact of using the covering number argument. As we will see in the next section, the direct analysis of Rademacher complexity shows that the dependence on $c_1$ can be completely removed with the price of additional dependence on the group size and input dataset.

*Remark* C.6. The covering argument above is strictly better than the method used in Lemman 3.2 of Bartlett et al. (2017) for covering matrix products. If we use this lemma, we get a covering number of the form:

$$\log(\mathcal{N}(\mathcal{H}_1, d, u)) \leq c_1 \left( \frac{b_x M_2 |G|}{u} \right)^2 \log(2|G|^2 c_1 c_0).$$

As one can see, we have now further dimension dependence in the bound, basically on the number of input channels $c_0$ and $|G|$. In the original paper, they could use mixed norms $\|\cdot\|_{q,s}$ to get rid of some of the dimension dependencies, which we cannot do in our setting.

### C.4 Homogeneous non-decreasing activation with max pooling

The following result provides a similar upper bound for max-pooling. A bit of notation: fix the training data matrix $\boldsymbol{X}$. Denote the permutation action of the group element $l \in G$ by $\Pi_l$. The vector $\boldsymbol{l} = (l_1, \ldots, l_m) \in [|G|]^m$ determines the group permutation index individually applied to each training data points. For each $\boldsymbol{l}$, we get the group-augmented version of the dataset denoted as $\boldsymbol{X_l} = (\Pi_{l_1} \boldsymbol{x}_1 \ldots \Pi_{l_m} \boldsymbol{x}_m)$.

**Theorem C.7.** *Consider the hypothesis space $\mathcal{H}$ defined in eq. 4. Suppose that $P(\cdot)$ is the max pooling operation, and $\sigma(\cdot)$ is a 1-Lipschitz and non-decreasing positively homogeneous activation function. Fix the training data matrix $\boldsymbol{X}$. Then with probability at least $1 - \delta$ and for all $h \in \mathcal{H}$, we have:*

$$\mathcal{L}(h) \leq \hat{\mathcal{L}}(h) + 2\frac{M_1 M_2 g(\boldsymbol{X})}{\sqrt{m}} + 4\sqrt{\frac{2\log(4/\delta)}{m}},$$

*with $g(\boldsymbol{X})$ defined as:*

$$g(\boldsymbol{X}) = \sqrt{8\log(|G|)M_{2,X}^{\max} + b_x^2 + \sqrt{8\log(|G|)M_{2,X}^{\max}b_x^2}}$$

*where $M_{2,X}^{\max} = \max_{\boldsymbol{l}} \left\| \boldsymbol{X_l}^\top \boldsymbol{X_l} \right\|_{2\to2}$ for the group-augmented training data matrix $\boldsymbol{X_l}$.*

The proof for max pooling leverages new techniques and is presented in Appendix C.4. The generalization bound in the above theorem is completely dimension-free (apart from a logarithmic dependence on the group size in $g(\boldsymbol{X})$). The norm dependency is also quite minimal. The bound merely depends on the Euclidean norm of parameters per layer. Note that the norm is computed for the convolutional kernel $\boldsymbol{w}$, which is tighter than both spectral and Frobenius norm of the respective matrix $\boldsymbol{W}$. Finally, there is no dependency on the dimension or the number of input and output channels $(c_0, c_1)$. The term $M_{2,X}^{\max}$ is new compared to other results in the literature. This term essentially captures the covariance of the data in the quotient feature space. The term $M_{2,X}^{\max}$ is defined in terms of the spectral norm, which is lower-bounded by the norm of column vectors, namely $b_x$. See Appendix C.4 for more discussions on $M_{2,X}^{\max}$.

We start from eq. 20, where the first layer is peeled off. One of the key ideas behind the proof is that if $\sigma(\cdot)$ is positively homogeneous and non-decreasing, the activation $\sigma(\cdot)$ and the max operation can swap their place. We have:

$$\mathbb{E}_{\boldsymbol{\epsilon}}\left(\sup_{\boldsymbol{w}}\left\|\sum_{i=1}^m \epsilon_i P \circ \sigma(\boldsymbol{W}\boldsymbol{x}_i)\right\|\right) = \mathbb{E}_{\boldsymbol{\epsilon}}\left(\sup_{\boldsymbol{w}}\left\|\sum_{i=1}^m \epsilon_i \begin{pmatrix} \max \sigma\left(\sum_{k=1}^{c_0} \boldsymbol{w}_{(1,k)} \circledast_G \boldsymbol{x}_i(k)\right) \\ \vdots \\ \max \sigma\left(\sum_{k=1}^{c_0} \boldsymbol{w}_{(c_1,k)} \circledast_G \boldsymbol{x}_i(k)\right) \end{pmatrix}\right\|\right)$$

$$= \mathbb{E}_{\boldsymbol{\epsilon}}\left(\sup_{\boldsymbol{w}}\left\|\sum_{i=1}^m \epsilon_i \begin{pmatrix} \sigma\left(\max\sum_{k=1}^{c_0} \boldsymbol{w}_{(1,k)} \circledast_G \boldsymbol{x}_i(k)\right) \\ \vdots \\ \sigma\left(\max\sum_{k=1}^{c_0} \boldsymbol{w}_{(c_1,k)} \circledast_G \boldsymbol{x}_i(k)\right) \end{pmatrix}\right\|\right)$$

Then, similar to the previous proof, we use the following moment inequality:

$$\mathbb{E}_{\boldsymbol{\epsilon}}\left(\sup_{\boldsymbol{w}}\left\|\sum_{i=1}^m \epsilon_i \begin{pmatrix} \sigma\left(\max\sum_{k=1}^{c_0} \boldsymbol{w}_{(1,k)} \circledast_G \boldsymbol{x}_i(k)\right) \\ \vdots \\ \sigma\left(\max\sum_{k=1}^{c_0} \boldsymbol{w}_{(c_1,k)} \circledast_G \boldsymbol{x}_i(k)\right) \end{pmatrix}\right\|\right)$$

$$\leq \sqrt{\mathbb{E}_{\boldsymbol{\epsilon}}\left(\sup_{\boldsymbol{w}}\left\|\sum_{i=1}^m \epsilon_i \begin{pmatrix} \sigma\left(\max\sum_{k=1}^{c_0} \boldsymbol{w}_{(1,k)} \circledast_G \boldsymbol{x}_i(k)\right) \\ \vdots \\ \sigma\left(\max\sum_{k=1}^{c_0} \boldsymbol{w}_{(c_1,k)} \circledast_G \boldsymbol{x}_i(k)\right) \end{pmatrix}\right\|^2\right)}$$

We can then use the peeling-off technique similar to the proof above for the term inside the square root:

$$\mathbb{E}_{\boldsymbol{\epsilon}}\left(\sup_{\boldsymbol{w}}\left\|\sum_{i=1}^{m}\epsilon_i\begin{pmatrix}\sigma\left(\max\sum_{k=1}^{c_0}\boldsymbol{w}_{(1,k)}\circledast_G\boldsymbol{x}_i(k)\right)\\\vdots\\\sigma\left(\max\sum_{k=1}^{c_0}\boldsymbol{w}_{(c_1,k)}\circledast_G\boldsymbol{x}_i(k)\right)\end{pmatrix}\right\|^2\right)$$

$$=\mathbb{E}_{\boldsymbol{\epsilon}}\left(\sup_{\boldsymbol{w}}\left\|\sum_{i=1}^{m}\epsilon_i\begin{pmatrix}\sigma\left(\max_{l\in[|G|]}\sum_{k=1}^{c_0}\boldsymbol{w}_{(1,k)}^\top\Pi_l\boldsymbol{x}_i(k)\right)\\\vdots\\\sigma\left(\max_{l\in[|G|]}\sum_{k=1}^{c_0}\boldsymbol{w}_{(c_1,k)}^\top\Pi_l\boldsymbol{x}_i(k)\right)\end{pmatrix}\right\|^2\right)$$

$$=\mathbb{E}_{\boldsymbol{\epsilon}}\left(\sup_{\boldsymbol{w}}\sum_{j=1}^{c_1}\left(\sum_{i=1}^{m}\epsilon_i\sigma\left(\max_{l\in[|G|]}\sum_{k=1}^{c_0}\boldsymbol{w}_{(j,k)}^\top\Pi_l\boldsymbol{x}_i(k)\right)\right)^2\right)$$

$$\leq M_2^2\mathbb{E}_{\boldsymbol{\epsilon}}\left(\sup_{\boldsymbol{w}:\|\boldsymbol{w}\|\leq 1}\left(\sum_{i=1}^{m}\epsilon_i\sigma\left(\max_{l\in[|G|]}\sum_{k=1}^{c_0}\boldsymbol{w}_{(k)}^\top\Pi_l\boldsymbol{x}_i(k)\right)\right)^2\right)$$

To continue, we define $\boldsymbol{l}\in[|G|]^m$ as the vector of $(l_1,\dots,l_m)$ with $l_i\in[|G|]$. Now, we can first use the contraction inequality and some other standard techniques to get the following:

$$\mathbb{E}_{\boldsymbol{\epsilon}}\left(\sup_{\boldsymbol{w}:\|\boldsymbol{w}\|\leq 1}\left(\sum_{i=1}^{m}\epsilon_i\sigma\left(\max_{l\in[|G|]}\sum_{k=1}^{c_0}\boldsymbol{w}_{(k)}^\top\Pi_l\boldsymbol{x}_i(k)\right)\right)^2\right)$$

$$\leq\mathbb{E}_{\boldsymbol{\epsilon}}\left(\sup_{\boldsymbol{w}:\|\boldsymbol{w}\|\leq 1}\left(\sum_{i=1}^{m}\epsilon_i\max_{l\in[|G|]}\left(\sum_{k=1}^{c_0}\boldsymbol{w}_{(k)}^\top\Pi_l\boldsymbol{x}_i(k)\right)\right)^2\right)\qquad(27)$$

$$\leq\mathbb{E}_{\boldsymbol{\epsilon}}\left(\sup_{\boldsymbol{w}:\|\boldsymbol{w}\|\leq 1}\max_{\boldsymbol{l}}\left(\sum_{i=1}^{m}\epsilon_i\left(\sum_{k=1}^{c_0}\boldsymbol{w}_{(k)}^\top\Pi_{l_i}\boldsymbol{x}_i(k)\right)\right)^2\right)$$

$$=\mathbb{E}_{\boldsymbol{\epsilon}}\left(\sup_{\boldsymbol{w}:\|\boldsymbol{w}\|\leq 1}\max_{\boldsymbol{l}}\left(\sum_{k=1}^{c_0}\boldsymbol{w}_{(k)}^\top\left(\sum_{i=1}^{m}\epsilon_i\Pi_{l_i}\boldsymbol{x}_i(k)\right)\right)^2\right)$$

$$\leq\mathbb{E}_{\boldsymbol{\epsilon}}\left(\max_{\boldsymbol{l}}\left\|\sum_{i=1}^{m}\epsilon_i\Pi_{l_i}\boldsymbol{x}_i\right\|^2\right)$$

where we abuse the notation $\Pi_{l_i}\boldsymbol{x}_i$ to denote the channel-wise application of the permutation to $\boldsymbol{x}_i$ (assuming $\boldsymbol{x}_i$ is shaped as $|G|\times c_0$). We use the same term $\Pi_{l_i}\boldsymbol{x}_i$ to denote its vectorized version to avoid the notation overhead.

Now, see that for $\beta>0$, :

$$\beta\mathbb{E}_{\boldsymbol{\epsilon}}\left(\max_{\boldsymbol{l}}\left\|\sum_{i=1}^{m}\epsilon_i\Pi_{l_i}\boldsymbol{x}_i\right\|^2\right)=\mathbb{E}_{\boldsymbol{\epsilon}}\left(\log\exp\max_{\boldsymbol{l}}\beta\left\|\sum_{i=1}^{m}\epsilon_i\Pi_{l_i}\boldsymbol{x}_i\right\|^2\right)$$

$$\leq\log\mathbb{E}_{\boldsymbol{\epsilon}}\left(\max_{\boldsymbol{l}}\exp\beta\left\|\sum_{i=1}^{m}\epsilon_i\Pi_{l_i}\boldsymbol{x}_i\right\|^2\right)$$

$$\leq\log\mathbb{E}_{\boldsymbol{\epsilon}}\left(\sum_{\boldsymbol{l}}\exp\beta\left\|\sum_{i=1}^{m}\epsilon_i\Pi_{l_i}\boldsymbol{x}_i\right\|^2\right)$$

We can now use the Lemma B.2. Note that, in our case, we have:

$$\left\| \sum_{i=1}^{m} \epsilon_i \Pi_{l_i} \boldsymbol{x}_i \right\|^2 = \| \boldsymbol{X_l}\boldsymbol{\epsilon} \|^2 = \boldsymbol{\epsilon}^\top \boldsymbol{X_l}^\top \boldsymbol{X_l} \boldsymbol{\epsilon},$$

where $\boldsymbol{\epsilon} = (\epsilon_1, \ldots, \epsilon_m)$, $\boldsymbol{l} = (l_1, \ldots, l_m)$, and $\boldsymbol{X_l} = (\Pi_{l_1}\boldsymbol{x}_1 \ldots \Pi_{l_m}\boldsymbol{x}_m)$. We now choose $\boldsymbol{B} = \boldsymbol{X_l}^\top \boldsymbol{X_l}$ in Lemma B.2 to get:

$$\log \mathbb{E}_{\boldsymbol{\epsilon}} \left( \sum_{\boldsymbol{l}} \exp \beta \left\| \sum_{i=1}^{m} \epsilon_i \Pi_{l_i}\boldsymbol{x}_i \right\|^2 \right) \leq \log \left( \sum_{\boldsymbol{l}} \exp \left( \frac{\beta \operatorname{Tr}(\boldsymbol{X_l}^\top \boldsymbol{X_l})}{1 - 8\beta \left\| \boldsymbol{X_l}^\top \boldsymbol{X_l} \right\|_{2\to 2}} \right) \right)$$

$$= \log \left( \sum_{\boldsymbol{l}} \exp \left( \frac{\beta \operatorname{Tr}(\boldsymbol{X}^\top \boldsymbol{X})}{1 - 8\beta \left\| \boldsymbol{X_l}^\top \boldsymbol{X_l} \right\|_{2\to 2}} \right) \right)$$

$$\leq \log \left( \sum_{\boldsymbol{l}} \exp \left( \frac{\beta m b_x^2}{1 - 8\beta M_{2,X}^{\max}} \right) \right)$$

$$\leq \log \left( |G|^m \exp \left( \frac{\beta m b_x^2}{1 - 8\beta M_{2,X}^{\max}} \right) \right)$$

where $M_{2,X}^{\max} = \max_{\boldsymbol{l}} \left\| \boldsymbol{X_l}^\top \boldsymbol{X_l} \right\|_{2\to 2}$. Note that we have assumed $8\beta M_{2,X}^{\max} \leq 1$. We then have:

$$\mathbb{E}_{\boldsymbol{\epsilon}} \left( \max_{\boldsymbol{l}} \left\| \sum_{i=1}^{m} \epsilon_i \Pi_{l_i}\boldsymbol{x}_i \right\|^2 \right) \leq \frac{m \log(|G|)}{\beta} + \frac{m b_x^2}{1 - 8\beta M_{2,X}^{\max}}$$

Note that

$$\max_{\beta} \frac{a}{\beta} + \frac{b}{1 - c\beta} = ac + b + 2\sqrt{abc},$$

obtained for $\beta = 1/\left( c + \sqrt{bc/a} \right)$. We need to assume that $8\beta M_{2,X}^{\max} \leq 1$, which means:

$$8 M_{2,X}^{\max} \leq \left( 8 M_{2,X}^{\max} + \sqrt{8 M_{2,X}^{\max} b_x^2 / \log |G|} \right).$$

This is always true, so choosing $\beta$ accordingly, we have:

$$\mathbb{E}_{\boldsymbol{\epsilon}} \left( \max_{\boldsymbol{l}} \left\| \sum_{i=1}^{m} \epsilon_i \Pi_{l_i}\boldsymbol{x}_i \right\|^2 \right) \leq 8m \log(|G|) M_{2,X}^{\max} + m b_x^2 + m\sqrt{8 \log(|G|) M_{2,X}^{\max} b_x^2},$$

which gives us the total bound:

$$\frac{M_1 M_2 \sqrt{8 \log(|G|) M_{2,X}^{\max} + b_x^2 + \sqrt{8 \log(|G|) M_{2,X}^{\max} b_x^2}}}{\sqrt{m}}$$

*Remark* C.8. Ignoring the term $M_{2,X}^{\max}$, the bound is dimension-free. The term $M_{2,X}^{\max}$ depends on the data matrix's spectral norm with normalized columns. Remember that:

$$M_{2,X}^{\max} = \max_{\boldsymbol{l}} \left\| \boldsymbol{X_l}^\top \boldsymbol{X_l} \right\|_{2\to 2}.$$

The spectral norm of $\boldsymbol{X}^\top \boldsymbol{X}$ is the spectral norm of the sample covariance matrix of the data. Consider an extreme case of i.i.d data samples with the diagonal covariance matrix; the spectral norm of $\boldsymbol{X}^\top \boldsymbol{X}$ will be $\mathcal{O}(m)$ as $m \to \infty$. The situation remains the same for an arbitrary covariance matrix. In other words, this bound is loose for large samples. However, in small samples, the spectral norm can be smaller (for example, assume $\boldsymbol{X}$ is approximately an orthogonal matrix).

It is important to note that the term $M_{2,X}^{\max}$ appears for bounding $\mathbb{E}_{\epsilon}\left(\max_{\boldsymbol{l}}\|\sum_{i=1}^{m}\epsilon_i\Pi_{l_i}\boldsymbol{x}_i\|^2\right)$, which itself appeared in the step eq. 27. We can consider max-pooling as a Lipschitz pooling operation and use techniques similar to Vardi et al. (2022); Graf et al. (2022) to bound the term. However, this would incur an additional logarithmic dependence on the input dimension and $m$, but it will be tighter as $m \to \infty$. Ideally, the ultimate generalization bound should be the minimum of these two cases.

### C.5 General Pooling

In this section, we consider the case of general pooling. The proof steps are similar to the classical average pooling case, so we present it more compactly. We first peel off the last layer and use the moment inequality:

$$\mathbb{E}_{\boldsymbol{\epsilon}}\left(\sup_{\boldsymbol{w}}\left\|\sum_{i=1}^{m}\epsilon_i P \circ \sigma(\boldsymbol{W}\boldsymbol{x}_i)\right\|\right) = \mathbb{E}_{\boldsymbol{\epsilon}}\left(\sup_{\boldsymbol{w}}\left\|\sum_{i=1}^{m}\epsilon_i\begin{pmatrix}\phi\left(\frac{1}{|G|}\mathbf{1}^{\top}\sigma\left(\sum_{k=1}^{c_0}\boldsymbol{w}_{(1,k)}\circledast_G \boldsymbol{x}_i(k)\right)\right)\\\vdots\\\phi\left(\frac{1}{|G|}\mathbf{1}^{\top}\sigma\left(\sum_{k=1}^{c_0}\boldsymbol{w}_{(c_1,k)}\circledast_G \boldsymbol{x}_i(k)\right)\right)\end{pmatrix}\right\|\right)$$

$$\leq \sqrt{\mathbb{E}_{\boldsymbol{\epsilon}}\left(\sup_{\boldsymbol{w}}\left\|\sum_{i=1}^{m}\epsilon_i\begin{pmatrix}\phi\left(\frac{1}{|G|}\mathbf{1}^{\top}\sigma\left(\sum_{k=1}^{c_0}\boldsymbol{w}_{(1,k)}\circledast_G \boldsymbol{x}_i(k)\right)\right)\\\vdots\\\phi\left(\frac{1}{|G|}\mathbf{1}^{\top}\sigma\left(\sum_{k=1}^{c_0}\boldsymbol{w}_{(c_1,k)}\circledast_G \boldsymbol{x}_i(k)\right)\right)\end{pmatrix}\right\|^2\right)}.$$

The next step is to use the peeling argument and contraction inequality step by step:

$$\mathbb{E}_{\boldsymbol{\epsilon}}\left(\sup_{\boldsymbol{w}}\left\|\sum_{i=1}^{m}\epsilon_i\begin{pmatrix}\phi\left(\frac{1}{|G|}\mathbf{1}^{\top}\sigma\left(\sum_{k=1}^{c_0}\boldsymbol{w}_{(1,k)}\circledast_G \boldsymbol{x}_i(k)\right)\right)\\\vdots\\\phi\left(\frac{1}{|G|}\mathbf{1}^{\top}\sigma\left(\sum_{k=1}^{c_0}\boldsymbol{w}_{(c_1,k)}\circledast_G \boldsymbol{x}_i(k)\right)\right)\end{pmatrix}\right\|^2\right)$$

$$= \mathbb{E}_{\boldsymbol{\epsilon}}\left(\sup_{\boldsymbol{w}}\sum_{j=1}^{c_1}\left(\sum_{i=1}^{m}\epsilon_i\phi\left(\frac{1}{|G|}\mathbf{1}^{\top}\sigma\left(\sum_{k=1}^{c_0}\boldsymbol{w}_{(j,k)}\circledast_G \boldsymbol{x}_i(k)\right)\right)\right)^2\right)$$

$$= \mathbb{E}_{\boldsymbol{\epsilon}}\left(\sup_{\boldsymbol{w}}\sum_{j=1}^{c_1}\left\|\boldsymbol{w}_{(j,:)}\right\|^2\left(\sum_{i=1}^{m}\epsilon_i\phi\left(\frac{1}{|G|}\mathbf{1}^{\top}\sigma\left(\frac{1}{\|\boldsymbol{w}_{(j,:)}\|}\sum_{k=1}^{c_0}\boldsymbol{w}_{(j,k)}\circledast_G \boldsymbol{x}_i(k)\right)\right)\right)^2\right)$$

$$\leq M_2^2\mathbb{E}_{\boldsymbol{\epsilon}}\left(\sup_{\boldsymbol{w}}\left(\sum_{i=1}^{m}\epsilon_i\phi\left(\frac{1}{|G|}\mathbf{1}^{\top}\sigma\left(\frac{1}{\|\boldsymbol{w}\|}\sum_{k=1}^{c_0}\boldsymbol{w}_{(k)}\circledast_G \boldsymbol{x}_i(k)\right)\right)\right)^2\right)$$

$$\leq M_2^2\mathbb{E}_{\boldsymbol{\epsilon}}\left(\sup_{\boldsymbol{w}}\left(\sum_{i=1}^{m}\epsilon_i\frac{1}{|G|}\mathbf{1}^{\top}\sigma\left(\frac{1}{\|\boldsymbol{w}\|}\sum_{k=1}^{c_0}\boldsymbol{w}_{(k)}\circledast_G \boldsymbol{x}_i(k)\right)\right)^2\right)$$

$$\leq M_2^2\frac{1}{|G|}\sum_{l=1}^{|G|}\mathbb{E}_{\boldsymbol{\epsilon}}\left(\sup_{\boldsymbol{w}}\left(\sum_{i=1}^{m}\epsilon_i\sigma\left(\frac{1}{\|\boldsymbol{w}\|}\sum_{k=1}^{c_0}\boldsymbol{w}_{(k)}^{\top}\Pi_l\boldsymbol{x}_i(k)\right)\right)^2\right)$$

And then we can re-use what we proved in eq. 22 to get:

$$\mathbb{E}_{\boldsymbol{\epsilon}}\left(\sup_{\boldsymbol{w}}\left(\sum_{i=1}^{m}\epsilon_i\sigma\left(\frac{1}{\|\boldsymbol{w}\|}\sum_{k=1}^{c_0}\boldsymbol{w}_{(k)}^{\top}\Pi_l\boldsymbol{x}_i(k)\right)\right)^2\right)$$

$$= \mathbb{E}_{\boldsymbol{\epsilon}}\left(\sup_{\boldsymbol{w}}\left(\sum_{i=1}^{m}\epsilon_i\left(\frac{1}{\|\boldsymbol{w}\|}\sum_{k=1}^{c_0}\boldsymbol{w}_{(k)}^{\top}\Pi_l\boldsymbol{x}_i(k)\right)\right)^2\right) \leq mb_x^2.$$

Putting all of this together, we get the final theorem.

## C.6 Proof for Multi-Layer Group Convolutional Networks

In this section, we prove Theorem 4.3. We start again by simply peeling off the last layer:

$$\mathbb{E}_{\boldsymbol{\epsilon}}\left(\sup_{\boldsymbol{u},\{\boldsymbol{w}^{(l)},l\in[L]\}}\frac{1}{m}\sum_{i=1}^{m}\epsilon_i\boldsymbol{u}^{\top}P\circ\sigma\left(\boldsymbol{W}^L\boldsymbol{x}_i^{(L-1)}\right)\right)=\frac{M_1}{m}\mathbb{E}_{\boldsymbol{\epsilon}}\left(\sup_{\{\boldsymbol{w}^{(l)},l\in[L]\}}\left\|\sum_{i=1}^{m}\epsilon_iP\circ\sigma(\boldsymbol{W}\boldsymbol{x}_i^{(L-1)})\right\|\right),\quad(28)$$

where, for brevity, we use $\boldsymbol{x}^{(L-1)}$ to denote the output of the hidden layer $L-1$. We can expand the average pooling operation and use the positive homogeneity property of ReLU function to

$$\mathbb{E}_{\boldsymbol{\epsilon}}\left(\sup_{\{\boldsymbol{w}^{(l)},l\in[L]\}}\left\|\sum_{i=1}^{m}\epsilon_iP\circ\sigma(\boldsymbol{W}\boldsymbol{x}_i^{(L-1)})\right\|\right)\leq\sum_{g=1}^{|G|}\frac{1}{|G|}\mathbb{E}_{\boldsymbol{\epsilon}}\left(\sup_{\{\boldsymbol{w}^{(l)},l\in[L]\}}\left\|\sum_{i=1}^{m}\epsilon_i\begin{pmatrix}\sigma\left(\sum_{k=1}^{c_{L-1}}\boldsymbol{w}_{(1,k)}^{(L),\top}\Pi_g\boldsymbol{x}_i^{(L-1)}(k)\right)\\\vdots\\\sigma\left(\sum_{k=1}^{c_{L-1}}\boldsymbol{w}_{(c_L,k)}^{(L),\top}\Pi_g\boldsymbol{x}_i^{(L-1)}(k)\right)\end{pmatrix}\right\|\right)$$

and then, we focus on a single term and use simple Jensen to get:

$$\mathbb{E}_{\boldsymbol{\epsilon}}\left(\sup_{\{\boldsymbol{w}^{(l)},l\in[L]\}}\left\|\sum_{i=1}^{m}\epsilon_i\begin{pmatrix}\sigma\left(\sum_{k=1}^{c_{L-1}}\boldsymbol{w}_{(1,k)}^{(L),\top}\Pi_g\boldsymbol{x}_i^{(L-1)}(k)\right)\\\vdots\\\sigma\left(\sum_{k=1}^{c_{L-1}}\boldsymbol{w}_{(c_L,k)}^{(L),\top}\Pi_g\boldsymbol{x}_i^{(L-1)}(k)\right)\end{pmatrix}\right\|\right)$$
$$\leq\sqrt{\mathbb{E}_{\boldsymbol{\epsilon}}\left(\sup_{\{\boldsymbol{w}^{(l)},l\in[L]\}}\left\|\sum_{i=1}^{m}\epsilon_i\begin{pmatrix}\sigma\left(\sum_{k=1}^{c_{L-1}}\boldsymbol{w}_{(1,k)}^{(L),\top}\Pi_g\boldsymbol{x}_i^{(L-1)}(k)\right)\\\vdots\\\sigma\left(\sum_{k=1}^{c_{L-1}}\boldsymbol{w}_{(c_L,k)}^{(L),\top}\Pi_g\boldsymbol{x}_i^{(L-1)}(k)\right)\end{pmatrix}\right\|^2\right)},$$

and we can continue with a similar approach to simplify this further:

$$\mathbb{E}_{\boldsymbol{\epsilon}}\left(\sup_{\{\boldsymbol{w}^{(l)},l\in[L]\}}\left\|\sum_{i=1}^{m}\epsilon_i\begin{pmatrix}\sigma\left(\sum_{k=1}^{c_{L-1}}\boldsymbol{w}_{(1,k)}^{(L),\top}\Pi_g\boldsymbol{x}_i^{(L-1)}(k)\right)\\\vdots\\\sigma\left(\sum_{k=1}^{c_{L-1}}\boldsymbol{w}_{(c_L,k)}^{(L),\top}\Pi_g\boldsymbol{x}_i^{(L-1)}(k)\right)\end{pmatrix}\right\|^2\right)$$
$$=\mathbb{E}_{\boldsymbol{\epsilon}}\left(\sup_{\{\boldsymbol{w}^{(l)},l\in[L]\}}\sum_{j=1}^{c_L}\left\|\boldsymbol{w}_{(j,:)}^{(L)}\right\|^2\left(\sum_{i=1}^{m}\epsilon_i\sigma\left(\frac{1}{\left\|\boldsymbol{w}_{(j,:)}^{(L)}\right\|}\sum_{k=1}^{c_{L-1}}\boldsymbol{w}_{(j,k)}^{(L),\top}\Pi_g\boldsymbol{x}_i^{(L-1)}(k)\right)\right)^2\right)$$
$$\leq M_L^2\mathbb{E}_{\boldsymbol{\epsilon}}\left(\sup_{\{\boldsymbol{w}^{(l)},l\in[L-1],\boldsymbol{w}^{(\tilde{L})}:\left\|\boldsymbol{w}^{(\tilde{L})}\right\|\leq1\}}\left(\sum_{i=1}^{m}\epsilon_i\sigma\left(\sum_{k=1}^{c_{L-1}}\tilde{\boldsymbol{w}}_{(k)}^{(L),\top}\Pi_g\boldsymbol{x}_i^{(L-1)}(k)\right)\right)^2\right)$$
$$\leq M_L^2\mathbb{E}_{\boldsymbol{\epsilon}}\left(\sup_{\{\boldsymbol{w}^{(l)},l\in[L-1]\}}\left\|\sum_{i=1}^{m}\epsilon_i\left(\boldsymbol{x}_i^{(L-1)}\right)\right\|^2\right),$$

where $\Pi_g$ is a unitary matrix and was removed. Now, we have removed the last layer from the bound, and we can focus on the rest of layers.

$$
\mathbb{E}_{\boldsymbol{\epsilon}} \left( \sup_{\{\boldsymbol{w}^{(l)}, l \in [L-1]\}} \left\| \sum_{i=1}^m \epsilon_i \left( \boldsymbol{x}_i^{(L-1)} \right) \right\|^2 \right)
$$

$$
= \mathbb{E}_{\boldsymbol{\epsilon}} \left( \sup_{\{\boldsymbol{w}^{(l)}, l \in [L-1]\}} \left\| \sum_{i=1}^m \epsilon_i \left( \begin{matrix} \sigma \left( \sum_{k=1}^{c_{L-2}} \boldsymbol{w}_{(1,k)}^{(L-1),\top} \circledast \boldsymbol{x}_i^{(L-2)}(k) \right) \\ \vdots \\ \sigma \left( \sum_{k=1}^{c_{L-2}} \boldsymbol{w}_{(c_{L-1},k)}^{(L-1),\top} \circledast \boldsymbol{x}_i^{(L-2)}(k) \right) \end{matrix} \right) \right\|^2 \right)
$$

$$
= \mathbb{E}_{\boldsymbol{\epsilon}} \left( \sup_{\{\boldsymbol{w}^{(l)}, l \in [L-1]\}} \sum_{g=1}^{|G|} \sum_{j=1}^{c_{L-1}} \left\| \boldsymbol{w}_{(j,:)}^{(L-1)} \right\|^2 \left( \sum_{i=1}^m \epsilon_i \sigma \left( \frac{1}{\left\| \boldsymbol{w}_{(j,:)}^{(L-1)} \right\|} \sum_{k=1}^{c_{L-2}} \boldsymbol{w}_{(j,k)}^{(L-1),\top} \Pi_g \boldsymbol{x}_i^{(L-2)}(k) \right) \right)^2 \right)
$$

$$
\leq \sum_{g=1}^{|G|} \mathbb{E}_{\boldsymbol{\epsilon}} \left( \sup_{\{\boldsymbol{w}^{(l)}, l \in [L-1]\}} \sum_{j=1}^{c_{L-1}} \left\| \boldsymbol{w}_{(j,:)}^{(L-1)} \right\|^2 \left( \sum_{i=1}^m \epsilon_i \sigma \left( \frac{1}{\left\| \boldsymbol{w}_{(j,:)}^{(L-1)} \right\|} \sum_{k=1}^{c_{L-2}} \boldsymbol{w}_{(j,k)}^{(L-1),\top} \Pi_g \boldsymbol{x}_i^{(L-2)}(k) \right) \right)^2 \right)
$$

$$
\leq |G| M_{L-1}^2 \mathbb{E}_{\boldsymbol{\epsilon}} \left( \sup_{\{\boldsymbol{w}^{(l)}, l \in [L-2]\}} \left\| \sum_{i=1}^m \epsilon_i \left( \boldsymbol{x}_i^{(L-2)} \right) \right\|^2 \right).
$$

Note that the last step involves the very same peeling argument for each term in the sum, and we removed it. Doing this iteratively, we can peel off all the layers, and get the final result. It is worth noting that the authors Golowich et al. (2018) provide a way of converting exponential depth dependence to polynomial dependence. Using this technique we can change the depth dependence from $|G|^{L-1/2}$ to $(L-1)\log|G|$ for $L > 1$. See Section 3 of Golowich et al. (2018) for more discussions.

### C.6.1 A generalization bound for gradual pooling

In this part, we consider a version of the multi-layer network with gradual pooling. We consider the following model:

$$
\hat{h}_{\boldsymbol{u}, \{\boldsymbol{w}^{(l)}, l \in [L]\}}(\boldsymbol{x}) := \boldsymbol{u}^\top P_L \circ \sigma(\boldsymbol{W}^{(L)} \sigma(\boldsymbol{W}^{(L-1)} \dots P_1 \circ \sigma(\boldsymbol{W}^{(1)}) \boldsymbol{x} \dots), \tag{29}
$$

where the pooling operation changes the group size of each layer gradually to 1 as follows:

$$
\hat{G}_0 = G \xrightarrow{P_1(G_1 = \hat{G}_0/\hat{G}_1)} \hat{G}_1 \xrightarrow{P_2(G_2 = \hat{G}_1/\hat{G}_2)} \hat{G}_2 \longrightarrow \dots \hat{G}_{L-1} \xrightarrow{P_L(G_L = \hat{G}_{L-1}/\hat{G}_L)} \hat{G}_L = 1.
$$

In other words, the pooling layer at layer $l$ pools from the cosets $G_l := \hat{G}_{l-1}/\hat{G}_l$, for example by averaging out the elements of each left cosets. It can be seen as moving the operation from the group $\hat{G}_{l-1}$ to $\hat{G}_l$. This also means that:

$$
G = |\hat{G}_0| = |G_1||\hat{G}_1| = |G_1||G_2||\hat{G}_2| = \dots = |G_1| \times \dots \times |G_L|.
$$

Define the corresponding hypothesis space of functions with gradual pooling as follows:

$$
\mathcal{H}^{(L-g.p.)} := \left\{ \hat{h}_{\boldsymbol{u}, \{\boldsymbol{w}^{(l)}, l \in [L]\}} : \|\boldsymbol{u}\| \leq M_1, \|\boldsymbol{w}_i\| \leq M_{i+1}, i \in [L] \right\}. \tag{30}
$$

We have the following theorem for this network.

**Theorem C.9.** *Consider the hypothesis space in eq. 30 consisting of functions defined in eq. 29. With probability at least $1 - \delta$ and for all $h \in \mathcal{H}^{(L-g.p.)}$, we have:*

$$
\mathcal{L}(h) \leq \hat{\mathcal{L}}(h) + 2 \left( \prod_{l=1}^L |\hat{G}_l| \right) \frac{b_x M_1 M_2 \dots M_{L+1}}{\sqrt{m}} + 4\sqrt{\frac{2\log(4/\delta)}{m}}.
$$

*Proof.* For the rest of the proof, for simplicity, we assume that the elements of $G_l = \hat{G}_{l-1}/\hat{G}_l$ is represented by a member in $\hat{G}_{l-1}$ such that any element in $\hat{G}_{l-1}$ can be written uniquely as $g'.g$ where $g' = \hat{G}_l, g \in G_l$.

The key for the proof is the intuitive observation that at each layer, we could use the group size utilized at that layer, more formally:

$$\mathbb{E}_{\boldsymbol{\epsilon}}\left(\sup_{\{\boldsymbol{w}^{(l)}, l\in[L-1]\}}\left\|\sum_{i=1}^{m}\epsilon_i P_{L-1}\circ\sigma\left(\sum_{k=1}^{c_{L-2}}\boldsymbol{w}_{(1,k)}^{(L-1)}\circledast\boldsymbol{x}_i^{(L-2)}(k)\right)\right\|^2\right)$$

$$= \mathbb{E}_{\boldsymbol{\epsilon}}\left(\sup_{\{\boldsymbol{w}^{(l)}, l\in[L-1]\}}\sum_{g'\in G_{\hat{L}-1}}\left\|\sum_{i=1}^{m}\epsilon_i\frac{1}{G_{L-1}}\sum_{g=1}^{G_{L-1}}\sigma\left(\sum_{k=1}^{c_{L-2}}\boldsymbol{w}_{(1,k)}^{(L-1),\top}\Pi_{g'g}^{L-1}\boldsymbol{x}_i^{(L-2)}(k)\right)\right\|^2\right)$$

$$\leq \frac{1}{G_{L-1}}\mathbb{E}_{\boldsymbol{\epsilon}}\left(\sup_{\{\boldsymbol{w}^{(l)}, l\in[L-1]\}}\sum_{g'\in G_{\hat{L}-1}}\sum_{g=1}^{G_{L-1}}\left\|\sum_{i=1}^{m}\epsilon_i\sigma\left(\sum_{k=1}^{c_{L-2}}\boldsymbol{w}_{(1,k)}^{(L-1),\top}\Pi_{g'g}^{L-1}\boldsymbol{x}_i^{(L-2)}(k)\right)\right\|^2\right)$$

$$\leq \frac{1}{G_{L-1}}\mathbb{E}_{\boldsymbol{\epsilon}}\left(\sup_{\{\boldsymbol{w}^{(l)}, l\in[L-1]\}}\sum_{g\in G_{\hat{L}-2}}\left\|\sum_{i=1}^{m}\epsilon_i\sigma\left(\sum_{k=1}^{c_{L-2}}\boldsymbol{w}_{(1,k)}^{(L-1),\top}\Pi_{g}^{L-1}\boldsymbol{x}_i^{(L-2)}(k)\right)\right\|^2\right)$$

$$\leq \frac{\hat{G}_{L-2}}{G_{L-1}}\left\|\boldsymbol{w}_{(1,:)}^{(L-1)}\right\|^2\mathbb{E}_{\boldsymbol{\epsilon}}\left(\sup_{\{\boldsymbol{w}^{(l)}, l\in[L-2]\}}\left\|\sum_{i=1}^{m}\epsilon_i\left(\boldsymbol{x}_i^{(L-2)}\right)\right\|^2\right)$$

$$\leq \hat{G}_{L-1}\left\|\boldsymbol{w}_{(1,:)}^{(L-1)}\right\|^2\mathbb{E}_{\boldsymbol{\epsilon}}\left(\sup_{\{\boldsymbol{w}^{(l)}, l\in[L-2]\}}\left\|\sum_{i=1}^{m}\epsilon_i\left(\boldsymbol{x}_i^{(L-2)}\right)\right\|^2\right).$$

Using the above argument, we can revisit the proof above and see that:

$$\mathbb{E}_{\boldsymbol{\epsilon}}\left(\sup_{\{\boldsymbol{w}^{(l)}, l\in[L-1]\}}\left\|\sum_{i=1}^{m}\epsilon_i\boldsymbol{x}_i^{(L-1)}\right\|^2\right)$$

$$= \mathbb{E}_{\boldsymbol{\epsilon}}\left(\sup_{\{\boldsymbol{w}^{(l)}, l\in[L-1]\}}\left\|\sum_{i=1}^{m}\epsilon_i\begin{pmatrix}P_{L-1}\circ\sigma\left(\sum_{k=1}^{c_{L-2}}\boldsymbol{w}_{(1,k)}^{(L-1),\top}\circledast\boldsymbol{x}_i^{(L-2)}(k)\right)\\\vdots\\P_{L-1}\circ\sigma\left(\sum_{k=1}^{c_{L-2}}\boldsymbol{w}_{(c_{L-1},k)}^{(L-1),\top}\circledast\boldsymbol{x}_i^{(L-2)}(k)\right)\end{pmatrix}\right\|^2\right)$$

$$\leq |\hat{G}_{L-1}|M_{L-1}^2\mathbb{E}_{\boldsymbol{\epsilon}}\left(\sup_{\{\boldsymbol{w}^{(l)}, l\in[L-2]\}}\left\|\sum_{i=1}^{m}\epsilon_i\left(\boldsymbol{x}_i^{(L-2)}\right)\right\|^2\right).$$

We omitted many steps in the proof as they are exactly similar to the proof above. $\square$

*Remark* C.10. We would like to highlight that the gradual pooling breaks the equivariance with respect to the original $G$, and it is not a common practice in geometric deep learning to do it, in contrast with typical convolutional neural networks.

## D    General Equivariant Networks

The general equivariant networks are defined in Fourier space as follows:

$$\mathcal{H}_{\hat{\boldsymbol{u}},\hat{\boldsymbol{w}}} := \left\{\hat{\boldsymbol{u}}^{\top}\boldsymbol{Q}_2\sigma(\boldsymbol{Q}_1\hat{\boldsymbol{W}}\hat{\boldsymbol{x}})\right\}.$$

Here, we assumed that the input is already represented in the Fourier space $\hat{\boldsymbol{x}}$. The input and hidden layer representations are the direct sums of the group irreps $\psi$ each, respectively, with the multiplicity $m_{0,\psi}$

and $m_{1,\psi}$. Since we are working with the point-wise non-linearity $\sigma$ in the *spatial domain*, two unitary transformations $\boldsymbol{Q}_1$ and $\boldsymbol{Q}_2$ are applied as Fourier transforms from the irrep space to the spatial domain. Finally, to get an invariant function, the vector $\hat{\boldsymbol{u}}$ only mixes the frequencies of the trivial representation $\psi_0$, and it is zero otherwise. To use an analogy with group convolutional networks, $\boldsymbol{Q}_1$ and $\boldsymbol{Q}_2$ are the Fourier matrices, and $\hat{\boldsymbol{u}}$ is a combination of the pooling, which projects into the trivial representation of the group, and the last aggregation step. The hypothesis space is represented as

$$\mathcal{H} := \left\{ \hat{\boldsymbol{u}}^\top \boldsymbol{Q}_2 \sigma \left( \boldsymbol{Q}_1 \bigoplus_\psi \bigoplus_{i=1}^{m_{1,\psi}} \sum_{j=1}^{m_{0,\psi}} \hat{\boldsymbol{W}}(\psi, i, j) \hat{\boldsymbol{x}}(\psi, j) \right) \right\}.$$

We start the proof as follows:

$$\mathbb{E}_{\boldsymbol{\epsilon}} \left( \sup_{\hat{\boldsymbol{u}}, \hat{\boldsymbol{W}}} \frac{1}{m} \sum_{i=1}^m \epsilon_i \hat{\boldsymbol{u}}^\top \boldsymbol{Q}_2 \sigma(\boldsymbol{Q}_1 \hat{\boldsymbol{W}} \hat{\boldsymbol{x}}_i) \right) \leq \frac{M_1}{m} \mathbb{E}_{\boldsymbol{\epsilon}} \left( \sup_{\hat{\boldsymbol{W}}} \left\| \sum_{i=1}^m \epsilon_i \sigma(\boldsymbol{Q}_1 \hat{\boldsymbol{W}} \hat{\boldsymbol{x}}_i) \right\| \right).$$

$$\leq \frac{M_1}{m} \left( \mathbb{E}_{\boldsymbol{\epsilon}} \left( \sup_{\hat{\boldsymbol{W}}} \left\| \sum_{i=1}^m \epsilon_i \sigma(\boldsymbol{Q}_1 \hat{\boldsymbol{W}} \hat{\boldsymbol{x}}_i) \right\|^2 \right) \right)^{1/2}$$

From this step, we can re-use the pooling operation:

$$\mathbb{E}_{\boldsymbol{\epsilon}} \left( \sup_{\hat{\boldsymbol{W}}} \left\| \sum_{i=1}^m \epsilon_i \sigma(\boldsymbol{Q}_1 \hat{\boldsymbol{W}} \hat{\boldsymbol{x}}_i) \right\|^2 \right)$$

$$= \mathbb{E}_{\boldsymbol{\epsilon}} \left( \sup_{\hat{\boldsymbol{W}}} \sum_j \left( \sum_{i=1}^m \epsilon_i \sigma \left( \boldsymbol{q}_{1,j}^\top \hat{\boldsymbol{W}} \hat{\boldsymbol{x}}_i \right) \right)^2 \right)$$

$$= \mathbb{E}_{\boldsymbol{\epsilon}} \left( \sup_{\hat{\boldsymbol{W}}} \sum_j \left\| \boldsymbol{q}_{1,j}^\top \hat{\boldsymbol{W}} \right\|^2 \left( \sum_{i=1}^m \epsilon_i \sigma \left( \frac{1}{\left\| \boldsymbol{q}_{1,j}^\top \hat{\boldsymbol{W}} \right\|} \boldsymbol{q}_{1,j}^\top \hat{\boldsymbol{W}} \hat{\boldsymbol{x}}_i \right) \right)^2 \right)$$

$$\leq \mathbb{E}_{\boldsymbol{\epsilon}} \left( \sup_{\hat{\boldsymbol{W}}} \sum_j \left\| \boldsymbol{q}_{1,j}^\top \hat{\boldsymbol{W}} \right\|^2 \sup_{\hat{\boldsymbol{W}}, j} \left( \sum_{i=1}^m \epsilon_i \sigma \left( \frac{1}{\left\| \boldsymbol{q}_{1,j}^\top \hat{\boldsymbol{W}} \right\|} \boldsymbol{q}_{1,j}^\top \hat{\boldsymbol{W}} \hat{\boldsymbol{x}}_i \right) \right)^2 \right)$$

$$\leq M_2^2 \mathbb{E}_{\boldsymbol{\epsilon}} \left( \sup_{\hat{\boldsymbol{W}}, j} \left( \sum_{i=1}^m \epsilon_i \sigma \left( \frac{1}{\left\| \boldsymbol{q}_{1,j}^\top \hat{\boldsymbol{W}} \right\|} \boldsymbol{q}_{1,j}^\top \hat{\boldsymbol{W}} \hat{\boldsymbol{x}}_i \right) \right)^2 \right)$$

Note that we used the following:

$$\sum_{j=1}^{c_1} \left\| \boldsymbol{q}_{1,j}^\top \hat{\boldsymbol{W}} \right\|^2 = \left\| \boldsymbol{Q}_1 \hat{\boldsymbol{W}} \right\|^2 = \left\| \hat{\boldsymbol{W}} \right\|^2 \leq M_2^2.$$

Then, using the contraction lemma, we obtain:

$$\mathbb{E}_{\boldsymbol{\epsilon}}\left(\sup_{\hat{\boldsymbol{W}},j}\left(\sum_{i=1}^{m}\epsilon_i\sigma\left(\frac{1}{\left\|\boldsymbol{q}_{1,j}^{\top}\hat{\boldsymbol{W}}\right\|}\boldsymbol{q}_{1,j}^{\top}\hat{\boldsymbol{W}}\hat{\boldsymbol{x}}_i\right)\right)^2\right)$$

$$\leq \mathbb{E}_{\boldsymbol{\epsilon}}\left(\sup_{\hat{\boldsymbol{W}},j}\left(\sum_{i=1}^{m}\epsilon_i\left(\frac{1}{\left\|\boldsymbol{q}_{1,j}^{\top}\hat{\boldsymbol{W}}\right\|}\boldsymbol{q}_{1,j}^{\top}\hat{\boldsymbol{W}}\hat{\boldsymbol{x}}_i\right)\right)^2\right)$$

$$= \mathbb{E}_{\boldsymbol{\epsilon}}\left(\sup_{\hat{\boldsymbol{W}},j}\left(\frac{1}{\left\|\boldsymbol{q}_{1,j}^{\top}\hat{\boldsymbol{W}}\right\|}\boldsymbol{q}_{1,j}^{\top}\hat{\boldsymbol{W}}\left(\sum_{i=1}^{m}\epsilon_i\hat{\boldsymbol{x}}_i\right)\right)^2\right)$$

$$\leq \mathbb{E}_{\boldsymbol{\epsilon}}\left(\left\|\sum_{i=1}^{m}\epsilon_i\hat{\boldsymbol{x}}_i\right\|^2\right) \leq mb_x^2.$$

## E   Lower Bound on Rademacher Complexity - Proof of Theorem 4.4

We now provide a lower bound on the Rademacher complexity for average pooling and ReLU activation function. The authors in Bartlett et al. (2017) and Golowich et al. (2018) provide lower bounds on the Rademacher complexity. Their results, however, do not apply for group equivariant architectures given the underlying structure of weight kernels. Similar to Golowich et al. (2018), our result holds for a class of data distributions.

As the starting point, we can again peel off of the last layer:

$$\mathbb{E}_{\boldsymbol{\epsilon}}\left(\sup_{\boldsymbol{u},\boldsymbol{w}}\frac{1}{m}\sum_{i=1}^{m}\epsilon_i\boldsymbol{u}^{\top}P\circ\sigma\left(\boldsymbol{W}\boldsymbol{x}\right)\right) = \frac{M_1}{m}\mathbb{E}_{\boldsymbol{\epsilon}}\left(\sup_{\boldsymbol{w}}\left\|\sum_{i=1}^{m}\epsilon_i P\circ\sigma(\boldsymbol{W}\boldsymbol{x}_i)\right\|\right). \tag{31}$$

The next step is to find a lower bound on the right hand side. For that, we consider only weight kernels that are non-zero in the first channel and zero otherwise. Plugging in the average pooling, this means:

$$\mathbb{E}_{\boldsymbol{\epsilon}}\left(\sup_{\boldsymbol{w}}\left\|\sum_{i=1}^{m}\epsilon_i P\circ\sigma(\boldsymbol{W}\boldsymbol{x}_i)\right\|\right) \geq \mathbb{E}_{\boldsymbol{\epsilon}}\left(\sup_{\boldsymbol{w}\in\mathcal{W}_1}\left\|\sum_{i=1}^{m}\epsilon_i\begin{pmatrix}\frac{1}{|G|}\mathbf{1}^{\top}\sigma\left(\sum_{k=1}^{c_0}\boldsymbol{w}_{(1,k)}\circledast_G\boldsymbol{x}_i(k)\right)\\ \mathbf{0}\\ \vdots\\ \mathbf{0}\end{pmatrix}\right\|\right) \tag{32}$$

where $\mathcal{W}_1$ is the set of weight kernels with zero channels everywhere except the first channel. This can be further simplified to:

$$\mathbb{E}_{\boldsymbol{\epsilon}}\left(\sup_{\boldsymbol{w}\in\mathcal{W}_1}\left\|\sum_{i=1}^{m}\epsilon_i\begin{pmatrix}\frac{1}{|G|}\mathbf{1}^{\top}\sigma\left(\sum_{k=1}^{c_0}\boldsymbol{w}_{(1,k)}\circledast_G\boldsymbol{x}_i(k)\right)\\ \mathbf{0}\\ \vdots\\ \mathbf{0}\end{pmatrix}\right\|\right)$$

$$= \mathbb{E}_{\boldsymbol{\epsilon}}\left(\sup_{\boldsymbol{w}\in\mathcal{W}_1}\left\|\sum_{i=1}^{m}\epsilon_i\sum_{l=1}^{|G|}\frac{1}{|G|}\begin{pmatrix}\sigma\left(\sum_{k=1}^{c_0}\boldsymbol{w}_{(1,k)}^{\top}\Pi_l\boldsymbol{x}_i(k)\right)\\ \mathbf{0}\\ \vdots\\ \mathbf{0}\end{pmatrix}\right\|\right)$$

$$= \mathbb{E}_{\boldsymbol{\epsilon}}\left(\sup_{\boldsymbol{w}\in\mathcal{W}_1}\left|\sum_{i=1}^{m}\epsilon_i\sum_{l=1}^{|G|}\frac{1}{|G|}\sigma\left(\sum_{k=1}^{c_0}\boldsymbol{w}_{(1,k)}^{\top}\Pi_l\boldsymbol{x}_i(k)\right)\right|\right)$$

We assume data distributions over $\boldsymbol{x}$ such that each channel $\boldsymbol{x}(k)$ is supported in a single orthant. Without loss of generality assume that $\boldsymbol{x}(k)$'s are supported over the positive orthant, which means that $\mathbf{1}^\top \boldsymbol{x}(k) = \|\boldsymbol{x}(k)\|_1$. We also assume that the data points have maximum norm $\|\boldsymbol{x}_i\| = B$. Now consider a subset $\hat{\mathcal{W}}_1^+$ of $\mathcal{W}_1$ such that $\boldsymbol{w}_{(1,k)} = w_{(1,k)}\mathbf{1}$, where $w_{(1,k)}$ is a positive scalar value. With all these assumptions, we get:

$$
\mathbb{E}_{\boldsymbol{\epsilon}}\left(\sup_{\boldsymbol{w}\in\mathcal{W}_1}\left|\sum_{i=1}^{m}\epsilon_i\sum_{l=1}^{|G|}\frac{1}{|G|}\sigma\left(\sum_{k=1}^{c_0}\boldsymbol{w}_{(1,k)}^\top\Pi_l\boldsymbol{x}_i(k)\right)\right|\right) \geq \mathbb{E}_{\boldsymbol{\epsilon}}\left(\sup_{\boldsymbol{w}\in\hat{\mathcal{W}}_1^+}\left|\sum_{i=1}^{m}\epsilon_i\sum_{l=1}^{|G|}\frac{1}{|G|}\sigma\left(\sum_{k=1}^{c_0}\boldsymbol{w}_{(1,k)}^\top\Pi_l\boldsymbol{x}_i(k)\right)\right|\right)
$$

$$
= \mathbb{E}_{\boldsymbol{\epsilon}}\left(\sup_{\boldsymbol{w}\in\hat{\mathcal{W}}_1^+}\left|\sum_{i=1}^{m}\epsilon_i\sum_{l=1}^{|G|}\frac{1}{|G|}\sigma\left(\sum_{k=1}^{c_0}w_{(1,k)}\mathbf{1}^\top\Pi_l\boldsymbol{x}_i(k)\right)\right|\right)
$$

$$
= \mathbb{E}_{\boldsymbol{\epsilon}}\left(\sup_{\boldsymbol{w}\in\hat{\mathcal{W}}_1^+}\left|\sum_{i=1}^{m}\epsilon_i\sum_{l=1}^{|G|}\frac{1}{|G|}\sigma\left(\sum_{k=1}^{c_0}w_{(1,k)}\mathbf{1}^\top\boldsymbol{x}_i(k)\right)\right|\right)
$$

$$
= \mathbb{E}_{\boldsymbol{\epsilon}}\left(\sup_{\boldsymbol{w}\in\hat{\mathcal{W}}_1^+}\left|\sum_{i=1}^{m}\epsilon_i\left(\sum_{k=1}^{c_0}w_{(1,k)}\mathbf{1}^\top\boldsymbol{x}_i(k)\right)\right|\right),
$$

where the last step follows from the positiveness of $w_{(1,k)}$'s and the assumption on the data distribution. We can now simplify the last bound further to get:

$$
\mathbb{E}_{\boldsymbol{\epsilon}}\left(\sup_{\boldsymbol{w}\in\hat{\mathcal{W}}_1^+}\left|\sum_{i=1}^{m}\epsilon_i\left(\sum_{k=1}^{c_0}w_{(1,k)}\mathbf{1}^\top\boldsymbol{x}_i(k)\right)\right|\right) = \mathbb{E}_{\boldsymbol{\epsilon}}\left(\sup_{\boldsymbol{w}\in\hat{\mathcal{W}}_1^+}\left|\sum_{k=1}^{c_0}w_{(1,k)}\left(\sum_{i=1}^{m}\epsilon_i\mathbf{1}^\top\boldsymbol{x}_i(k)\right)\right|\right)
$$

$$
= M_2\mathbb{E}_{\boldsymbol{\epsilon}}\left(\left\|\sum_{i=1}^{m}\epsilon_i\begin{pmatrix}\mathbf{1}^\top\boldsymbol{x}_i(1)\\ \vdots\\ \mathbf{1}^\top\boldsymbol{x}_i(c_0)\end{pmatrix}\right\|\right).
$$

Using Khintchine's inequality, we can conclude that there is a constant $c > 0$ such that

$$
\mathbb{E}_{\boldsymbol{\epsilon}}\left(\left\|\sum_{i=1}^{m}\epsilon_i\begin{pmatrix}\mathbf{1}^\top\boldsymbol{x}_i(1)\\ \vdots\\ \mathbf{1}^\top\boldsymbol{x}_i(c_0)\end{pmatrix}\right\|\right) \geq c\sqrt{\sum_{i=1}^{m}\left\|\begin{pmatrix}\mathbf{1}^\top\boldsymbol{x}_i(1)\\ \vdots\\ \mathbf{1}^\top\boldsymbol{x}_i(c_0)\end{pmatrix}\right\|_2^2}
$$

$$
= c\sqrt{\sum_{i=1}^{m}\sum_{k=1}^{c_0}(\mathbf{1}^\top\boldsymbol{x}_i(k))^2}
$$

$$
= c\sqrt{\sum_{i=1}^{m}\sum_{k=1}^{c_0}\|\boldsymbol{x}_i(k)\|_1^2}
$$

$$
\geq c\sqrt{\sum_{i=1}^{m}\sum_{k=1}^{c_0}\|\boldsymbol{x}_i(k)\|_2^2} = c\sqrt{\sum_{i=1}^{m}\|\boldsymbol{x}_i\|_2^2} = B\sqrt{m}.
$$

For the last steps, we have used the assumptions on the data distribution. This yields the intended lower bound.

*Remark* E.1. The lower bounds on the sample complexity are commonly obtained via fat-shattering dimension as in Vardi et al. (2022). The construction of input-label samples shattered by non-equivariant networks would not extend to equivariant networks (ENs), as the ENs can only shatter data points that satisfy the exact symmetry. There are works on VC dimension of ENs, however they are not dimension-free and do not include norm bounds similar to ours.

## F  Frequency Domain Analysis

Let's consider the representation in the frequency domain. The Fourier transform is given by a unitary matrix $\boldsymbol{F}$. For simplicity, here we assume the simplest setting of a *commutative compact group $G$*, for which we have:

$$\boldsymbol{F}(\boldsymbol{w} \circledast_G \boldsymbol{x}) = \mathrm{diag}(\hat{\boldsymbol{w}})\hat{\boldsymbol{x}},$$

where $\hat{\boldsymbol{x}} = \boldsymbol{F}\boldsymbol{x}$, and $\mathrm{diag}(\hat{\boldsymbol{w}})$ arises from the Fourier based decomposition of the circulant matrix $\boldsymbol{W}$. Each frequency component in $\hat{\boldsymbol{w}}$ is an irreducible representation of the group $G$ denoted by $\psi$. Commutativity implies that each irrep has a multiplicity equal to 1 in the Fourier transform. Therefore, using the direct sum notation, we have:

$$\mathrm{diag}(\hat{\boldsymbol{w}}) = \bigoplus_{\psi} \hat{w}(\psi), \tag{33}$$

See the Supplementary Materials A for more details. Note that the point-wise non-linearity should be applied in the spatial domain, so we need an inverse Fourier transform before activation. The network representation in the Fourier space is given by:

$$h_{\boldsymbol{u},\boldsymbol{w}}(\boldsymbol{x}) := \boldsymbol{u}^\top P \circ \sigma \begin{pmatrix} \boldsymbol{F}^* \sum_{i=1}^{c_0} \left( \bigoplus_\psi \hat{w}_{(1,i)}(\psi) \right) \hat{\boldsymbol{x}(i)}) \\ \vdots \\ \boldsymbol{F}^* \sum_{i=1}^{c_0} \left( \bigoplus_\psi \hat{w}_{(c_1,i)}(\psi) \right) \hat{\boldsymbol{x}(i)}) \end{pmatrix} \tag{34}$$

where $\boldsymbol{F}^*$ denotes the conjugate transpose of the unitary matrix $\boldsymbol{F}$. Note that we have not touched the last layers. The last layer merely focuses on getting an invariant representation.

We show that conducting the Rademacher analysis in the frequency domain brings no additional gain. The situation may be different if the input is bandlimited. We summarize this result in the following proposition.

**Proposition F.1.** For the hypothesis space $\mathcal{H}$ defined in eq. 4, the average pooling operation, $\sigma(\cdot)$ as a 1-Lipschitz positively homogeneous activation function, the generalization error is bounded as $\mathcal{O}(\frac{b_x M_1 M_2}{\sqrt{m}})$.

*Proof.* Consider the network represented in the frequency domain as follows:

$$h_{\boldsymbol{u},\boldsymbol{w}}(\boldsymbol{x}) := \boldsymbol{u}^\top P \circ \sigma \begin{pmatrix} \boldsymbol{F}^* \sum_{i=1}^{c_0} \left( \bigoplus_\psi \hat{w}_{(1,i)}(\psi) \right) \hat{\boldsymbol{x}(i)}) \\ \vdots \\ \boldsymbol{F}^* \sum_{i=1}^{c_0} \left( \bigoplus_\psi \hat{w}_{(c_1,i)}(\psi) \right) \hat{\boldsymbol{x}(i)}) \end{pmatrix} \tag{35}$$

A few clarifications before continuing further. Consider the circular convolution $\boldsymbol{w} \circledast_G \boldsymbol{x}$. Note that the decomposition $\left( \bigoplus_\psi \hat{w}(\psi) \right)$ is obtained by using the Fourier decomposition of the equivalent circulant matrix:

$$\boldsymbol{F}^* \left( \bigoplus_\psi \hat{w}(\psi) \right) \boldsymbol{F} = \boldsymbol{W},$$

and therefore, when using Parseval's relation, we should be careful to include the group size as follows:

$$\sum_\psi \hat{w}(\psi)^2 = \|\boldsymbol{W}\|_F = |G| \, \|\boldsymbol{w}\|^2 .$$

Now, let's continue the proof. For average pooling, we can start by peeling off the last linear layers and average pooling and continue from there, namely from eq. 21. Note that $l$'th entry can be computed using an

inner product with the $l$'th row of $\boldsymbol{F}^*$, represented by $\boldsymbol{f}_l^*$. We have:

$$\mathbb{E}_{\boldsymbol{\epsilon}}\left(\sup_{\boldsymbol{w}}\left\|\sum_{i=1}^m \epsilon_i \begin{pmatrix} \sigma\left(\boldsymbol{f}_l^* \sum_{k=1}^{c_0}\left(\bigoplus_\psi \hat{w}_{(1,k)}(\psi)\right)\boldsymbol{x}_i\hat{(}k))\right) \\ \vdots \\ \sigma\left(\boldsymbol{f}_l^* \sum_{k=1}^{c_0}\left(\bigoplus_\psi \hat{w}_{(c_1,k)}(\psi)\right)\boldsymbol{x}_i\hat{(}k))\right) \end{pmatrix}\right\|^2\right)$$

$$= \mathbb{E}_{\boldsymbol{\epsilon}}\left(\sup_{\boldsymbol{w}} \sum_{j=1}^{c_1}\left(\sum_{i=1}^m \epsilon_i \sigma\left(\boldsymbol{f}_l^* \sum_{k=1}^{c_0}\left(\bigoplus_\psi \hat{w}_{(j,k)}(\psi)\right)\boldsymbol{x}_i\hat{(}k))\right)\right)^2\right)$$

$$= \mathbb{E}_{\boldsymbol{\epsilon}}\left(\sup_{\boldsymbol{w}} \sum_{j=1}^{c_1}\left\|\hat{\boldsymbol{w}}_{(j,:)}\right\|^2\left(\sum_{i=1}^m \epsilon_i \sigma\left(\frac{1}{\left\|\hat{\boldsymbol{w}}_{(j,:)}\right\|}\boldsymbol{f}_l^* \sum_{k=1}^{c_0}\left(\bigoplus_\psi \hat{w}_{(j,k)}(\psi)\right)\boldsymbol{x}_i\hat{(}k))\right)\right)^2\right),$$

where we used the positively homogeneity property of the activation function, and we defined:

$$\left\|\hat{\boldsymbol{w}}_{(j,:)}\right\|^2 = \sum_{\psi,k\in[c_0]} \hat{w}_{(j,k)}(\psi)^2.$$

We can use Parseval's theorem, and based on the discussion above, we know that:

$$\sum_{j\in[c_1]}\left\|\hat{\boldsymbol{w}}_{(j,:)}\right\|^2 = |G|\sum_{j\in[c_1]}\left\|\boldsymbol{w}_{(j,:)}\right\|^2 \le |G|M_2^2.$$

And then, we can continue similarly to the proof of average pooling:

$$\mathbb{E}_{\boldsymbol{\epsilon}}\left(\sup_{\boldsymbol{w}} \sum_{j=1}^{c_1}\left\|\hat{\boldsymbol{w}}_{(j,:)}\right\|^2\left(\sum_{i=1}^m \epsilon_i \sigma\left(\frac{1}{\left\|\hat{\boldsymbol{w}}_{(j,:)}\right\|}\boldsymbol{f}_l^* \sum_{k=1}^{c_0}\left(\bigoplus_\psi \hat{w}_{(j,k)}(\psi)\right)\boldsymbol{x}_i\hat{(}k))\right)\right)^2\right)$$

$$\le |G|M_2^2 \mathbb{E}_{\boldsymbol{\epsilon}}\left(\sup_{\hat{\boldsymbol{w}}:\|\hat{\boldsymbol{w}}\|\le 1}\left(\sum_{i=1}^m \epsilon_i \sigma\left(\boldsymbol{f}_l^* \sum_{k=1}^{c_0}\left(\bigoplus_\psi \hat{w}_{(k)}(\psi)\right)\boldsymbol{x}_i\hat{(}k))\right)\right)^2\right)$$

From which we can continue using contraction inequality:

$$\mathbb{E}_{\boldsymbol{\epsilon}}\left(\sup_{\hat{\boldsymbol{w}}:\|\hat{\boldsymbol{w}}\|\le 1}\left(\sum_{i=1}^m \epsilon_i \sigma\left(\boldsymbol{f}_l^* \sum_{k=1}^{c_0}\left(\bigoplus_\psi \hat{w}_{(k)}(\psi)\right)\boldsymbol{x}_i\hat{(}k))\right)\right)^2\right)$$

$$\le \mathbb{E}_{\boldsymbol{\epsilon}}\left(\sup_{\hat{\boldsymbol{w}}:\|\hat{\boldsymbol{w}}\|\le 1}\left(\sum_{i=1}^m \epsilon_i \left(\boldsymbol{f}_l^* \sum_{k=1}^{c_0}\left(\bigoplus_\psi \hat{w}_{(k)}(\psi)\right)\boldsymbol{x}_i\hat{(}k))\right)\right)^2\right)$$

$$\le \mathbb{E}_{\boldsymbol{\epsilon}}\left(\sup_{\hat{\boldsymbol{w}}:\|\hat{\boldsymbol{w}}\|\le 1}\left(\boldsymbol{f}_l^* \sum_{k=1}^{c_0}\left(\bigoplus_\psi \hat{w}_{(k)}(\psi)\right)\left(\sum_{i=1}^m \epsilon_i\boldsymbol{x}_i\hat{(}k))\right)\right)^2\right)$$

$$\le \sup_{\hat{\boldsymbol{w}}:\|\hat{\boldsymbol{w}}\|\le 1}\left\|\left(\boldsymbol{f}_l^*\left(\bigoplus_\psi \hat{w}_{(k)}(\psi)\right)\right)_{k\in[c_0]}\right\|_F^2 \mathbb{E}_{\boldsymbol{\epsilon}}\left(\left\|\sum_{i=1}^m \epsilon_i\hat{\boldsymbol{x}}_i\right\|^2\right)$$

$$\le \sup_{\hat{\boldsymbol{w}}:\|\hat{\boldsymbol{w}}\|\le 1}\left\|\left(\boldsymbol{f}_l^*\left(\bigoplus_\psi \hat{w}_{(k)}(\psi)\right)\right)_{k\in[c_0]}\right\|_F^2 mb_x^2.$$

To simplify the norm on the left-hand side, it is important to note that each entry of $\boldsymbol{f}_l^*$ has the modulus $1/\sqrt{|G|}$, which means that

$$\sup_{\hat{\boldsymbol{w}}:\|\hat{\boldsymbol{w}}\|\leq 1}\left\|\left(\boldsymbol{f}_l^*\left(\bigoplus_{\psi}\hat{w}_{(k)}(\psi)\right)\right)_{k\in[c_0]}\right\|_F^2\leq\frac{1}{|G|}.$$

The term $1/|G|$ cancels the term $|G|$ in the previous inequality and yields the bound. The result shows that there is no gain in the frequency domain analysis. $\qquad\square$

## G  Proofs for Weight Sharing

Consider the network:

$$h_{\boldsymbol{u},\boldsymbol{w}}(\boldsymbol{x}):=\boldsymbol{u}^\top P\circ\sigma\begin{pmatrix}\sum_{c=1}^{c_0}\left(\sum_{k=1}^{|G|}\boldsymbol{w}_{(1,c)}(k)\boldsymbol{B}_k\right)\boldsymbol{x}(c)\\\vdots\\\sum_{i=1}^{c_0}\left(\sum_{k=1}^{|G|}\boldsymbol{w}_{(c_1,c)}(k)\boldsymbol{B}_k\right)\boldsymbol{x}(c)\end{pmatrix}.$$

For the pooling operation $P(\cdot)$, we consider the average pooling operation. The Rademacher complexity analysis starts similarly to the proof of group convolution networks. This means that we can peel off the first layer $\boldsymbol{u}$ and consider a single term in the average pooling operation, namely:

$$\mathbb{E}_{\boldsymbol{\epsilon}}\left(\sup_{\boldsymbol{w}}\left\|\sum_{i=1}^m\epsilon_i P\circ\sigma\begin{pmatrix}\sum_{c=1}^{c_0}\left(\sum_{k=1}^{|G|}\boldsymbol{w}_{(1,c)}(k)\boldsymbol{B}_k\right)\boldsymbol{x}_i(c)\\\vdots\\\sum_{c=1}^{c_0}\left(\sum_{k=1}^{|G|}\boldsymbol{w}_{(c_1,c)}(k)\boldsymbol{B}_k\right)\boldsymbol{x}_i(c)\end{pmatrix}\right\|\right)$$

$$=\mathbb{E}_{\boldsymbol{\epsilon}}\left(\frac{1}{|G|}\sup_{\boldsymbol{w}}\left\|\sum_{l=1}^{|G|}\sum_{i=1}^m\epsilon_i\sigma\begin{pmatrix}\sum_{c=1}^{c_0}\left(\sum_{k=1}^{|G|}\boldsymbol{w}_{(1,c)}(k)\boldsymbol{b}_{k,l}^\top\right)\boldsymbol{x}_i(c)\\\vdots\\\sum_{c=1}^{c_0}\left(\sum_{k=1}^{|G|}\boldsymbol{w}_{(c_1,c)}(k)\boldsymbol{b}_{k,l}^\top\right)\boldsymbol{x}_i(c)\end{pmatrix}\right\|\right)$$

$$\leq\frac{1}{|G|}\sum_{l=1}^{|G|}\mathbb{E}_{\boldsymbol{\epsilon}}\left(\sup_{\boldsymbol{w}}\left\|\sum_{i=1}^m\epsilon_i\sigma\begin{pmatrix}\sum_{c=1}^{c_0}\left(\sum_{k=1}^{|G|}\boldsymbol{w}_{(1,c)}(k)\boldsymbol{b}_{k,l}^\top\right)\boldsymbol{x}_i(c)\\\vdots\\\sum_{c=1}^{c_0}\left(\sum_{k=1}^{|G|}\boldsymbol{w}_{(c_1,c)}(k)\boldsymbol{b}_{k,l}^\top\right)\boldsymbol{x}_i(c)\end{pmatrix}\right\|\right),$$

where $\boldsymbol{b}_{k,l}^\top$ is the $l$'th row of $\boldsymbol{B}_k$. From here, we can continue similarly. First, we use the following moment inequality:

$$\mathbb{E}_{\boldsymbol{\epsilon}}\left(\sup_{\boldsymbol{w}}\left\|\sum_{i=1}^m\epsilon_i\sigma\begin{pmatrix}\sum_{c=1}^{c_0}\left(\sum_{k=1}^{|G|}\boldsymbol{w}_{(1,c)}(k)\boldsymbol{b}_{k,l}^\top\right)\boldsymbol{x}_i(c)\\\vdots\\\sum_{c=1}^{c_0}\left(\sum_{k=1}^{|G|}\boldsymbol{w}_{(c_1,c)}(k)\boldsymbol{b}_{k,l}^\top\right)\boldsymbol{x}_i(c)\end{pmatrix}\right\|\right)$$

$$\leq\sqrt{\mathbb{E}_{\boldsymbol{\epsilon}}\left(\sup_{\boldsymbol{w}}\left\|\sum_{i=1}^m\epsilon_i\sigma\begin{pmatrix}\sum_{c=1}^{c_0}\left(\sum_{k=1}^{|G|}\boldsymbol{w}_{(1,c)}(k)\boldsymbol{b}_{k,l}^\top\right)\boldsymbol{x}_i(c)\\\vdots\\\sum_{c=1}^{c_0}\left(\sum_{k=1}^{|G|}\boldsymbol{w}_{(c_1,c)}(k)\boldsymbol{b}_{k,l}^\top\right)\boldsymbol{x}_i(c)\end{pmatrix}\right\|^2\right)}$$

$$=\sqrt{\mathbb{E}_{\boldsymbol{\epsilon}}\left(\sup_{\boldsymbol{w}}\sum_{j=1}^{c_1}\left(\sum_{i=1}^m\epsilon_i\sigma\left(\sum_{c=1}^{c_0}\left(\sum_{k=1}^{|G|}\boldsymbol{w}_{(j,c)}(k)\boldsymbol{b}_{k,l}^\top\right)\boldsymbol{x}_i(c)\right)\right)^2\right)}.$$

For the rest, denote (by abuse of notation, we drop the index $l$ when it is evident in context):

$$\left\|\boldsymbol{w}_{(j,:)}\right\|_{\boldsymbol{B}} := \left\|\left(\sum_{k=1}^{|G|}\boldsymbol{w}_{(j,c)}(k)\boldsymbol{b}_{k,l}^{\top}\right)_{c\in[c_0]}\right\|_F , \left\|\boldsymbol{w}\right\|_{\boldsymbol{B}} := \max_{l\in[|G|]}\left\|\left(\sum_{k=1}^{|G|}\boldsymbol{w}_{(j,c)}(k)\boldsymbol{b}_{k,l}^{\top}\right)_{c\in[c_0],j\in[c_1]}\right\|_F$$

Note that:

$$\sum_{j=1}^{c_1}\left\|\boldsymbol{w}_{(j,:)}\right\|_{\boldsymbol{B}}^2 = \left\|\left(\sum_{k=1}^{|G|}\boldsymbol{w}_{(j,c)}(k)\boldsymbol{b}_{k,l}^{\top}\right)_{c\in[c_0],j\in[c_1]}\right\|_F^2 \le \max_{l\in[|G|]}\left\|\left(\sum_{k=1}^{|G|}\boldsymbol{w}_{(j,c)}(k)\boldsymbol{b}_{k,l}^{\top}\right)_{c\in[c_0],j\in[c_1]}\right\|_F^2 \le (M_2^{w.s.})^2.$$

We can continue the proof as follows:

$$\mathbb{E}_{\boldsymbol{\epsilon}}\left(\sup_{\boldsymbol{w}}\sum_{j=1}^{c_1}\left(\sum_{i=1}^{m}\epsilon_i\sigma\left(\sum_{c=1}^{c_0}\left(\sum_{k=1}^{|G|}\boldsymbol{w}_{(j,c)}(k)\boldsymbol{b}_{k,l}^{\top}\right)\boldsymbol{x}_i(c)\right)\right)^2\right)$$

$$\le \mathbb{E}_{\boldsymbol{\epsilon}}\left(\sup_{\boldsymbol{w}}\sum_{j=1}^{c_1}\left\|\boldsymbol{w}_{(j,:)}\right\|_{\boldsymbol{B}}^2\left(\sum_{i=1}^{m}\epsilon_i\sigma\left(\frac{1}{\left\|\boldsymbol{w}_{(j,:)}\right\|_{\boldsymbol{B}}}\sum_{c=1}^{c_0}\left(\sum_{k=1}^{|G|}\boldsymbol{w}_{(j,c)}(k)\boldsymbol{b}_{k,l}^{\top}\right)\boldsymbol{x}_i(c)\right)\right)^2\right)$$

$$\le \mathbb{E}_{\boldsymbol{\epsilon}}\left(\sup_{\boldsymbol{w}}\sum_{j=1}^{c_1}\left\|\boldsymbol{w}_{(j,:)}\right\|_{\boldsymbol{B}}^2\left(\sum_{i=1}^{m}\epsilon_i\sigma\left(\frac{1}{\left\|\boldsymbol{w}_{(j,:)}\right\|_{\boldsymbol{B}}}\sum_{c=1}^{c_0}\left(\sum_{k=1}^{|G|}\boldsymbol{w}_{(j,c)}(k)\boldsymbol{b}_{k,l}^{\top}\right)\boldsymbol{x}_i(c)\right)\right)^2\right)$$

$$\le (M_2^{w.s.})^2\mathbb{E}_{\boldsymbol{\epsilon}}\left(\sup_{\boldsymbol{w}}\sup_{j}\left(\sum_{i=1}^{m}\epsilon_i\sigma\left(\frac{1}{\left\|\boldsymbol{w}_{(j,:)}\right\|_{\boldsymbol{B}}}\sum_{c=1}^{c_0}\left(\sum_{k=1}^{|G|}\boldsymbol{w}_{(j,c)}(k)\boldsymbol{b}_{k,l}^{\top}\right)\boldsymbol{x}_i(c)\right)\right)^2\right)$$

$$\le (M_2^{w.s.})^2\mathbb{E}_{\boldsymbol{\epsilon}}\left(\sup_{\hat{\boldsymbol{w}}}\left(\sum_{i=1}^{m}\epsilon_i\sigma\left(\sum_{c=1}^{c_0}\left(\sum_{k=1}^{|G|}\hat{\boldsymbol{w}}_{(c)}(k)\boldsymbol{b}_{k,l}^{\top}\right)\boldsymbol{x}_i(c)\right)\right)^2\right),$$

where the supremum is taked over all vectors $\hat{\boldsymbol{w}}$ satisfying:

$$\left\|\left(\sum_{k=1}^{|G|}\hat{\boldsymbol{w}}_{(c)}(k)\boldsymbol{b}_{k,l}^{\top}\right)_{c\in[c_0]}\right\|_F \le 1.$$

We can use the contraction inequality and the Cauchy-Schwartz inequality to obtain the following:

$$\mathbb{E}_{\boldsymbol{\epsilon}}\left(\sup_{\hat{\boldsymbol{w}}}\left(\sum_{i=1}^{m}\epsilon_i\sigma\left(\sum_{c=1}^{c_0}\left(\sum_{k=1}^{|G|}\hat{\boldsymbol{w}}_{(c)}(k)\boldsymbol{b}_{k,l}^{\top}\right)\boldsymbol{x}_i(c)\right)\right)^2\right)$$

$$\le \mathbb{E}_{\boldsymbol{\epsilon}}\left(\sup_{\hat{\boldsymbol{w}}}\left(\sum_{i=1}^{m}\epsilon_i\left(\sum_{c=1}^{c_0}\left(\sum_{k=1}^{|G|}\hat{\boldsymbol{w}}_{(c)}(k)\boldsymbol{b}_{k,l}^{\top}\right)\boldsymbol{x}_i(c)\right)\right)^2\right)$$

$$= \mathbb{E}_{\boldsymbol{\epsilon}}\left(\sup_{\hat{\boldsymbol{w}}}\left(\sum_{c=1}^{c_0}\left(\sum_{k=1}^{|G|}\hat{\boldsymbol{w}}_{(c)}(k)\boldsymbol{b}_{k,l}^{\top}\right)\left(\sum_{i=1}^{m}\epsilon_i\boldsymbol{x}_i(c)\right)\right)^2\right)$$

$$\le \sup_{\hat{\boldsymbol{w}}}\left\|\left(\sum_{k=1}^{|G|}\hat{\boldsymbol{w}}_{(c)}(k)\boldsymbol{b}_{k,l}^{\top}\right)_{c\in[c_0]}\right\|_F^2\mathbb{E}_{\boldsymbol{\epsilon}}\left(\left\|\sum_{i=1}^{m}\epsilon_i\boldsymbol{x}_i\right\|^2\right) \le mb_x^2.$$

## H    Proofs for Local Filters - Proposition 7.1

Consider the network defined as:

$$h_{\boldsymbol{u},\boldsymbol{w}}(\boldsymbol{x}) = \boldsymbol{u}^\top P \circ \sigma \begin{pmatrix} \left( \sum_{k=1}^{c_0} \left( \boldsymbol{w}_{(1,k)}^\top \phi_l(\boldsymbol{x}(k)) \right) \right)_{l \in [|G|]} \\ \vdots \\ \left( \sum_{k=1}^{c_0} \left( \boldsymbol{w}_{(c_1,k)}^\top \phi_l(\boldsymbol{x}(k)) \right) \right)_{l \in [|G|]} \end{pmatrix} = \boldsymbol{u}^\top P \circ \sigma(\boldsymbol{W}\boldsymbol{x}),$$

where similar to Vardi et al. (2022), $\boldsymbol{W}$ is the matrix *conforming* to the patches $\Phi$ and the weights. The first steps for average pooling are similar to what we have done in Section C.2. The only difference is that $\Pi_l \boldsymbol{x}(k)$ is replaced with $\phi_l(\boldsymbol{x}(k))$. We will not repeat the arguments and condense them in the following steps:

$$\mathbb{E}_{\boldsymbol{\epsilon}} \left( \sup_{\boldsymbol{u},\boldsymbol{w}} \frac{1}{m} \sum_{i=1}^m \epsilon_i \boldsymbol{u}^\top P \circ \sigma \left( \boldsymbol{W}\boldsymbol{x} \right) \right) \leq \frac{M_1}{m} \mathbb{E}_{\boldsymbol{\epsilon}} \left( \sup_{\boldsymbol{w}} \left\| \sum_{i=1}^m \epsilon_i P \circ \sigma(\boldsymbol{W}\boldsymbol{x}_i) \right\| \right).$$

$$\leq \frac{M_1}{m} \sum_{l=1}^{|G|} \frac{1}{|G|} \mathbb{E}_{\boldsymbol{\epsilon}} \left( \sup_{\boldsymbol{w}} \left\| \sum_{i=1}^m \epsilon_i \begin{pmatrix} \sigma \left( \sum_{k=1}^{c_0} \boldsymbol{w}_{(1,k)}^\top \phi_l(\boldsymbol{x}_i(k)) \right) \\ \vdots \\ \sigma \left( \sum_{k=1}^{c_0} \boldsymbol{w}_{(c_1,k)}^\top \phi_l(\boldsymbol{x}_i(k)) \right) \end{pmatrix} \right\| \right)$$

And, we can repeat the peeling arguments to get:

$$\mathbb{E}_{\boldsymbol{\epsilon}} \left( \sup_{\boldsymbol{w}} \left\| \sum_{i=1}^m \epsilon_i \begin{pmatrix} \sigma \left( \sum_{k=1}^{c_0} \boldsymbol{w}_{(1,k)}^\top \phi_l(\boldsymbol{x}_i(k)) \right) \\ \vdots \\ \sigma \left( \sum_{k=1}^{c_0} \boldsymbol{w}_{(c_1,k)}^\top \phi_l(\boldsymbol{x}_i(k)) \right) \end{pmatrix} \right\| \right) \leq M_2 \sqrt{\mathbb{E}_{\boldsymbol{\epsilon}} \left( \sup_{\tilde{\boldsymbol{w}}:\|\tilde{\boldsymbol{w}}\| \leq 1} \left( \sum_{i=1}^m \epsilon_i \sigma \left( \sum_{k=1}^{c_0} \tilde{\boldsymbol{w}}_{(k)}^\top \phi_l(\boldsymbol{x}_i(k)) \right) \right)^2 \right)}.$$

The only change in the proof is about the way we simplify the term in the square root:

$$\mathbb{E}_{\boldsymbol{\epsilon}} \left( \sup_{\tilde{\boldsymbol{w}}:\|\tilde{\boldsymbol{w}}\| \leq 1} \left( \sum_{i=1}^m \epsilon_i \sigma \left( \sum_{k=1}^{c_0} \tilde{\boldsymbol{w}}_{(k)}^\top \phi_l(\boldsymbol{x}_i(k)) \right) \right)^2 \right) \leq \mathbb{E}_{\boldsymbol{\epsilon}} \left( \sup_{\tilde{\boldsymbol{w}}:\|\tilde{\boldsymbol{w}}\| \leq 1} \left( \sum_{i=1}^m \epsilon_i \sum_{k=1}^{c_0} \tilde{\boldsymbol{w}}_{(k)}^\top \phi_l(\boldsymbol{x}_i(k)) \right)^2 \right)$$

$$\leq \mathbb{E}_{\boldsymbol{\epsilon}} \left( \sup_{\tilde{\boldsymbol{w}}:\|\tilde{\boldsymbol{w}}\| \leq 1} \left( \sum_{k=1}^{c_0} \tilde{\boldsymbol{w}}_{(k)}^\top \left( \sum_{i=1}^m \epsilon_i \phi_l(\boldsymbol{x}_i(k)) \right) \right)^2 \right)$$

$$\leq \mathbb{E}_{\boldsymbol{\epsilon}} \left( \left\| \sum_{i=1}^m \epsilon_i \phi_l(\boldsymbol{x}_i) \right\|^2 \right) = \sum_{i=1}^m \|\phi_l(\boldsymbol{x}_i)\|^2. \qquad (36)$$

Now we use Jensen inequality to get the proof:

$$\mathbb{E}_{\boldsymbol{\epsilon}} \left( \sup_{\boldsymbol{u},\boldsymbol{w}} \frac{1}{m} \sum_{i=1}^m \epsilon_i \boldsymbol{u}^\top P \circ \sigma \left( \boldsymbol{W}\boldsymbol{x} \right) \right) \leq \frac{M_1 M_2}{m} \frac{1}{|G|} \sum_{l=1}^{|G|} \sqrt{\sum_{i=1}^m \|\phi_l(\boldsymbol{x}_i)\|^2}$$

$$\leq \frac{M_1 M_2}{m} \sqrt{\frac{1}{|G|} \sum_{l=1}^{|G|} \sum_{i=1}^m \|\phi_l(\boldsymbol{x}_i)\|^2}$$

$$\leq \frac{M_1 M_2}{m} \sqrt{\frac{m O_\Phi b_x^2}{|G|}} = \frac{M_1 M_2 b_x}{\sqrt{m}} \sqrt{\frac{O_\Phi}{|G|}}.$$

## I    Uncertainty Principle for Local Filters - Proposition 7.2

We start by stating the uncertainty principle for finite Abelian groups.

| Reference | Bound |
|---|---|
| Equivariance (Theorem 4.1) | $\frac{b_x M_1 M_2}{\sqrt{m}}$ |
| Weight sharing (Proposition 6.1) | $\frac{b_x M_1 M_2^{w.s.}}{\sqrt{m}}$ |
| Locality (Proposition 7.1) | $\sqrt{\frac{O_\Phi}{|G|}} \frac{b_x M_1 M_2}{\sqrt{m}}$ |
| Non-Equivariant Networks Golowich et al. (2018) | $\frac{b_x M_1 \|\boldsymbol{W}\|_F}{\sqrt{m}}$ |
| Weight sharing, orthogonal $\boldsymbol{b}_{k,l}, k \in [|G|]$ | $\frac{b_x M_1 M_2}{\sqrt{m}}$ |
| $B$-sparse filters (Proposition 7.2 | $b_x M_1 M_2 / \sqrt{mB}$ |
| Linear convolution Vardi et al. (2022) | $\sqrt{O_\Phi} \frac{b_x M_1 \|\boldsymbol{W}\|_{2\to 2}}{\sqrt{m}}$ |
| CNN with Pooling Vardi et al. (2022) | $\frac{b_x \|\boldsymbol{W}\|_{2\to 2} \log m \sqrt{\log(mc_1 |G|)}}{\sqrt{m}}$ |
| Graf et al. (2022) | $\mathcal{O}\left(\frac{b_x M_2 \log(|G|\max(c_0,c_1))\sqrt{\|\boldsymbol{W}\|_{2\to 2}+\|\boldsymbol{w}\|_{2,1}}}{\sqrt{m}}\right)$ |

Table 2: The table summarizes the main results of the paper and compares with the selected prior works. The top three rows are the main results of the paper expressed in their full generality (Reminder : $M_2^{w.s.} = \max_{l\in[|G|]} \left\|\left(\sum_{k=1}^{|G|} \boldsymbol{w}_{(j,c)}(k)\boldsymbol{b}_{k,l}^\top\right)_{c\in[c_0],j\in[c_1]}\right\|_F$). Next three rows summarize the result for the case where the network is not equivariant, the optimal weight sharing and the band-limited filters. The bound from the most relevant results are also summarized in the last rows of the table. The explicit dimension dependency of these bounds are highlighted with a different color.

**Theorem I.1.** *Let $G$ be a finite Abelian group. Suppose the function $f : G \to \mathbb{C}$ is non-zero, and $\hat{f}$ is its Fourier transform. We have*

$$|\mathrm{supp}(f)|.|\mathrm{supp}(\hat{f})| \geq |G|.$$

If the filters have $B$ non-zero entries in the frequency domain, then the uncertainty principle implies that the filter in the spatial domain has at least $|G|/B$ non-zero entries. In other words, the smallest filter size will have $|G|/B$ non-zero elements. Therefore, the smallest possible upper bound we can obtain using our approach will use $O_\Phi = |G|/B$. Plugging this in Proposition 7.1 yields the proposition.

## J  Comparison with other existing bounds

The following works do not consider the generalization error for equivariant networks. However, the convolutional networks, which are studied in their work, are closely related to our results. We have also summarized our results and their comparison in Table 2.

**Comparison with Vardi et al. (2022).**  Our work is closely related to Vardi et al. (2022). We focus on equivariant networks and upper bounds on the sample complexity. The networks considered in this work are slightly more general with per-channel linear and non-linear pooling operations. They also provide a bound in Theorem 7 of their paper for pooling operations $\rho : \mathbb{R}^{|G|} \to \mathbb{R}$ that are 1-Lipschitz with respect to the $\ell_\infty$ norm. This class includes average and max pooling operations. Their bound has logarithmic dimension dependence, the artifact of using the covering number-based arguments (see below for more discussions). The bound is also dependent on the spectral norm of the matrix $\boldsymbol{W}$ instead of the norm of the weight $\boldsymbol{w}$. It is not clear if the dependence on the spectral norm can be removed in their proof. For local filters, they provide a dimension-free bound that also depends on the spectral norm of $\boldsymbol{W}$. However, this time, the dependence on the spectral norm can be substituted with the norm of the weight vector, as it is evident from their proof. In general, all our bounds depend only on the norms of the weight vector, which is tighter than using Frobenius or spectral norm of $\boldsymbol{W}$. The authors in Vardi et al. (2022) provide lower bounds on the sample complexity, which we relegate to future works.

**Comparison with Graf et al. (2022).** The paper of Graf et al. (2022) relies mainly on Maurey's empirical lemma and covers a number of arguments. Rademacher complexity bounds either work directly on bounding the Rademacher sum through a set of inequalities or utilize chaining arguments and Dudley's integral inequality. Dudley's integral requires an estimate of the covering number. The main techniques of finding the covering number, like volumetric, Sudakov, and Maurey's lemma techniques, manifest different dimension dependencies, with the latter usually providing the mildest dependence. Completely dimension-free bounds are obtained mainly via the direct analysis of the Rademacher sum. Therefore, the bounds in Graf et al. (2022) are a mixture of dimension-dependent and norm-dependent terms. Their bounds are, however, quite general and consider different activation functions and residual connections.

To investigate their bound further, let's consider their setup, focusing on what matters for the current paper. They consider a network with $L$ residual blocks, where the residual block $i$ has $L_i$ layers. That is:

$$f = \sigma_L \circ f_L \circ \cdots \circ \sigma_1 \circ f_1 \text{ with } f_i(x) = g_i(x) + \sigma_{iL_i} \circ h_{iL_i} \circ \cdots \circ \sigma_{i1} \circ h_{i1}(x).$$

The activation functions $\sigma_i$ and $\sigma_{ij}$ have respectively the Lipschitz constants of $\rho_i$ and $\rho_{ij}$. The function $g_i(\cdot)$ represents the residual connection with $g_i(0) = 0$. The map $h_{ij}$ represents the convolutional operation with the Kernel $K_{ij}$. The Lipschitz constant of the layer is given by $s_{ij}$, which corresponds to the spectral norm of the matrix that *conforms* with the layer (similar to the spectral norm of $\boldsymbol{W}$). The $\ell_{2,1}$ norm of the kernels $K_{ij}$ is bounded by $b_{ij}$ (this can be initialization dependent or independent). The norm of $K \in \mathbb{R}^{c_1 \times c_0 \times d}$ is defined as

$$\|K\|_{2,1} = \sum_{i,j} \|K(i,:,j)\|_2.$$

Define $s_i = \text{Lip}(g_i) + \prod_{j=1}^{L_i} \rho_{ij} s_{ij}$. Denote the total number of layers by $\overline{L}$, $W_{ij}$ is the number of weights in the $j$-th layer in the $i$-th residual block. Set $W = \max W_{ij}$. And $C_{ij}$ represents the dependence on the weight and data norms:

$$C_{ij} = \frac{2\|\boldsymbol{X}\|}{\sqrt{m}} \left( \prod_{l=1}^{L} s_l \rho_l \right) \frac{\prod_{k=1}^{L_i} s_{ik} \rho_{ik}}{s_i} \frac{b_{ij}}{s_{ij}}.$$

Define $\tilde{C}_{ij} = 2C_{ij}/\gamma$ where $\gamma$ is the margin. The authors have two bounds where the dependence on $W$ appears logarithmically or directly. We focus on the former. The equation (16) gives the generalization error as

$$\mathcal{O} \left( \sqrt{\frac{\log(2W) \sum_{i,j} \overline{L}^2 \tilde{C}_{ij}^2}{m}} \right)$$

First, we have $W = |G| \max(c_0, c_1)$. Secondly, for the first layer, we have:

$$C_{11} \leq b_x (s_{11} s_{12}) \frac{s_{11} s_{12}}{s_{11} s_{12}} \frac{b_{11}}{s_{11}} = b_x b_{11} s_{12} \leq b_x M_2 \|\boldsymbol{w}\|_{2,1},$$

with

$$\|\boldsymbol{w}\|_{2,1} = \sum_{j=1}^{c_0} \|\boldsymbol{w}_{:,j}\|_2.$$

Finally, for the last layer, we get:

$$C_{12} \leq b_x (s_{11} s_{12}) \frac{s_{11} s_{12}}{s_{11} s_{12}} \frac{b_{11}}{s_{11}} = b_x b_{12} s_{11} \leq b_x M_2 \|\boldsymbol{W}\|_{2 \to 2}$$

Ignoring the constants and margins, the generalization error will be:

$$\mathcal{O} \left( \frac{b_x M_2 \log(|G| \max(c_0, c_1)) \sqrt{\|\boldsymbol{W}\|_{2 \to 2} + \|\boldsymbol{w}\|_{2,1}}}{\sqrt{m}} \right).$$

Apart from its dimension dependence, both norms $\|\boldsymbol{W}\|_{2 \to 2}$ and $\|\boldsymbol{w}\|_{2,1}$ are bigger than $\|\boldsymbol{w}\|$ used in our bounds. Therefore, our bound is tighter. However, they consider more general and exhaustive scenarios, including deep networks with residual connections and general Lipschitz activations.

**Comparison with Wang and Wu (2023).** In Wang and Wu (2023), they study the same set of problems as ours, related to locality and weight sharing, but from different perspectives. First, the approximation theory perspective, studying the class of functions a model can approximate, is complementary to our paper on the generalization error using statistical learning theory. The learning theory part of the paper is focused on a particular "separation task" (see (5)) with a fixed input distribution. They also use Rademacher complexity but bound it with a covering number. We work with general input distributions and arbitrary tasks. Besides, our bounds are dimension independent, while their bound explicitly depends on the input dimension, an artifact of using covering number for bounding RC.

**Comparison with Li et al. (2021).** In Li et al. (2021), the authors consider the sample complexity of convolutional and fully connected models and construct a distribution for which there is a fundamental gap between them in the sample complexity. Their analysis is based on the VC dimension and uses Benedek-Itai's lower bound. Their final bound is, therefore, dependent on the input dimension. In contrast, one of the main interests of our paper was to explore dimension-free bounds. We additionally study the impact of locality and weight sharing.

### J.1 Regarding Norm-based bounds and Other Desiderata of Learning Theory

**Dimension Free and Norm Based Bounds.** Dimension-free bounds are interesting because they hint at why the generalization error is unaffected by over-parametrizing the model. In this sense, the notion of dimension refers mainly to the input dimension, the number of channels, the width of a layer, and the number of layers. The same motivation existed for obtaining norm-based bounds for RKHS SVM models. The works of Golowich et al. (2018) and Vardi et al. (2022) are two recent examples, with the former work providing an extensive review of other dimension-free bounds. Choosing a proper norm to get dimension-free bounds is one of the main questions in learning theory. Naturally, this is easier to get for larger norms like Frobenius norms, and the question is if we can have similar dimension-free results for smaller norms like spectral or, as we show in the paper, the norm of effective parameters.

**Tightness of Rademacher Complexity.** In the literature around deep learning theory, there are many works questioning the relevance of classical learning theoretic results, for example, using Rademacher complexity, norm-based bounds, and uniform convergence. We can refer, for example, to works such as Nagarajan and Kolter (2019); Jiang et al. (2020); Zhang et al. (2017); Zhu et al. (2021). It is fair to ask if Rademacher complexity analysis, in light of these criticisms, can provide tight bounds for our case. First of all, the preceding works focused on state-of-the-art deep neural networks with a large number of layers. Our study is limited to single-hidden layer neural networks. Second, Rademacher complexity is indeed tight for support vector machines Steinwart and Christmann (2008); Shalev-Shwartz and Ben-David (2014); Bartlett and Mendelson (2002). If we fix the first layer of our network and only train the last layer, then we have a linear model, and it is known that training it with Gradient descent will lead to minimum $\ell_2$-norm solutions. For this sub-class of learning algorithms, the analysis boils to previous works on minimum norm classifiers as in Shalev-Shwartz and Ben-David (2014) for which the Rademacher complexity analysis is tight. Therefore, we believe that Rademacher complexity can still be a relevant tool for shallower models, including those discussed in this paper.

**Implicit Bias of Linear Networks** Previous works also tried to characterize the effect of equivairance on the training dynamics. Lawrence et al. (2022) showed that deep equivariant linear networks trained with gradient descent provably converge to solutions that are sparse in the Fourier domain. While this work focuses on the effect of equivariance on optimization, the authors suggest that this implicit bias towards sparse Fourier solutions can be beneficial towards generalization in bandlimited datasets - which is common for non-discrete compact groups such as rotation groups and a common assumption in steerable CNNs. Similar results were also previously obtained in Gunasekar et al. (2018) for standard convolutional networks, which can be seen as an instance of GCNN with discrete abelian groups.

**Discussions on the Limitations.** This paper provided a Rademacher complexity-based bound for single-hidden layer equivariant networks. We obtained a lower bound, which showed the tightness of our analysis.

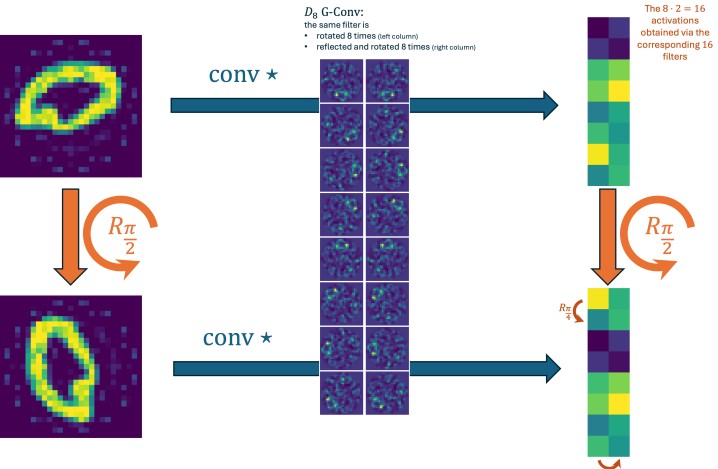

Figure 5: Graphical representation of the equivariant linear projection used to preprocess the image data. An example image is projected with a single filter rotated 8 times and mirrored and rotated 8 more times. The output is a $8 \cdot 2 = 16$ dimensional vector representing a signal over $G = D_8$. A rotation or mirroring of the input image results in a periodic shift or permutation of the output channels.

The numerical results showed compelling scaling behaviors for our bound. However, there is still a gap between our bound and the generalization error. We conjecture that the gap is related to the idea of implicit bias. We conjecture that stochastic gradient descent training implicitly biases toward only a subset of the hypothesis space even smaller than norm-bounded functions. It is an important next step to tie these two analyses together by characterizing the implicit bias, for example, in terms of some kind of norms, and then using Rademacher complexity analysis to find a bound on the generalization error based on the implicit bias.

On the other hand, we provided a lower bound on the Rademacher complexity. It would be interesting to obtain such bounds by finding the fat-shattering dimension. If two points in the dataset can be transformed into each other by the action of group $G$, then the space of $G$-invariant functions cannot shatter such datasets. Therefore, the data points should be picked on different orbits so they can be shattered. Such construction will be interesting in the future.

## K   Experiment Setup

To pre-process the MNIST and the CIFAR10 datasets, we first create a single $D_{32}$ steerable convolution layer with `kernel size` equal to the images' size, one input channel and $100 \times |D_{32}| = 6400$ output channels. In particular, under the steerable CNNs framework, we use 100 copies of the *regular representation* of the group $D_{32}$ as output feature type. Alternatively, in the group-convolution framework, this steerable convolution layer corresponds to *lifting convolution* with 100 output channels, mapping a 1-channel scalar image to a 100-channels signal over the whole $\mathbb{R}^2 \rtimes D_{32}$ group; because the `kernel size` is as large as the input image (because we use no padding), the spatial resolution of the output of the convolution is a single pixel, leaving only a feature vector over $G = D_{32}$.

This construction guarantees that a rotation of the raw image by an element of $D_{32}$ (and, therefore, of any of its subgroups) results in a corresponding shift of the projected 100-channels signal over $D_{32}$.

Fig. 5 shows an example of an input image projected with a single filter rotated 8 times and mirrored and rotated 8 more times (i.e. encoded via a $G = D_8$ steerable convolution). The output is a $8 \cdot 2 = 16$ dimensional vector representing a signal over $G = D_8$. A rotation or mirroring of the input image results in a periodic shift or permutation of the output channels.

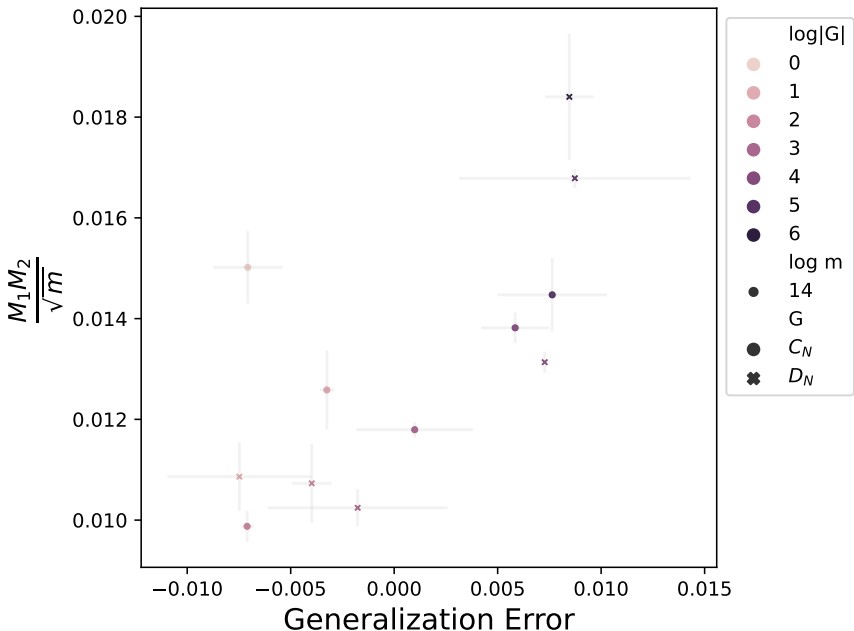

Figure 6: Subset of Fig. 2c focusing on the models training with the largest dataset size $m = 25600$. The models using the smallest equivariance groups show increased norms.

Finally, to avoid interpolation artifacts, we augment our dataset by directly transforming the projected features (which happens via simple permutation matrices since the group $D_{32}$ is discrete), rather than pre-rotating the raw images.

Our equivariant networks consist of a linear layer (i.e. a group convolution), followed by a ReLU activation and a final pooling and invariant linear layer as in eq. 3. All MNIST models are trained using the Adam optimizer with a learning rate of $1e - 2$ for 30 epochs, while the CIFAR10 models are trained using with a learning rate of $1e - 3$ for 50 epochs. Precisely, we use the MNIST12K dataset, which has 12K images in the training split and 50K in the test split.

All experiments were run on a single GPU NVIDIA GeForce RTX 2080 Ti.

