# OpenReview forum: "On the Sample Complexity of One Hidden Layer Networks with Equivariance, Locality and Weight Sharing"
_TMLR — Accepted by TMLR_

### Review · Reviewer_Gyfb · 2024-11-11

**Summary Of Contributions:**

This paper examines how certain design features—equivariance, locality, and weight sharing—impact the sample efficiency of one-hidden-layer neural networks. Using statistical learning theory, the authors find that these features can improve generalization, leading to smaller sample requirements. They derive bounds that are independent of network size, relying only on the filter norms. The study also highlights a trade-off in generalization based on whether filters are defined in spatial or frequency domains, especially for band-limited data. These insights offer guidelines for building more sample-efficient neural networks.

**Audience:**

Yes

**Claims And Evidence:**

Yes

**Requested Changes:**

If possible, adding diagrams or brief explanations of key ideas (such as the trade-offs in locality and frequency domain analysis) could make the content more accessible.

Extending the analysis to deeper architectures, or at least discussing the potential implications for such networks, would make the paper more relevant to current machine learning practice.

**Strengths And Weaknesses:**

Strengths:

The paper provides a thorough theoretical examination of one-hidden-layer networks, leveraging Rademacher complexity analysis to derive generalization bounds. The derivation of generalization bounds that are independent of input or output dimensions is particularly noteworthy. This feature makes the findings applicable across a wide range of network configurations.

The comparison of generalization benefits across equivariant, weight-sharing, and local filter architectures enriches the understanding of their relative contributions. By offering insights into how these architectural choices affect sample complexity, the paper helps bridge theoretical analysis and practical design decisions in neural network architecture.


Weaknesses:

While the theoretical findings are robust, the paper lacks sufficient empirical validation to support its claims. Although some numerical results are mentioned, these are not extensive or detailed enough to show how the theoretical bounds hold in practice for varied datasets or tasks.

The analysis is restricted to one-hidden-layer networks, which may limit the applicability of the results to deeper architectures that are more common in practical machine learning.

The trade-off discussion between locality in spatial and frequency domains could be expanded with more intuitive examples or empirical support to highlight its practical significance.

---

> ### Author Response · Authors · 2024-12-06
> **Answers to the reviewer**
>
> We would like to thank the reviewer for the suggestions and feedbacks, particularly for suggesting to run more experiments and adding a result on multi-layer networks. We have uploaded a new version of our paper, and below, provide our answers to questions raised by the reviewer.
>
> >Reviewer: While the theoretical findings are robust, the paper lacks sufficient empirical validation to support its claims. Although some numerical results are mentioned, these are not extensive or detailed enough to show how the theoretical bounds hold in practice for varied datasets or tasks.
>
> In the new version, we have added the following experiments to explore further the relevance of our theoretical results for practical scenarios. In particular we have added the following:
>
> * We add new experiments on CIFAR10 dataset, where we show again that our bound correctly predicts the generalization ability of equivariant architectures across different dataset sizes and groups.
> * We add new experiments on the role of locality and band-limitedness on the generalization error. Again, we observe our bound can capture the effect of these design choices on generalization.
> * We include a new analysis studying the trade-offs between expressiveness and generalization when varying the band-limitedness of the models
>
> >Reviewer: The analysis is restricted to one-hidden-layer networks, which may limit the applicability of the results to deeper architectures that are more common in practical machine learning.
>
> In the new version, we have provided an extended version of the theorem for multi-layer networks. We have also included a discussion on pros and cons of the bound.
>
> >Reviewer: The trade-off discussion between locality in spatial and frequency domains could be expanded with more intuitive examples or empirical support to highlight its practical significance.
>
> We thank the reviewer for suggesting new experiments. We have provided further explanation in the new version that highlights the impact of locality on the generalization error. Besides, we have added new experiments to empirically support the intuitions provided by the theory, see Fig. 3 and Fig. 4.
>
> >Reviewer: If possible, adding diagrams or brief explanations of key ideas (such as the trade-offs in locality and frequency domain analysis) could make the content more accessible.
>
> We would like to thank the reviewer for this suggestion. We have added new tables that summarize the main take-aways as well as further comparison with the previous works. Please see the table in Appendix J. We have also updated Figure 1 for a brief summary.
>
> >Reviewer: Extending the analysis to deeper architectures, or at least discussing the potential implications for such networks, would make the paper more relevant to current machine learning practice.
>
> As mentioned above, we have included a new theorem for deeper architectures.

---

### Review · Reviewer_Vs27 · 2024-11-22

**Summary Of Contributions:**

The paper studies Rademacher complexity based generalization bounds for 1-hidden layer neural networks whose input is a signal over a group G. Bounds are given for group convolutional layers, layers with weight sharing, and group convolutional layers with localized convolution filters. The bounds are dimension free and independent of the size of G.

**Audience:**

Yes

**Broader Impact Concerns:**

-

**Claims And Evidence:**

Yes

**Requested Changes:**

I request the appropriate changes according to the answers to my questions above.

Minor changes:
1. Unless I missed it, it seems like $m, \mathcal{L}, \hat{\mathcal{L}}$ were not defined prior to their use in Thm 4.1.
2. A visualization of the weight-sharing in equation (5), similar to equation (2) would be helpful.
3. `\eqref` is more appropriate than `\ref` for referring to equations.

**Strengths And Weaknesses:**

### Strengths
1. The generalization trade-off between equivariant vs. non-equivariant neural networks is an interesting topic. New results can potentially interest a large community of researchers.
2. The paper is comprehensive and includes introductions to equivariant neural networks and Rademacher analysis in the appendix, making it quite self-contained.

### Weaknesses
1. The relation of the presented bounds to prior work could be made more explicit. For instance, it would be helpful to write out what the corresponding bound to Thm 4.1 is when $\mathbf{W}$ is not assumed to be convolutional.

Further, I have a couple of specific questions that I would like to ask the authors to clarify:

Q1. In general, I have difficulty understanding the benefits of using norm-based bounds for weight-sharing. Weight-sharing typically reduces the number of trainable parameters but not the norm of the expanded weights. In my understanding, it is the norm of the expanded weights that matters in the end (as also discussed under "Average pooling." on page 5), so it seems like these bounds do not use the weight sharing. Can the authors clarify this?

Q2. A follow-up to the previous question. If we do not assume a convolutional structure on $\mathbf{W}$ and instead assume something like $||\mathbf W|| \leq \sqrt{|G|}M_2$, do we get the same bound as in Thm 4.1 using the technique from Golowich et al. (2018)?

Q3. Do the plots in the experiments show more than the fact that the generalization error goes down as the number of training samples increases?

Q4. The paper builds on the prior works by Golowich et al. (2018) and Vardi et al. (2022) where results are presented in terms of Rademacher complexities. In the present paper, the Rademacher complexities are processed with standard techniques (Thm B.3) to obtain generalization bounds. Is there a reason for deviating from prior works and not presenting bounds in terms of Rademacher complexities?

---

> ### Author Response · Authors · 2024-12-06
> **Answers to the reviewer**
>
> We would like to thank the reviewer the useful feedbacks and suggesting the new visualization for weight sharing. We have uploaded a new version of the paper and below provide further answers to the questions.
>
> >Reviewer:The relation of the presented bounds to prior work could be made more explicit. For instance, it would be helpful to write out what the corresponding bound to Thm 4.1 is when
> $\mathbf{W}$ is not assumed to be convolutional.
>
> We would like to thank the reviewer for this suggestion. We had included a detailed comparison in Appendix J with a table containing the results.
>
> >Reviewer: In general, I have difficulty understanding the benefits of using norm-based bounds for weight-sharing. Weight-sharing typically reduces the number of trainable parameters but not the norm of the expanded weights. In my understanding, it is the norm of the expanded weights that matters in the end (as also discussed under "Average pooling." on page 5), so it seems like these bounds do not use the weight sharing. Can the authors clarify this?
>
> We would like to highlight that if the number of channels, $c_1$, increases, the overall number of parameters increases despite the weight sharing. Underlying the use of norm bounds is the observation that having more convolutional filters does not negatively impact the generalization error. Therefore, the weight sharing cannot explain the generalization only based on the reduced number of used parameters. Our bounds do use weight sharing, however they highlight the importance of weight sharing mechanism imposed by the matirces $\mathbf{B}_k$ and their rows. The results suggest that if the weight sharing mechanism has properly chosen, we get the same generalization gain as the group convolutional ones, in the class of functions with bounded norm weights; for examples, if the same rows in the matirces $\mathbf{B}_k$ are orthogonal.
>
> >Reviewer: A follow-up to the previous question. If we do not assume a convolutional structure on $\mathbf{W}$
>  and instead assume something like $\Vert \mathbf{W}\Vert\leq \sqrt{|G|} M_2$ , do we get the same bound as in Thm 4.1 using the technique from Golowich et al. (2018)?
>
> The answer is positive for average pooling, as it can be seen as a linear layer. Theorem 1 in \cite{golowich_size-independent_2018}, ignoring constant terms, provides a similar norm based bound in this case, as we have discussed in the discussion around average pooling.
>
> >Reviewer: Do the plots in the experiments show more than the fact that the generalization error goes down as the number of training samples increases?
>
> Our experiments show that our theoretical bound can indeed predict the generalization ability of the architectures, not only when the training samples increase, but also across a variety of equivariance groups (and also when varying these two factors simultaneously). Moreover, in the updated experiments, we show this also holds across different datasets (MNIST and CIFAR10) and different equivariant design choices (i.e. varying the maximum frequency used to parameterize the filters).
>
>
> >Reviewer: The paper builds on the prior works by Golowich et al. (2018) and Vardi et al. (2022) where results are presented in terms of Rademacher complexities. In the present paper, the Rademacher complexities are processed with standard techniques (Thm B.3) to obtain generalization bounds. Is there a reason for deviating from prior works and not presenting bounds in terms of Rademacher complexities?
>
> The only reason was to make the ultimate bound more concrete for the researchers not familiar with Rademacher complexity. We thought bounding the generalization error directly can be more interpretable for many practitioners.
>
> >Reviewer: Unless I missed it, it seems like $\mathcal{L}$ and $\hat{\mathcal{L}}$ were not defined prior to their use in Thm 4.1.  A visualization of the weight-sharing in equation (5), similar to equation (2) would be helpful. \textbackslash eqref is more appropriate than \textbackslash ref for referring to equations
>
> We would like to thank the reviewer for mentioning these details. We have fixed all these issues. In particular, we have update Figure 1 to have a visualization of equation (5).

---

> > ### Comment · Reviewer_Vs27 · 2024-12-09
> >
> > Thank you for the answers. The added experiments and theoretical results strengthen the paper. I will read through them in more depth later this week.
> >
> > I am still unsure if I understand the norm-based bounds. Let me concretize: If we have a group convolutional layer with $||\mathbf{w}||\leq M_2$, we get the bound in Thm 4.1. But, we also get the same bound if $||\mathbf{W}||\leq \sqrt{|G|}M_2$ for general $\mathbf{W}$, which is satisfied, for instance, when each row of $\mathbf{W}$ has norm bounded by $M_2$. So, according to these bounds, is anything gained in generalization from having a group convolutional layer (or other weight sharing)?

---

> > > ### Comment · Reviewer_Vs27 · 2024-12-16
> > >
> > > I've now read through the added experimental results in more detail. I think the experiments are relevant. The conclusion from Figure 3 seems to be that the generalization properties are entirely determined by the norm of the weights. This is also consistent with the following excerpt from the paper Conclusion:
> > >
> > > > The first insight from our analysis is that the suitable weight-sharing techniques should be
> > > able to provide similar guarantees. This is not surprising, as we did not assume any symmetry in the data
> > > distribution.
> > >
> > > So the paper provides bounds that do not use the unique advantage of equivariance, i.e. alignment with the data symmetries. Thus the bounds are in some sense of limited interest. Nevertheless I believe that the paper satisfies the 2 criteria for acceptance to TMLR as I indicated in my initial review.

---

> > > > ### Author Response · Authors · 2024-12-17
> > > > **Answer to the reviewer - follow up**
> > > >
> > > > We thank the reviewer for the nice summary. Let us restate and expand on the summary provided by the reviewer.
> > > >
> > > > The case of average pooling is a special one. The pooling operation is essentially a linear operation and therefore, as we mentioned in the paper, it becomes a simple two-layers networks. In this case, *if we work with the upper bounds on the layer norm*, the convolution does not seem to provide any additional benefit in term of generalization for *general distributions*.
> > > >
> > > > However, it is important to  highlight **two points**.
> > > >
> > > > First, that the generalization error has been obtained for an arbitrary distribution, which is common practice in learning theory. It is not for data distributions that share the symmetry of the model. In this case, our lower bound, Theorem 4.4, already shows that there is a distribution for which the bound is tight. The proof shows that the distribution does not have any symmetry.
> > > > It is already known in the literature that the equivariant models provide additional gain when there is data symmetry (see Sokolic et al. (2017a)).
> > > >
> > > > The second point is about the norms used to compute the generalization bound. Our experiments confirm that when the model symmetry approximates well the data symmetry, the final norm of the layers tends to be smaller. This means that although the bound seems similar for equivariant and non-equivariant models, the weight norm of the equivariant model tends to be  smaller.
> > > >
> > > > To summarize, there are distributions for which there is no gain in generalization error by using the convolutional layer, and the magnitude of the norm can vary across models as well. We thank the reviewer for asking this question, and we will add this discussion to the paper.

---

> > > > > ### Comment · Reviewer_Vs27 · 2024-12-17
> > > > >
> > > > > Thanks for the expanded answer. I agree that it is a contribution to explicitly show that equivariant models do not give generalization benefits for general data distributions.
> > > > >
> > > > > Can you point me to the graph that demonstrates the below?
> > > > >
> > > > > > Our experiments confirm that when the model symmetry approximates well the data symmetry, the final norm of the layers tends to be smaller

---

> > > > > > ### Author Response · Authors · 2024-12-20
> > > > > >
> > > > > > The main figures to look for are those in Figure 2. For example, in Figure 2.b, let's look at $m=6000$. It can be seen that the norm bound for the smallest group, interpreted as \textit{less equivariant}, is high, and then for example, the groups of size 16 has the smallest norm.  The zoomed out version of Fig. 2.c for CIFAR10 (see Fig.6 in appendix of updated manuscript) shows a similar result. However, these experimental results can have many variations due to many factors: hyperparameter selection, the inductive bias of the training algorithm, the data symmetries only being approximate, and therefore the answer should be more nuanced. For example, in case of CIFAR10, the largest groups have high norm as well. What is safe to say from our experiments is that: the generalization error seems to correlate well with the norm-based bound, both decrease well with the training set size, and incorporating symmetries seems to be beneficial for the generalization with respect to no generalization.

---

> > > > > > > ### Comment · Reviewer_Vs27 · 2024-12-23
> > > > > > >
> > > > > > > For the record, I do not agree that we can see from these experimental results that being equivariant w.r.t. a larger group corresponds to weights with lower norms. It seems like the lowest norm is obtained at log|G| = 2 in Figure 6, and this is one of the smallest log|G|. I agree with the authors that there could be many reasons for this. But it should not be considered a contribution of the present paper to experimentally demonstrate, as the authors initially claimed, that
> > > > > > >
> > > > > > > > when the model symmetry approximates well the data symmetry, the final norm of the layers tends to be smaller
> > > > > > >
> > > > > > > In any case, I recommend accepting this submission to TMLR as it satisfies the two criteria for acceptance.

---

### Review · Reviewer_vmUN · 2024-11-22

**Summary Of Contributions:**

Characterizes a class of models with given weight matrix norms, 1-Lipschitz constant, and positively homogeneous activation, for which it is possible to bound the Rademacher complexity by a value proportional to the product of the input magnitude bound, the magnitude of the weight matrix, and the magniude of the aggregation of the pooled values, divided by the square root of the data size. Towards the end, there is a slight generalization to include a notion of locality. Finally, there is a very brief experiment on a MNIST-like dataset.

**Audience:**

Yes

**Claims And Evidence:**

Yes

**Requested Changes:**

Suggestions for improvement:
- Give a generalization of the result beyond one-hidden-layer networks, even if quite naive. Obviously it's great to motivate the result with a simplified result, but the message would be more compelling if accompanied by some intuition for how it actually works (even if not even close to optimal -- anyone theoretical enough to appreciate this paper would have some intuition for translating simple bounds).
 - The entire paper is phrased in quite image-centric language (filters, convolutions, etc.), and yet the only demonstration of the idea is on MNIST. I would love to see a demonstration of the idea on RGB images, even just CIFAR10. For example, a few pictures showing rotated, permuted entries to some patch and the sensitivity of the block action could be pedagogically interesting.
- Develop everything first in a simple ReLU (or other whatever activation is easiest) + max pool and average pool framework, without needing to talk about Abelian groups or irreps, etc.
- The experiments are, I am sorry to say, very confusing. I'm expecting bounds to be some sort of line, all I see are a bunch of similarly-colored dots that kind of have some order? But not completely since there's 2 - 12, but also C_N, etc. Figure should stand on their own mostly, and these totally do not. I'm lost from the first sentence of the description (cyclic and dihedral groups? I kind of figure it out later, but a rocky start.), and I don't know what the MNIST data set is, what a steerable convolution is, etc.
 - Ideally there would be a truly simple "validation" of the bound where there is some toy class of problems where the bound is tight, and loosened. Then perhaps some empirical examination in a simple, but fully-featured problem, that does not artificially preprocess the data, or whatever (e.g. it follows some standard pipeline like a PyTorch example or minGPT). It's a real shame for nice theoretical work to be torpedoed by unconvincing experiments, so please don't skip out on this section!
- The abstract promises a careful analysis of weight sharing, equivariance, and locality, but I'm not sure I really see it in the results. If, indeed, there's a complete decomposition of the additivity of each effect, it would be good to perhaps have a table where examples of the presence and absence of each are given, along with the corresponding result. This table could also include existing results, for example it seems like the bound being dimension-free is important, and why this is so should not be in appendix J!
- Related to the above, I would quite love to see a concrete motivation, such as the following: a generate data from a known distribution exhibiting weight sharing, equivariance, and locality and show how a model that takes account of these structures indeed generalizes better. Then tie the results to these illustrative examples. E.g. perhaps some of the examples could be constructed to obtain the upper or lower bounds.
- There are some jolting inconsistencies in the level of detail shown in different areas of the paper. Altogether it is mostly free of typesetting problems and typos, it develops in great detail notation and foundational concepts, and appears to not take shortcuts. But at the same time, fundamental notations like $\mathcal{L}$ and $\hat{}$ that make up the core of the main result are not introduced previously. Even if I already know about Rademacher complexity bounds, the lack of details is noteworthy.
- What are the issues of extending the single-layer results to multiple-layers? Obviously fine to only have results for one layer, but they should be accompanied with a prominent discussion of how and what changes for the more general situation.

Other quick wins:
- I prefer the term "positively homogeneous" function, since "positive homogeneous" is most naturally parsed as "a homogeneous function that is positive".
- "parameterize" and "parametrize" are both used -- should be uniform.
- Typo "homoegenous" on page 2.
- I don't think that Figure 1 pulls its weight -- I'm obviously used to seeing diagrams like this, but usually when the visuals say something much more clearly than equations or words. I don't think that's the case here.
- We obtain a similar norm-based bound??? for general... [page 2]
- Group Convolutional Neural Networkss [page 1]
- I think it is well understood that the norm of an operation may be much lower than the condition number of its implied matrix (e.g. "Operator norm inequalities between tensor unfoldings on the partition lattice," by Wang). For example, I discovered this in just playing with spectral normalization in PyTorch. See E. K. Ryu, J. Liu, S. Wang, X. Chen, Z. Wang, and W. Yin. "Plug-and-Play Methods Provably Converge with Properly Trained Denoisers." ICML, 2019.)

**Strengths And Weaknesses:**

Positives:

 - Tackles an interesting and important question.
 - From what I can tell, the results appear correct.
 - The paper is mostly free of typos and errors and the presentation is done carefully (but see below).

Weaknesses:

 - The three main theorems seem to all be relatively simple extensions of the same basic result? There is a lower bound, but this appears already known.
 - The result appears to be mostly a kind of normalization: basically the product of the input magnitude, and the norms of the various operations of the block layers. These terms kind of have to feature (by simple Lipschitz continuity-type argumnts), and the relevance of dimension-free bounds is discussed mostly in an appendix.
 - The presentation seems quite inefficient: a lot of the real estate is devoted to laying out a powerful and general notation that, ultimately amounts to linear operators.
 - There are not really pedagogically useful examples, diagrams, or experiments to help engage readers not already very invested in the question. There is probably too much detail in areas, e.g. can the abstract group-theoretic stuff be moved to an appendix?

Conclusion: By its nature, the quality of this paper hinges entirely on the importance of Theorem 4.1, and I am sadly not familiar enough with the literature to judge this. My best-effort conclusion is that it's a nice result, but not the sort of high-impact finding that can carry a paper alone. I recommend that the authors either (1) make it clearer that the result advances the field via pointing to existing, open questions that it answers, or else (2) further develop consequences of the analysis to make the utility of the finding more evident and applicable (whilst addressing some problems). More concrete ideas follow. I do not recommend for publication currently, though if reviewers more expert than me find Theorem 4.1, this critique does not follow.

---

> ### Author Response · Authors · 2024-12-06
> **Answers to the reviewer - Part 1**
>
> We would like to thank the reviewer for the extensive feedbacks and suggestions. Particularly, the suggestion of adding CIFAR to the paper was very valuable, as the result aligned very well with our theoretical findings. We are grateful of that suggestion. According to the review, we made various changes to the paper that hopefully addresses the reviewer's questions. We also provide additional answers below.
>
>
> >Reviewer: The three main theorems seem to all be relatively simple extensions of the same basic result. There is a lower bound, but this appears already known.
> The result appears to be mostly a kind of normalization: basically the product of the input magnitude, and the norms of the various operations of the block layers. These terms kind of have to feature (by simple Lipschitz continuity-type arguments), and the relevance of dimension-free bounds is discussed mostly in an appendix.
>
> We understand the reviewer's view and would like to add some explanation. First, we believe that the lower bound is new and has not appeared before. In general, the lower bounds on the sample complexity are important to evaluate the tightness of the bound. Otherwise, the obtained upper bound on the sample complexity might just be loose. Theorem 4.3 shows that if we restrict ourselves to the class of networks with bounded norm, then our obtained upper bound is tight. Therefore, it is an important result in the paper. Besides, the main technical challenge of the paper is to formalize properly the equivariant architectures and adapt learning theoretic arguments for these models. Leveraging the prior works on statistical learning theory and adapt to the field of geometric deep learning is not straightforward as witnessed by lack of prior works in this area. Besides, our goal was not to contribute to the field of learning theory but rather bring it to the domain of goemetric deep learning and provide useful insights. Nonetheless, theoretical challenges had to be addressed along the way. Finally, we have new results and analysis that had not previously appeared nor could be infered from similar work. As examples, we would like to mention our analysis of locality and the connection with the uncertainty principle and the frequency domain analysis.
>
> >Reviewer: The presentation seems quite inefficient: a lot of the real estate is devoted to laying out a powerful and general notation that, ultimately amounts to linear operators.
>
> We have made various changes to the presentations, added new diagram and tables, and conducted new experiments.  Moreover, we note that while convolution (as well as group-convolution) is a special type of linear operation, its specific structure is the key characteristic giving neural network the equivariance property and, in particular, differentiating CNNs from MLPs. For this reason, we believe precisely describing the complex structure of the linear operations in our architectures is important for the clear presentation of our theoretical results.
>
>
> >Reviewer: There are not really pedagogically useful examples, diagrams, or experiments to help engage readers not already very invested in the question. There is probably too much detail in areas, e.g. can the abstract group-theoretic stuff be moved to an appendix?
>
> We have updated Figure 1 to summarize the main findings of the paper. We added a simple example of convolution to give some intuition behind group theory. We also added Figure 5 that visualizes the equivariant linear projection operation. Finally, upon writing the paper, we had to reflect on keeping only the essential group theory in the main paper, however, if the reviewer points us to the parts that could be moved to the appendix, we will be happy to consider it.
>
> >Reviewer: By its nature, the quality of this paper hinges entirely on the importance of Theorem 4.1, and I am sadly not familiar enough with the literature to judge this. My best-effort conclusion is that it's a nice result, but not the sort of high-impact finding that can carry a paper alone.
>
> Theorem 4.1 together with Theorem 4.3 show that the sample complexity of class of group equivariant networks with bounded norms can be \textit{tightly} bounded in a dimension free way. An interesting insight is that a large number of filters does not necessarily impact negatively the generalization error, and only the overall norm of the effective parameters matter. We also believe that the paper provides various useful insights through the theoretical results. We would like to highlight for example the impact filter size on the generalization error and the trade-off with the final performance.

---

> > ### Author Response · Authors · 2024-12-06
> > **Answers to the reviewer - Part 2**
> >
> > >Reviewer: I recommend that the authors either (1) make it clearer that the result advances the field via pointing to existing, open questions that it answers, or else (2) further develop consequences of the analysis to make the utility of the finding more evident and applicable (whilst addressing some problems). More concrete ideas follow.
> >
> > We would like to thank the reviewer for this suggestion. We have added a few more experiments that highlight the utility of the findings.
> >
> > >Reviewer: Give a generalization of the result beyond one-hidden-layer networks, even if quite naive. Obviously it's great to motivate the result with a simplified result, but the message would be more compelling if accompanied by some intuition for how it actually works (even if not even close to optimal -- anyone theoretical enough to appreciate this paper would have some intuition for translating simple bounds).
> >
> > We would like to thank the reviewer for the suggestion. We added a new theorem for multi-layer network. Our result, interestingly, does not have explicit dependence on the number of filters used across the layers, although it ends up with a dependence on the number of layers and the group size.
> >
> > >Reviewer: The entire paper is phrased in quite image-centric language (filters, convolutions, etc.), and yet the only demonstration of the idea is on MNIST. I would love to see a demonstration of the idea on RGB images, even just CIFAR10. For example, a few pictures showing rotated, permuted entries to some patch and the sensitivity of the block action could be pedagogically interesting.
> >
> > As recommended by the reviewer, we extended our empirical analysis in Section 8 with a number of experiments on RGB images from the CIFAR10 dataset. Moreover, we included a new Fig. 5 in Appendix K that shows an example of rotated images and the corresponding outputs of the initial convolution layer, highlighting the permutation action of the group on these feature vectors.
> >
> > >Reviewer: Develop everything first in a simple ReLU (or other whatever activation is easiest) + max pool and average pool framework, without needing to talk about Abelian groups or irreps, etc. The experiments are, I am sorry to say, very confusing. I'm expecting bounds to be some sort of line, all I see are a bunch of similarly-colored dots that kind of have some order? But not completely since there's 2 - 12, but also C-N, etc. Figure should stand on their own mostly, and these totally do not. I'm lost from the first sentence of the description (cyclic and dihedral groups? I kind of figure it out later, but a rocky start.), and I don't know what the MNIST data set is, what a steerable convolution is, etc.
> >
> > We hope our new plots have improved in terms of clarity. We have also extended the captions of our figures to better explain what data is being visualized, in order to make them stand alone.  Regarding the cyclic and dihedral groups, we only gave an intuitive description of them in the main paper to simplify the presentation, but now included a precise definition of them in the group theoretic appendix. For the new multi-layer result, we followed the reviewer's suggestion and focused on simple ReLU with average pooling. Finally, we welcome any suggestion about parts and sections that the reviewer think we should move to the appendix.
> >
> > >Reviewer: The abstract promises a careful analysis of weight sharing, equivariance, and locality, but I'm not sure I really see it in the results. If, indeed, there's a complete decomposition of the additivity of each effect, it would be good to perhaps have a table where examples of the presence and absence of each are given, along with the corresponding result. This table could also include existing results, for example it seems like the bound being dimension-free is important, and why this is so should not be in appendix J!
> >
> > We have added a new table in Appendix J to compare with the existing results as well as discussing each effect. Given all the new experimental results and the new theory, and given the space limitation, we still kept it in Appendix J. However, based on our undertanding, the final version can exceed 12 pages, and we will move it to the main paper.

---

> > > ### Author Response · Authors · 2024-12-06
> > > **Answers to the reviewer - part 3**
> > >
> > > >Reviewer: Related to the above, I would quite love to see a concrete motivation, such as the following: a generate data from a known distribution exhibiting weight sharing, equivariance, and locality and show how a model that takes account of these structures indeed generalizes better. Then tie the results to these illustrative examples. E.g. perhaps some of the examples could be constructed to obtain the upper or lower bounds.
> > >
> > > In the new table and the updated Figure 1, some of the comparisons are made more explicit. We would like to highlight that the proof of Theorem 4.3 is based on constructing an example that achieves the upper bound given in Theorem 4.1 and shows its tightness. We also had a discussion on weight-sharing mechanisms that were not optimal (please see the discussion at the end of Section 6). Proposition 7.2 and the discussions after it pointed to how locality can be effectively chosen.
> > >
> > > >Reviewer: Ideally there would be a truly simple "validation" of the bound where there is some toy class of problems where the bound is tight, and loosened. Then perhaps some empirical examination in a simple, but fully-featured problem, that does not artificially preprocess the data, or whatever (e.g. it follows some standard pipeline like a PyTorch example or minGPT). It's a real shame for nice theoretical work to be torpedoed by unconvincing experiments, so please don't skip out on this section!
> > >
> > > We thank the reviewer for this suggestion. We have added many experiments that try to clarify the intuition behind the bounds in various scenarios. We hope the new experiments are more convincing. Regarding the preprocessing step, we would like to provide some reasonings behind it. We adopted a linear projection via steerable convolution since it allows us to easily rotate the input features of our model without performing any interpolation of the pixel images. However, this step amounts to a simple (random, fixed) invertible linear operation which is analogous to vectorizing a pixel image before feeding it into an MLP, with a particular care to preserving the rotation and reflection equivariance in the process. We argue this does not artificially pre-process in any way that would affect our result, but simultaneously allows for 1) working with realistic image datasets while 2) maintaining a simple description of the equivariant networks. See also Figure 5 in Appendix K. We also agree that introducing steerable convolution in the main paper can cause confusion; for this reason, we prefer pointing to previous works on the topic for a precise description, and limit our presentation to an intuitive analogy which we added in the footnotes.
> > >
> > > >Reviewer: There are some jolting inconsistencies in the level of detail shown in different areas of the paper. Altogether it is mostly free of typesetting problems and typos, it develops in great detail notation and foundational concepts, and appears to not take shortcuts. But at the same time, fundamental notations like $\mathcal{L}$
> > >  and $\hat{\mathcal{L}}$ that make up the core of the main result are not introduced previously. Even if I already know about Rademacher complexity bounds, the lack of details is noteworthy.
> > >
> > > We are sorry for overseeing this. We have included the definition of these terms in the notation section as well as the section on Rademacher complexity in the appendix. We hope those sections contain enough details about Rademacher complexity and related tools. However, we are open to include further details.
> > >
> > >
> > > >Reviewer: What are the issues of extending the single-layer results to multiple-layers? Obviously fine to only have results for one layer, but they should be accompanied with a prominent discussion of how and what changes for the more general situation.
> > >
> > > We have included a new theorem for multi-layer networks in the paper. As we highlighted in the paper, the generalization behavior of deeper networks tend to be complicated and not easily explainable by the current state of the theory. That's partially why we focused on the case of single hidden layer network.

---

> > > > ### Author Response · Authors · 2024-12-06
> > > > **Answers to the reviewer - Part 4**
> > > >
> > > > >Reviewer: I prefer the term "positively homogeneous" function, since "positive homogeneous" is most naturally parsed as "a homogeneous function that is positive" - "parameterize" and "parametrize" are both used -- should be uniform.
> > > > Typo "homoegenous" on page 2  - I don't think that Figure 1 pulls its weight -- I'm obviously used to seeing diagrams like this, but usually when the visuals say something much more clearly than equations or words. I don't think that's the case here.
> > > > We obtain a similar norm-based bound??? for general... [page 2]; Group Convolutional Neural Networkss [page 1]
> > > >
> > > >
> > > >
> > > > We thank the reviewer for the noting the typos in the manuscript: we have now corrected them in the new version.
> > > > We have also updated Figure 1, which now gives an overview of the different architecture variations we considered and the associated theoretical bounds we derived. We have also fixed other issues raised by the reviewer. Also, we would be grateful if the reviewer could clarify the comment on "norm-based bound".
> > > >
> > > > >Reviewer: I think it is well understood that the norm of an operation may be much lower than the condition number of its implied matrix (e.g. "Operator norm inequalities between tensor unfoldings on the partition lattice," by Wang). For example, I discovered this in just playing with spectral normalization in PyTorch. See E. K. Ryu, J. Liu, S. Wang, X. Chen, Z. Wang, and W. Yin. "Plug-and-Play Methods Provably Converge with Properly Trained Denoisers." ICML, 2019.)
> > > >
> > > > We thank the reviewer for sharing these references. We have not used the condition number in the paper, and mentioned only the Frobenius norm and the spectral norm. We would be happy to clarify further if the reviewer could point us to the section of the paper that has created this confusion.

---

> > > > ### Comment · Reviewer_vmUN · 2024-12-09
> > > >
> > > > The deeper network bound isn't really what I had in mind since it's still just a single pooling operation on at the end. And the bound is just increased by the product of the operator norms of each of the layers? There is an interesting paper -- "Lipschitz regularity of deep neural networks: analysis and efficient estimation" that could give you some interesting techniques for improving this analysis.

---

> > > ### Comment · Reviewer_vmUN · 2024-12-09
> > >
> > > I like the table, thanks!

---

> > > ### Comment · Reviewer_vmUN · 2024-12-09
> > >
> > > Anything with a simple example + max pooling? The discussion around Theorem C.2 shows that max pooling has a bound that is different in nature than average pooling (unsuprising given that avg pooling is just more linearity). There are some works showing that max pooling is fundamentally different than average pooling such as "Benefits of Max Pooling in Neural Networks: Theoretical and Experimental Evidence".
> > >
> > > About more efficiently organizing the space: why not adopt the notation from another paper to avoid redeveloping it yourself? The nice thing about these sorts of notations for very general linear operation is that they can be adequately understood generally without tracking every detail.

---

> > > > ### Author Response · Authors · 2024-12-17
> > > > **Answer to the reviewer - follow up**
> > > >
> > > > We thank the reviewer for the prompt answer. We are glad to see that our change has addressed the reviewer's points. We will emphasize that the novelty of the lower bound in the final version.
> > > >
> > > > We provide further answers to some on the points above.
> > > >
> > > > >The deeper network bound isn't really what I had ...
> > > >
> > > > It is understandable that one might want to follow the common practice in convolutional neural networks (CNNs), and alternate pooling blocks one in every layer. Our pooling operation at the last layer turns the equivariant network into a group-invariant network, after which adding group convolution layers does not have justification. Alternatively, one can consider using partial pooling layers - which pool over sub-groups - in the intermediate layers. However, this is not common practice in geometric deep learning since it breaks the equivariance property (similarly, pooling breaks equivariance to small translations in CNNs and preserve only equivariance to coarser translations), but we can add that if the reviewer is interested.
> > > >
> > > > Regarding the bound for the multi-layers, we would like to clarify that we are not using the operator norm, but the norm of effective parameters. As we discussed in the paper, the operator (spectral norm) of a group circulant matrix can be larger, and is never smaller, than the norm of the effective parameters, which is the norm of a row of the matrix. We will look into the paper you mentioned in more details. The paper is an interesting contribution to the computation of Lipschitz constant of neural networks. In our analysis, Lipschitz constant is not used explicitly. The norm bound is used in the peeling argument step of the proof, and based on our initial investigation, it is not clear how to replace that with Lipschitz constant. Nonetheless, we will give it more thought.
> > > >
> > > > >Anything with a simple example + max pooling? ...
> > > >
> > > > Deriving bounds for the max pooling operation requires different techniques, as it was mentioned in the paper. The proof is usually lengthier but leads essentially to a similar result with an additional and generally milder dependency on the dimension (see Theorem 7 of Vardi et al.). Therefore, for the new multi-layer results, we just considered the average pooling. If the reviewer believes the max-pooling example can be crucial for the paper, we can add another theorem for max-pooling and re-work the proof accordingly.
> > > >
> > > > >About more efficiently organizing the space...
> > > >
> > > > This is a great suggestion. Out of experience, we noticed that some readers prefer to have a fully self-contained paper faithful to mathematical details of the paper, and therefore, we included the details as much as possible. We have tried, to our best effort, to re-use the notation from Vardi et al. 2022, Weiler et al, 2019, and Behboodi et al., 2022. As an example, the equation (3) tries to work only with function compositions and linear operations. Apart from Section 5, the rest of main sections use a similar abstraction. If the reviewer thinks that the notation in Section 5 can make the reading more difficult, we can move it to the supplementary materials.
> > > >
> > > > * Gal Vardi, Ohad Shamir, and Nathan Srebro. The Sample Complexity of One-Hidden-Layer Neural Networks.  In Conference on Neural Information Processing Systems (NeurIPS), 2022.
> > > >
> > > > * Maurice Weiler and Gabriele Cesa. General E(2)-Equivariant Steerable CNNs. In Conference on Neural Information Processing Systems (NeurIPS), 2019.
> > > >
> > > > * Arash Behboodi, Gabriele Cesa, and Taco S. Cohen. A pac-bayesian generalization bound for equivariant
> > > > networks. Advances in Neural Information Processing Systems, 35:5654–5668, 2022

---

> > > > > ### Comment · Reviewer_vmUN · 2024-12-18
> > > > >
> > > > > Hi,
> > > > > I won't have time to take a look at Vardi Theorem 7 at this point, but isn't a mild dependence on dimension contrary to the promise of dimension-free bounds? I think you should address this since average pooling is not often used in practice in computer vision architectures.
> > > > >
> > > > > Speaking of which, pooling over sub groups is widely used in practice, e.g. Alexnet. I don't know much abpout geometric deep learning, but it's always good to be inspired as much as possible by practice.
> > > > >
> > > > > The same motivation underlies my suggestion to tie your work to spectral norms / Lipschitz constants. These are well-understood quantities that -- whatever their theoretical properties, are important to the applied performance of networks with pooling.

---

> > ### Comment · Reviewer_vmUN · 2024-12-09
> >
> > Thanks for explaining the lower bound -- I would emphasize in the paper that it's novel!

---

> ### Comment · Reviewer_vmUN · 2024-12-09
>
> The deeper network bound isn't really what I had in mind since it's still just a single pooling operation on at the end. Ideally we'd want alternating nonlinearity plus pooling blocks. And the bound is just increased by the product of the operator norms of each of the layers? There is an interesting paper -- "Lipschitz regularity of deep neural networks: analysis and efficient estimation" that could give you some interesting techniques for improving this analysis.

---

> ### Author Response · Authors · 2024-12-19
> **Answer to the reviewer - follow up**
>
> Thanks much for the clarification. We will then include a result on max pooling and gradual pooling, although, since the interaction period finishes in the next few days, we might not be able to update the paper by then, but we will try our best. (
> **Update**: we know have the gradual pooling result in the paper - see C.5.1 A generalization bound for gradual pooling).
>
> Now, to answer some of the questions raised by the reviewer: it is true that the dimension dependence is not desirable. We believe the dimension dependence to be a proof artifact and will likely arise in our analysis as well. That is what we wanted to communicate above: max pooling results will give a similar result but with an additional dimension dependency, which is not desirable and not particularly more insightful. Regarding gradual pooling, to complement what the reviewer stated, we would like to note that standard practice in geometric deep learning is different and does not include gradual pooling. Furthermore, the pooling operation in AlexNet breaks the translation equivariant/invariant. For some papers related to this discussion, see for example [Zhang, 2019] and [Xu et al., 2021] for more details.
>
> * Zhang, Richard. Making convolutional networks shift-invariant again. ICML, 2019.
> * J Xu, H Kim, T Rainforth, Y Teh, Group equivariant subsampling, NeurIPS 2021

---

> > ### Author Response · Authors · 2025-01-03
> > **New version of the paper with new theorems**
> >
> > **Update:** we have now added new results on gradual pooling (see C.6.1) as well as a completely new result on max pooling (see C.3). The max pooling result can now be considered another new contribution of the paper. We had to use new techniques, based on covering number and Dudley's inequality, to prove the result. Although we tried to use a very simple network, the proof for max-pooling is more complicated and demanding. Besides, the mild-dimension dependencies appear as well in the paper, which highlights the theoretical challenge of working with max pooling operations. We hope we have addressed the reviewer's request.

---

### Author Response · Authors · 2024-12-06
**Summary of our answers**

We would like to thank all the reviewers for their valuable feedbacks. Their feedbacks has greatly improved our paper. To summarize, we have provided the following changes to the paper:

* We have updated the previous experiments with more details, and added new experiments on CIFAR10 dataset, and the impact of frequency domain locality on the generalization error. The experiments conform well with our theoretical findings.
* We added a new theorem for multi-layer equivariant networks. Although the bound is not fully dimension independent anymore, it is still full independent of the number of channels used in all the filters.
* We also updated the main figure of the paper and added a new table to summarize the main findings.

All the changes are with color red in the version. With the new experiments, we believe our papers brings learning theoretical insights to geometric deep learning, which are verified by our experimental results.

---

### Author Response · Authors · 2025-01-23
**Thanks to the reviewer and AC**

We would like to thank the reviewers and the AC for fruitful interactions during the review address. The camera ready version has been uploaded. We hope to have addressed the raised questions and requests in the final version. The submission focused on the single hidden layer networks. In the camera-ready version, upon the request of the reviewers, we have added new results on the networks with max-pooling and multi-layer networks, although these new results have mild dimension dependences. In the camera ready version, we have added new experimental results to substantiate further the theoretical results. We have also added various discussions on the limitations of the current work with ideas for future work.

---

### Decision · Action_Editor_oNYM · 2025-01-07

**Recommendation:** Accept with minor revision

**Comment:**

The paper investigates the generalization properties of 1 hidden layer neural networks, specifically focusing on how architectural choices like equivariance, locality, and weight-sharing impact sample complexity (as in convolutional networks).

The reviewers generally agree that the paper is technically sound and the results are correct. However, they have raised several concerns and suggestions to improve the paper. The authors made changes in response: improved presentation by moving some of the group-theoretic discussion to the appendix; included more concrete examples and diagrams; further explained the impact of locality on the generalization error. Additionally, concerns were raised about the limited empirical validation, and the restriction to one-hidden-layer networks. The authors added new experimental results that seem to be consistent with the theory, and included new theory for multi-layer equivariant networks (which, unlike in a 1 hidden layer case, has dimension dependence).

Some of the suggested edits for the final version:
 - Be more upfront about the limitations of the dimension-free bounds: acknowledge the mild dimension dependence in some cases and discuss its implications.
 - Expand the discussion of multi-layer networks: provide more insights into the challenges and potential approaches for analyzing deeper architectures (similarly, for max-pooling).
 - (related to the point above) Discuss the (lack of) susceptibility of this work to uniform convergence (UC) failures: the work by Nagarajan and Kolter points out the limitations of UC approaches for studying generalization in deep learning, include a discussion on how this paper circumvents the roadblocks of UC, at least in the one hidden layer case.
 - Expand related work section: the part discussing papers on generalization of convolutional neural networks should contain more details on related approaches. Right now it barely mentions some keywords, without making any connections to the current submission.

**Audience:**

The reviewers found the paper to be interesting and relevant to the TMLR audience. Equivariant neural networks are used in practical applications, and some TMLR readers might be interested in theoretical analyses of these models. Relevant theory on the role of weight sharing, locality, equivariance could have potential practical implications. This paper may inspire others working on similar theoretical questions to improve upon the proposed theory.

**Claims And Evidence:**

The reviewers generally agree that the paper is technically sound and the results are correct. The authors have provided sample complexity bounds for a one-hidden-layer networks, mostly leveraging Rademacher complexity analysis (at least for the original results). Some concerns about the clarity and strength of the results in support of all the claims remain. For example, the paper emphasizes dimension-free bounds, but the reviewers point out that some results have a mild dependence on dimension, particularly for max pooling and multi-layer equivariant network analysis (it is worth noting that these results were added to the paper during the rebuttal process).